# Necessary and sufficient graphical conditions for optimal adjustment sets in causal graphical models with hidden variables

**Jakob Runge**
German Aerospace Center
Institute of Data Science
07745 Jena, Germany
and
Technische Universität Berlin
10623 Berlin, Germany
jakob.runge@dlr.de

## Abstract

The problem of selecting optimal backdoor adjustment sets to estimate causal effects in graphical models with hidden and conditioned variables is addressed. Previous work has defined optimality as achieving the smallest asymptotic estimation variance and derived an optimal set for the case without hidden variables. For the case with hidden variables there can be settings where no optimal set exists and currently only a sufficient graphical optimality criterion of limited applicability has been derived. In the present work optimality is characterized as maximizing a certain adjustment information which allows to derive a necessary and sufficient graphical criterion for the existence of an optimal adjustment set and a definition and algorithm to construct it. Further, the optimal set is valid if and only if a valid adjustment set exists and has higher (or equal) adjustment information than the Adjust-set proposed in Perković et al. [Journal of Machine Learning Research, 18: 1–62, 2018] for any graph. The results translate to minimal asymptotic estimation variance for a class of estimators whose asymptotic variance follows a certain information-theoretic relation. Numerical experiments indicate that the asymptotic results also hold for relatively small sample sizes and that the optimal adjustment set or minimized variants thereof often yield better variance also beyond that estimator class. Surprisingly, among the randomly created setups more than 90% fulfill the optimality conditions indicating that also in many real-world scenarios graphical optimality may hold.

## 1 Introduction

A standard problem setting in causal inference is to estimate the causal effect between two variables given a causal graphical model that specifies qualitative causal relations among observed variables [Pearl, 2009], including a possible presence of hidden confounding variables. The graphical model then allows to employ graphical criteria to identify valid adjustment sets, the most well-known being the *backdoor criterion* [Pearl, 1993] and the *generalized adjustment criterion* [Shpitser et al., 2010, Perković et al., 2015, 2018], providing a complete identification of all valid adjustment sets. Estimators of causal effects based on such a valid adjustment set as a covariate are then unbiased, but for different adjustment sets the estimation variance may strongly vary. An *optimal adjustment set* may be characterized as one that has minimal asymptotic estimation variance. In **current work**, following Kuroki and Cai [2004] and Kuroki and Miyakawa [2003], Henckel et al. [2019] (abbreviated

HPM19 in the following) showed that graphical optimality always holds for linear models in the causally sufficient case where all relevant variables are observed. In Witte et al. [2020] an alternative characterization of the optimal adjustment set is discussed and the approach was integrated into the IDA algorithm [Maathuis et al., 2009, 2010] that does not require the causal graph to be known. Rotnitzky and Smucler [2019] extended the results in HPM19 to asymptotically linear non-parametric graphical models. HPM19's optimal adjustment set holds for the causally sufficient case (no hidden variables) and the authors gave an example with hidden variables where optimality does not hold in general, i.e., the optimal adjustment set depends on the coefficients and noise terms (more generally, the distribution), rather than just the graph. Most recently, Smucler et al. [2021] (SSR20) partially extended these results to the non-parametric hidden variables case together with *dynamic treatment regimes*, i.e., conditional causal effects. SSR20 provide a sufficient criterion for an optimal set to exist and a definition based on a certain undirected graph-construction using a result by van der Zander et al. [2019]. However, their sufficient criterion is very restrictive and a current major open problem is a *necessary* and sufficient condition for an optimal adjustment set to exist in the hidden variable case and a corresponding definition of an optimal set.

My **main theoretical contribution** is a solution to this problem. Optimality for conditional causal effects in the hidden variables case is fully characterized by an information-theoretic approach involving a certain difference of conditional mutual informations among the observed variables termed the adjustment information. Maximizing the adjustment information formalizes the common intuition to choose adjustment sets that maximally constrain the effect variable and minimally constrain the cause variable. This allows to derive a necessary and sufficient graphical criterion for the existence of an optimal adjustment set. The derived optimal adjustment set also has the property of minimum cardinality, i.e., no node can be removed without sacrificing optimality. Further, the optimal set is valid if and only if a valid adjustment set exists and has higher (or equal) adjustment information than the Adjust-set proposed in Perković et al. [2018] for any graph, whether graphical optimality holds or not. The results translate to minimal asymptotic estimation variance for a class of estimators whose asymptotic variance follows a certain information-theoretic relation that, at present, I could only verify theoretically for the linear case. As **practical contributions** the paper provides extensive numerical experiments that corroborate the theoretical results and show that the optimal adjustment set or minimized variants thereof often yield better variance also beyond the theoretically analyzed estimator class. Code is available in the python package `https://github.com/jakobrunge/tigramite`. More detailed preliminaries, proofs, algorithms, and further numerical experiments are given in the Supplementary Material.

## 1.1 Preliminaries and problem setting

We consider causal effects in causal graphical models over a set of variables $\mathbf{V}$ with a joint distribution $\mathcal{P} = \mathcal{P}(\mathbf{V})$ that is consistent with an acyclic directed mixed graph (ADMG) $\mathcal{G} = (\mathbf{V}, \mathcal{E})$. Two nodes can have possibly more than one edge which can be *directed* ($\leftarrow$) or *bi-directed* ($\leftrightarrow$). See Fig. 1A for an example. Kinships are defined as usual: parents $pa(X)$ for "$\bullet \rightarrow X$", spouses $sp(X)$ for "$X \leftrightarrow \bullet$", children $ch(X)$ for "$X \rightarrow \bullet$". These sets all exclude $X$. Correspondingly descendants $des(X)$ and ancestors $an(X)$ are defined, which, on the other hand, both include $X$. The mediator nodes on causal paths from $X$ to $Y$ are denoted $\mathbf{M} = \mathbf{M}(X, Y)$ and exclude $X$ and $Y$. For detailed preliminaries, including the definition of open and blocked paths, see Supplementary Section A. In this work we only consider a univariate intervention variable $X$ and effect variable $Y$. We simplify set notation and denote unions of variables as $\{W\} \cup \mathbf{M} \cup \mathbf{A} = W\mathbf{MA}$.

A (possibly empty) set of adjustment variables $\mathbf{Z}$ for the total causal effect of $X$ on $Y$ in an ADMG is called *valid* relative to $(X, Y)$ if the interventional distribution for setting $do(X = \mathbf{x})$ [Pearl, 2009] factorizes as $p(Y|do(X = \mathbf{x})) = \int p(Y|\mathbf{x}, \mathbf{z})p(\mathbf{z})d\mathbf{z}$ for non-empty $\mathbf{Z}$ and as $p(Y|do(X = \mathbf{x})) = p(Y|\mathbf{x})$ for empty $\mathbf{Z}$. Valid adjustment sets, the set of which is here denoted $\mathcal{Z}$, can be read off from a given causal graph using the generalized adjustment criterion [Perković et al., 2015, 2018] which generalizes Pearl's back-door criterion [Pearl, 2009]. To this end define

$$\mathbf{forb}(X, Y) = X \cup des(Y\mathbf{M}) \tag{1}$$

(henceforth just denoted as $\mathbf{forb}$). A set $\mathbf{Z}$ is valid if both of the following conditions hold: (i) $\mathbf{Z} \cap \mathbf{forb} = \emptyset$, and (ii) all non-causal paths from $X$ to $Y$ are blocked by $\mathbf{Z}$. An adjustment set is called *minimal* if no strict subset of $\mathbf{Z}$ is still valid. The validity conditions can in principle be manually checked directly from the graph, but, more conveniently, Perković et al. [2018] define an

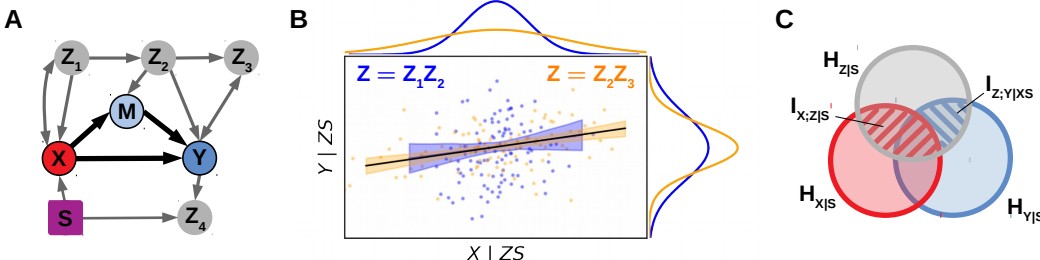

Figure 1: (**A**) Problem setting of optimal adjustment sets in causal graphs with hidden variables represented through bi-directed edges. The goal is to estimate the total causal effect of $X$ on $Y$ potentially through mediators $\mathbf{M}$, and given conditioned variables $\mathbf{S}$. The task is to select a valid adjustment set $\mathbf{Z}$ such that the estimator has minimal asymptotic variance. (**B**) illustrates two causal effect estimates for a linear Gaussian model consistent with the graph in **A**, see discussion in text. (**C**) For a certain class of estimators a minimal asymptotic estimation variance can be translated into an information-theoretical optimization problem, here visualized in a Venn diagram. An optimal adjustment set $\mathbf{Z}$ must maximize the adjustment information $I_{\mathbf{Z};Y|X\mathbf{S}} - I_{X;\mathbf{Z}|\mathbf{S}}$ (blue and red hatched, respectively).

adjustment set called 'Adjust' that is valid if and only if a valid adjustment set exist. In our setting including conditioning variables $\mathbf{S}$ we call this set the *valid ancestors* defined as

$$\mathbf{vancs}(X, Y, \mathbf{S}) = an(XY\mathbf{S}) \setminus \mathbf{forb} \tag{2}$$

and refer to this set as **vancs** or Adjust-set.

Our quantity of interest is the average total causal effect of an intervention to set $X$ to $x$ vs. $x'$ on the effect variable $Y$ given a set of selected (conditioned) variables $\mathbf{S} = \mathbf{s}$

$$\Delta_{yxx'|\mathbf{s}} = E(Y|do(x), \mathbf{s}) - E(Y|do(x'), \mathbf{s}). \tag{3}$$

We denote an estimator given a valid adjustment set $\mathbf{Z}$ as $\widehat{\Delta}_{yxx'|\mathbf{s}.\mathbf{z}}$. In the linear case $\Delta_{yxx'|\mathbf{s}}$ for $x = x' + 1$ corresponds to the regression coefficient $\beta_{YX.\mathbf{ZS}}$ in the regression of $Y$ on $X$, $\mathbf{Z}$, and $\mathbf{S}$. The ordinary least squares (OLS) estimator $\hat{\beta}_{YX.\mathbf{ZS}}$ is a consistent estimator of $\beta_{YX.\mathbf{ZS}}$.

Figure 1A illustrates the **problem setting**: We are interested in the total causal effect of (here univariate) $X$ on $Y$ (conditioned on $\mathbf{S}$), which is here due to a direct link and an indirect causal path through a mediator $M$. There are six valid backdoor adjustment sets $\mathcal{Z} = \{Z_1, Z_2, Z_1Z_2, Z_2Z_3, Z_1Z_3, Z_1Z_2Z_3\}$. $Z_4 \in \mathbf{forb}$ cannot be included in any set because it is a descendant of $Y\mathbf{M}$. Here $\mathbf{vancs} = Z_1Z_2S$. All valid adjustment sets remove the bias due to confounding by their definition. The question is which of these valid adjustment sets is statistically optimal in that it minimizes the asymptotic estimation variance? More formally, the task is, given a graph $\mathcal{G}$ and $(X, Y, \mathbf{S})$, to chose a valid optimal set $\mathbf{Z}_{\text{optimal}} \in \mathcal{Z}$ such that the causal effect estimator's asymptotic variance $\text{Var}(\widehat{\Delta}_{yxx'|\mathbf{s}.\mathbf{z}}) = E[(\Delta_{yxx'|\mathbf{s}} - \widehat{\Delta}_{yxx'|\mathbf{s}.\mathbf{z}})^2]$ is minimal:

$$\mathbf{Z}_{\text{optimal}} \in \operatorname{argmin}_{\mathbf{Z} \in \mathcal{Z}} \text{Var}(\widehat{\Delta}_{yxx'|\mathbf{s}.\mathbf{z}}). \tag{4}$$

My proposed approach to optimal adjustment sets is based on information theory [Cover and Thomas, 2006]. The main quantity of interest there is the conditional mutual information (CMI) defined as a difference $I_{X;Y|Z} = H_{Y|Z} - H_{Y|ZX}$ of two (conditional) Shannon entropies $H_{Y|X} = -\int_{x,y} p(x,y) \ln p(y|x) dx dy$. Its main properties are non-negativity, $I_{X;Y|Z} = 0$ if and only if $X \perp\!\!\!\perp Y|Z$, and the chain rule $I_{XW;Y|Z} = I_{X;Y|Z} + I_{W;Y|ZX}$. All random variables in a CMI can be multivariate.

Throughout the present paper we will assume the following.

**Assumptions 1** (General setting and assumptions). *We assume a causal graphical model over a set of variables $\mathbf{V}$ with a joint distribution $\mathcal{P} = \mathcal{P}(\mathbf{V})$ that is consistent with an ADMG $\mathcal{G} = (\mathbf{V}, \mathcal{E})$. We assume a non-zero causal effect from $X$ on $Y$, potentially through a set of mediators $\mathbf{M}$, and given selected conditioned variables $\mathbf{S}$, where $\mathbf{S} \cap \mathbf{forb} = \emptyset$. We assume that at least one valid adjustment set (given $\mathbf{S}$) exists and, hence, the causal effect is identifiable (except when stated otherwise). Finally, we assume the usual Causal Markov Condition (implicit in semi-Markovian models) and Faithfulness.*

## 2 Optimal adjustment sets

### 2.1 Information-theoretic characterization

Figure 1B illustrates two causal effect estimates for a linear Gaussian model consistent with the graph in Fig. 1A. With $\mathbf{Z} = Z_1 Z_2$ (blue) the error is much larger than with $\mathbf{O} = Z_2 Z_3$ (orange) for two reasons: $\mathbf{Z}$ constrains the residual variance $Var(Y|\mathbf{ZS})$ of the effect variable $Y$ less than $\mathbf{O}$ and, on the other hand, $\mathbf{Z}$ constrains the residual variance $Var(X|\mathbf{ZS})$ of the cause variable $X$ more than $\mathbf{O}$. Smaller estimator variance also holds for $\mathbf{O}$ compared to any other valid set in $\mathcal{Z}$ here.

We information-theoretically formalize the resulting intuition to choose an adjustment set $\mathbf{Z}$ that maximally constrains the effect variable $Y$ and minimally constrains the cause variable $X$. In terms of CMIs and given selected fixed conditions $\mathbf{S}$ the quantity to maximize can be stated as follows.

**Definition 1** (Adjustment information)**.** *Consider a causal effect of $X$ on $Y$ for an adjustment set $\mathbf{Z}$ given a condition set $\mathbf{S}$. The* (conditional) adjustment (set) information*, abbreviated $J_{\mathbf{Z}}$, is defined as*

$$J_{XY|\mathbf{S}.\mathbf{Z}} \equiv I_{\mathbf{Z};Y|X\mathbf{S}} - I_{X;\mathbf{Z}|\mathbf{S}} \tag{5}$$
$$= \underbrace{H_{Y|X\mathbf{S}} - H_{X|\mathbf{S}}}_{\text{not related to } \mathbf{Z}} - \underbrace{(H_{Y|X\mathbf{ZS}} - H_{X|\mathbf{ZS}})}_{\text{adjustment entropy}} \tag{6}$$

$J_{\mathbf{Z}}$ is not necessarily positive if the dependence between $X$ and $\mathbf{Z}$ (given $\mathbf{S}$) is larger than that between $\mathbf{Z}$ and $Y$ given $X\mathbf{S}$. Equation (6) follows from the CMI definition. Fig. 1C illustrates the two CMIs in Eq. (5) in a Venn diagram.

Before discussing the range of estimators for which maximizing the adjustment information $J_{\mathbf{Z}}$ leads to a minimal asymptotic estimation variance in Sect. 2.2, we characterize graphical optimality in an information-theoretic framework. Our goal is to provide graphical criteria for optimal adjustment sets, i.e., criteria that depend only on the structure of the graph $\mathcal{G}$ and not on the distribution.

**Definition 2** (Information-theoretical graphical optimality)**.** *Given Assumptions 1 we say that* (information-theoretical) graphical optimality holds *if there is a $\mathbf{Z} \in \mathcal{Z}$ such that either there is no other $\mathbf{Z}' \neq \mathbf{Z} \in \mathcal{Z}$ or for all other $\mathbf{Z}' \neq \mathbf{Z} \in \mathcal{Z}$ and* all *distributions $\mathcal{P}$ consistent with $\mathcal{G}$ we have $J_{\mathbf{Z}} \geq J_{\mathbf{Z}'}$.*

My main result builds on the following lemma which relates graphical optimality to information-theoretic inequalities in a necessary and sufficient comparison condition for an optimal set to exist.

**Lemma 1** (Necessary and sufficient comparison criterion for existence of an optimal set)**.** *Given Assumptions 1, if and only if there is a $\mathbf{Z} \in \mathcal{Z}$ such that either there is no other $\mathbf{Z}' \neq \mathbf{Z} \in \mathcal{Z}$ or for all other $\mathbf{Z}' \neq \mathbf{Z} \in \mathcal{Z}$ and all distributions $\mathcal{P}$ consistent with $\mathcal{G}$ it holds that*

$$\underbrace{I_{\mathbf{Z}\backslash\mathbf{Z}';Y|\mathbf{Z}'X\mathbf{S}}}_{(i)} \geq \underbrace{I_{\mathbf{Z}'\backslash\mathbf{Z};Y|\mathbf{Z}X\mathbf{S}}}_{(iii)} \quad and \quad \underbrace{I_{X;\mathbf{Z}'\backslash\mathbf{Z}|\mathbf{Z}\mathbf{S}}}_{(ii)} \geq \underbrace{I_{X;\mathbf{Z}\backslash\mathbf{Z}'|\mathbf{Z}'\mathbf{S}}}_{(iv)}, \tag{7}$$

*then graphical optimality holds and $\mathbf{Z}$ is optimal implying $J_{\mathbf{Z}} \geq J_{\mathbf{Z}'}$.*

In SSR20 and HPM19 the corresponding conditional independence statements to the terms (iii) and (iv) in the inequalities (7) are used as a sufficient pairwise comparison criterion. However, Lemma 1 shows that for graphical optimality it is not necessary that terms (iii) and (iv) vanish, they just need to fulfill the inequalities (7) for a *necessary* and sufficient criterion.

In principle, Lemma 1 can be used to cross-compare all pairs of sets, but firstly, it is difficult to explicitly evaluate (7) for all distributions $\mathcal{P}$ consistent with $\mathcal{G}$ and, secondly, iterating through all valid adjustment sets is computationally prohibitive even for small graph sizes. As an example, consider a confounding path consisting of 5 nodes. Then this path can be blocked by $2^5 - 1$ different subsets. In the main result of this work (Thm. 3) a necessary and sufficient criterion based purely on graphical properties is given.

### 2.2 Applicable estimator class

The above characterization only relates optimality of adjustment sets to the adjustment information $J_{\mathbf{Z}}$ defined in Eq. (5), but not to any particular estimator. Now the question is for which class of

causal effect estimators $\widehat{\Delta}_{yxx'|\mathbf{s}.\mathbf{z}}$ the intuition of maximizing the adjustment information $J_{\mathbf{Z}}$ leads to a minimal asymptotic estimation variance. In its most general form this class is characterized as fulfilling

$$\mathbf{Z}_{\text{optimal}} \in \text{argmax}_{\mathbf{Z} \in \mathcal{Z}} J_{\mathbf{Z}} \; \Leftrightarrow \; \text{Var}(\widehat{\Delta}_{yxx'|\mathbf{s}.\mathbf{z}_{\text{optimal}}}) = \min_{\mathbf{Z} \in \mathcal{Z}} \text{Var}(\widehat{\Delta}_{yxx'|\mathbf{s}.\mathbf{z}}), \tag{8}$$

where we assume that $\widehat{\Delta}_{yxx'|\mathbf{s}.\mathbf{z}}$ is consistent due to a valid adjustment set and correct functional model specification. One can also further restrict the class to estimators whose (square-root of the) asymptotic variance can be expressed as

$$\sqrt{\text{Var}(\widehat{\Delta}_{yxx'|\mathbf{s}.\mathbf{z}})} = f(H_{Y|X\mathbf{ZS}} - H_{X|\mathbf{ZS}}), \tag{9}$$

for a real-valued, strictly monotonously increasing function of the adjustment entropy. Minimizing the adjustment entropy is by Eq. (6) equivalent to maximizing the adjustment information. The following assumption and lemma then relates $J_{\mathbf{Z}} \geq J_{\mathbf{Z}'}$ to the corresponding asymptotic variances of a given estimator.

**Assumptions 2** (Estimator class assumption). *The model class of the estimator for the causal effect* (3) *is correctly specified and its asymptotic variance can be expressed as in relation* (9).

**Lemma 2** (Asymptotic variance and adjustment information). *Given Assumptions 1 and an estimator fulfilling Assumptions 2, if and only if for two different adjustment sets $\mathbf{Z}, \mathbf{Z}' \in \mathcal{Z}$ we have $J_{\mathbf{Z}} \geq J_{\mathbf{Z}'}$, then the adjustment set $\mathbf{Z}$ has a smaller or equal asymptotic variance compared to $\mathbf{Z}'$.*

*Proof.* By Equations (6) and (9) $J_{\mathbf{Z}} \geq J_{\mathbf{Z}'}$ (for fixed $X, Y, \mathbf{S}$) is directly related to a smaller or equal asymptotic variance for $\mathbf{Z}$ compared to $\mathbf{Z}'$, and vice versa. □

The paper's theoretical results currently hold for estimators fulfilling relation (9), but at least the main result on graphical optimality in Thm. 3 can also be relaxed to estimators fulfilling the less restrictive relation (8). In this work, we leave the question of which general classes of estimators fulfill either relation (8) or the more restricted relation (9) to further research and only show that it holds for the OLS estimator $\widehat{\beta}_{YX \cdot \mathbf{ZS}}$ for Gaussian distributions.

For Gaussians the entropies in (9) are given by $H(Y|X\mathbf{ZS}) = \frac{1}{2} + \frac{1}{2}\ln(2\pi\sigma^2_{Y|X\mathbf{ZS}})$ and $H(X|\mathbf{ZS}) = \frac{1}{2} + \frac{1}{2}\ln(2\pi\sigma^2_{X|\mathbf{ZS}})$ where $\sigma(\cdot|\cdot)$ denotes the square-root of the conditional variance. Then

$$\sqrt{\text{Var}(\widehat{\Delta}_{yxx'|\mathbf{s}.\mathbf{z}})} = \frac{1}{\sqrt{n}} e^{H_{Y|X\mathbf{ZS}} - H_{X|\mathbf{ZS}}} = \frac{1}{\sqrt{n}} \frac{\sigma_{Y|X\mathbf{ZS}}}{\sigma_{X|\mathbf{ZS}}}. \tag{10}$$

This relation is also the basis of the results for the causally sufficient case in Henckel et al. [2019] where it is shown that it holds more generally for causal linear models that do not require the noise terms to be Gaussian.

## 2.3 Definition of O-set

The optimal adjustment set for the causally sufficient case is simply $\mathbf{P} = pa(Y\mathbf{M}) \setminus \mathbf{forb}$ and was derived in HPM19 and Rotnitzky and Smucler [2019]. In Section B.2 the derivation is discussed from an information-theoretic perspective. In the case with hidden variables we need to account for bidirected edges "$\leftrightarrow$" which considerably complicate the situation. Then the parents of $Y\mathbf{M}$ are not sufficient to block all non-causal paths. Further, just like conditioning on parents of $Y\mathbf{M}$ leads to optimality in the sufficient case since parents constrain information in $Y\mathbf{M}$, in the hidden variables case also conditioning on spouses of $Y\mathbf{M}$ constrains information about $Y\mathbf{M}$.

**Example A.** A simple graph (ADMG) to illustrate this is $X \to Y \leftrightarrow Z_1$ (shown with an additional $\mathbf{S}$ in Fig. 2A below, or Fig. 4 in SSR20). Here $\mathbf{Z}' = \emptyset = \mathbf{vancs}$ is a valid set, but it is not optimal. Consider $\mathbf{O} = Z_1$, then term (iii) $= 0$ in the inequalities (7) since $\mathbf{Z}' \setminus \mathbf{O} = \emptyset$. Even though not needed to block non-causal paths (there is none), $Z_1$ still constrains information in $Y$ while being independent of $X$ (hence, term (iv) $= 0$) which leads to $J_{\mathbf{O}} > J_{\emptyset}$ according to the inequalities (7).

Not only direct spouses can constrain information in $Y$ as Fig. 2B below illustrates. Since for $W \in Y\mathbf{M}$ the motif "$W \leftrightarrow \boxed{C_1} \leftarrow\ast C_2$" ("$\ast$" denotes either edge mark) is open, it holds that $I(C_1 C_2; W) = I(C_1; Y) + I(C_2; W|C_1) \geq I(C_1; Y)$ and we can even further increase the first term in the adjustment

information by conditioning also on subsequent spouses. This chain of colliders only ends if we reach a tail or there is no further adjacency. However, we have to make sure that conditioning on colliders does not open non-causal paths. This leads to the notion of a *valid collider path* (related to the notion of a *district* in Evans and Richardson [2014]).

**Definition 3** (Valid collider paths). *Given a graph $\mathcal{G}$, a collider path of $W$ for $k \geq 1$ is defined by a sequence of edges $W \leftrightarrow C_1 \leftrightarrow \cdots \leftrightarrow C_k$. We denote the set of path nodes (excluding $W$) along a path indexed by $i$ as $\pi_W^i$. Using the set of valid ancestors $\mathbf{vancs} = an(XY\mathbf{S}) \setminus \mathbf{forb}$ for the causal effect of $X$ on $Y$ given $\mathbf{S}$ we call a collider path node set $\pi_W^i$ for $W \in Y\mathbf{M}$ valid wrt. to $(X, Y, \mathbf{S})$ if for each path node $C \in \pi_W^i$ both of the following conditions are fulfilled:*

$$(1)\ C \notin \mathbf{forb}, \quad and \quad (2a)\ C \in \mathbf{vancs}\ or\ (2b)\ C \perp\!\!\!\perp X \mid \mathbf{vancs}. \tag{11}$$

Condition (1) is required for any valid adjustment set. If jointly (2a) and (2b) are not fulfilled, i.e. $C \notin \mathbf{vancs}$ and $C \not\perp\!\!\!\perp X \mid \mathbf{vancs}$, then the collider path stops before $C$. Our candidate optimal adjustment set is now constructed based on the parents of $Y\mathbf{M}$, valid collider path nodes of $Y\mathbf{M}$, and their parents to 'close' these collider paths.

**Definition 4** (O-set). *Given Assumptions 1 and the definition of valid colliders in Def. 3, define the set $\mathbf{O}(X, Y, \mathbf{S}) = \mathbf{P} \cup \mathbf{C} \cup \mathbf{P_C}$ where*

$$\mathbf{P} = pa(Y\mathbf{M}) \setminus \mathbf{forb}, \quad \mathbf{C} = \uplus_{W \in Y\mathbf{M}} \uplus_i \{\pi_W^i : \ \pi_W^i \text{ is valid wrt. to } (X, Y, \mathbf{S})\}, \quad \mathbf{P_C} = pa(\mathbf{C}).$$

In the following we will abbreviate $\mathbf{O} = \mathbf{O}(X, Y, \mathbf{S})$. Algorithm C.1 states efficient pseudo-code to construct the $\mathbf{O}$-set and detect whether a valid adjustment set exists. Since none of the conditions of Def. 3 for adding collider nodes depends on previously added nodes, the algorithm is order-independent. The statement occurring in lines 11 and 21 ("No valid adjustment set exists.") is proven in Thm. 1. If the graph is a DAG, then lines 4-22 can be omitted. The algorithm is of low complexity and the most time-consuming part is checking for a path in line 12, Def. 3(2b) $C \perp\!\!\!\perp X \mid \mathbf{vancs}$, which can be implemented with (bi-directional) breadth-first search as proposed in van der Zander et al. [2019].

Numerical experiments in Section 3 will show that further interesting adjustment sets are the *minimized* $\mathbf{O}$-set $\mathbf{O}_{\min}$, where $\mathbf{O}$ is minimized such that no subset can be removed without making $\mathbf{O}_{\min}$ invalid, and the *collider-minimized* $\mathbf{O}$-set $\mathbf{O}_{\mathrm{Cmin}}$ where only $\mathbf{CP_C} \setminus \mathbf{P} \subseteq \mathbf{O}$ is minimized such that no collider-subset can be removed without making $\mathbf{O}_{\mathrm{Cmin}}$ invalid. Both adjustment sets can be constructed with Alg. C.2 similar to the efficient algorithms in van der Zander et al. [2019]. Also the minimized sets are order-independent since the nodes are removed only after the for-loops. Based on the idea in $\mathbf{O}_{\mathrm{Cmin}}$, in the numerical experiments we also consider $\mathrm{Adjust}_{\mathrm{Xmin}}$, where only $\mathrm{Adjust} \setminus pa(Y\mathbf{M})$ is minimized and $pa(Y\mathbf{M})$ is always included. Finally, we also evaluate $\mathrm{Adjust}_{\min}$ where Adjust is fully minimized.

Before discussing the optimality of the $\mathbf{O}$-set, we need to assure that it is a valid adjustment set. Similar to the proof given in Perković et al. [2018] for the validity of the $\mathbf{vancs}$-set (for the case without $\mathbf{S}$), we can state that the $\mathbf{O}$-set is valid if and only if a valid adjustment set exists.

**Theorem 1** (Validity of O-set). *Given Assumptions 1 but* without *a priori assuming that a valid adjustment set exists (apart from the requirement $\mathbf{S} \cap \mathbf{forb} = \emptyset$). If and only if a valid backdoor adjustment set exists, then $\mathbf{O}$ is a valid adjustment set.*

## 2.4 Graphical optimality

We now move to the question of optimality. It is known that there are graphs where no graphical criterion exists to determine optimality. Examples, discussed later, are the graphs in Figs. 2E,F.

Before stating necessary and sufficient conditions for graphical optimality, I mention that next to the $\mathbf{O}$-set defined above and the Adjust set $\mathbf{vancs}$ [Perković et al., 2018], I am not aware of any other systematically constructed set that will yield a valid adjustment set for the case with hidden variables. van der Zander et al. [2019] provide algorithms to list all valid adjustment sets, but the question is which of these a user should choose. As mentioned above, Lemma 1 can be used to cross-compare all pairs of sets, but this is not really feasible. Hence, for automated causal effect estimation, rather than the question of whether graphical optimality holds, it is crucial to have a set with better properties than other systematically constructable sets. The following theorem states that

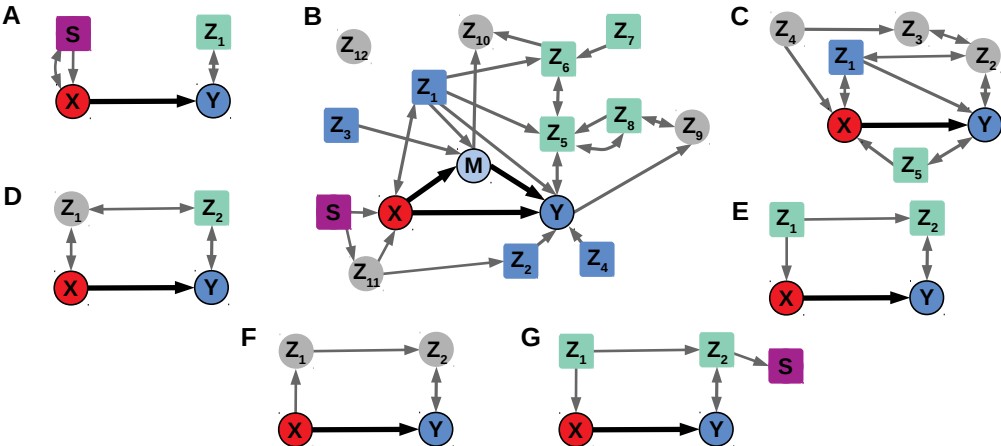

Figure 2: Examples illustrating (optimal) adjustment sets. In all examples the causal effect along causal paths (thick black edges) between $X$ (red circle) and $Y$ (blue circle) potentially through mediators $\mathbf{M}$ (light blue circle), and conditioned on some variables $\mathbf{S}$ (purple box), is considered. The adjustment set $\mathbf{O}$ consists of $\mathbf{P}$ (blue boxes) and $\mathbf{CP_C} \setminus \mathbf{P}$ (green boxes). See main text for details.

the adjustment informations follow $J_\mathbf{O} \geq J_\mathbf{vancs}$ for *any* graph (whether graphical optimality holds or not).

**Theorem 2** (O-set vs. Adjust-set ). *Given Assumptions 1 with $\mathbf{O}$ defined in Def. 4 and the Adjust-set defined in Eq. (2), it holds that $J_\mathbf{O} \geq J_\mathbf{vancs}$ for any graph $\mathcal{G}$. We have $J_\mathbf{O} = J_\mathbf{vancs}$ only if (1) $\mathbf{O} = \mathbf{vancs}$, or (2) $\mathbf{O} \subseteq \mathbf{vancs}$ and $X \perp\!\!\!\perp \mathbf{vancs} \setminus \mathbf{O} \mid \mathbf{OS}$.*

In the following the $\mathbf{O}$-set is illustrated and conditions for graphical optimality are explored. SSR20 provide a sufficient condition for optimality, which states that either all nodes are observed (no bidirected edges exist) or for all observed nodes $\mathbf{V} \subset \mathbf{vancs}$. This is a very strict assumption and not fulfilled for any of the examples (except for Example G) discussed in the following.

**Example B.** Figure 2B depicts a larger example to illustrate the $\mathbf{O}$-set $\mathbf{O} = \mathbf{PCP_C}$ with $\mathbf{P} = Z_1 Z_2 Z_3 Z_4$ (blue boxes) and $\mathbf{CP_C} \setminus \mathbf{P} = Z_5 Z_6 Z_7 Z_8$ (green boxes). We also have a conditioned variable $\mathbf{S}$. Among $\mathbf{P}$, only $Z_1 Z_2$ are needed to block non-causal paths to $X$, $Z_3 Z_4$ are only there to constrain information in $Y$. Here the same holds for the whole set $\mathbf{CP_C} \setminus \mathbf{P}$ which was constructed from the paths $Y \leftrightarrow Z_5 \leftrightarrow Z_6 \leftarrow Z_7$ and $Y \leftrightarrow Z_5 \leftrightarrow Z_8$ which does not include $Z_9$ since it is a descendant of $Y\mathbf{M}$. Including an independent variable like $Z_{12}$ in $\mathbf{O}$ would not decrease the adjustment information $J_\mathbf{O}$, but then $\mathbf{O}$ would not be of minimum cardinality anymore (proven in Cor. B.1). Here, again, the condition of SSR20 does not hold (e.g., $Z_5$ is not an ancestor of $XY\mathbf{S}$). $\mathbf{O}$ is optimal here which can be seen as follows: For term (iii) in the inequalities (7) to even be non-zero, we would need a valid $\mathbf{Z}$ such that $\mathbf{Z} \setminus \mathbf{O}$ has a path to $Y$ given $\mathbf{OS}X$. But these are all blocked. Note that while $Z_{10}$ or $Z_9 \in \mathbf{Z}$ would open a path to $Y$, both of these are descendants of $\mathbf{M}$ or $Y$ and, hence, cannot be in a valid $\mathbf{Z}$. For term (iv) to even be non-zero $\mathbf{O} \setminus \mathbf{Z}$ would need to have a path to $X$ given $\mathbf{ZS}$. But since any valid $\mathbf{Z}$ has to contain $Z_1$ and $Z_2$ (or $Z_{11}$), the only nodes in $\mathbf{O}$ with a path to $X$ are parents of $Y\mathbf{M}$ and paths from these parents to $X$ all need to be blocked for a valid $\mathbf{Z}$. Hence, $\mathbf{O}$ is optimal here.

**Example C.** In Fig. 2C a case is shown where $\mathbf{O} = Z_1 Z_5$. $Z_2$ is not part of $\mathbf{O}$ because none of the conditions in Def. 3(2) is fulfilled: $Z_2 \notin \mathbf{vancs} = Z_1 Z_4 Z_5$ and $Z_2 \not\perp\!\!\!\perp X \mid \mathbf{vancs}$. Hence, we call $Z_2$ an N-node. But $Z_2$ cannot be part of any valid $\mathbf{Z}$ because it has a collider path to $X$ through $Z_1$ which is always open because it is part of $\mathbf{vancs}$. Hence, term (iii) is always zero. Term (iv) is zero because $\mathbf{O} \setminus \mathbf{Z}$ is empty for any valid $\mathbf{Z}$ here. Here even $J_\mathbf{O} > J_\mathbf{Z}$ since $\mathbf{O}$ is minimal and term (ii) $I_{X;\mathbf{Z} \setminus \mathbf{O} \mid \mathbf{O}} > 0$ for any $\mathbf{Z} \neq \mathbf{O}$ (generally proven in Corollary B.1).

**Example D.** The example in Fig. 2D depicts a case with $\mathbf{O} = Z_2$ where $Z_1$ is an N-node. Next to $\mathbf{Z} = \emptyset$ another valid set is $\mathbf{Z} = Z_1$. Then term (iii) is non-zero and in the same way term (iv) is non-zero. The sufficient pairwise comparison criterion in SSR20 and HPM19 is, hence, not applicable. However, it holds that always term (iii) $\leq$ (i) because the dependence between $Z_1$ and $Y$ given $X$ is

always smaller than the dependence between $Z_2$ and $Y$ given $X$ and correspondingly term (iv) $\leq$ (ii). Hence, **O** is optimal here. If a link $Y \rightarrow Z_1$ exists, then the only other valid set is $\mathbf{Z} = \emptyset$ and both terms are strictly zero.

**Example E.** The example in Fig. 2E (Fig. 3 in SSR20 and also discussed in HPM19) is not graphically optimal. Here $\mathbf{O} = Z_1 Z_2$. Other valid adjustment sets are $Z_1$ or the empty set. From using $Z_1 \perp\!\!\!\perp Y | X$ and $X \perp\!\!\!\perp Z_2 | Z_1$ in the inequalities (7) one can derive in information-theoretic terms that both $Z_1 Z_2$ and $\emptyset$ are better than $\mathbf{vancs} = Z_1$, but since $J_{Z_1 Z_2} = J_\emptyset + I_{Z_2;Y|XZ_1} - I_{X;Z_1}$, a superior adjustment set depends on how strong the link $Z_1 \rightarrow X$ vs. $Z_2 \leftrightarrow Y$ is. The graph stays non-optimal also with a link $Z_1 \leftrightarrow Z_2$.

**Example F.** The example in Fig. 2F is also not graphically optimal. Here $\mathbf{O} = \emptyset$ and $Z_2$ is an N-node with a non-collider path to $X$. Other valid adjustment sets are $Z_1$ and $Z_1 Z_2$. Higher adjustment information here depends on the distribution. Also the same graph with the link $Z_1 \leftrightarrow X$ is non-optimal. If, however, there is another link $Z_1 \rightarrow Y$, then $\mathbf{O} = \emptyset$ is optimal (then $Z_1$ is a mediator).

**Example G.** The example in Fig. 2G is only a slight modification of Example E with an added selected condition **S**. Then $Z_1, Z_2 \in \mathbf{vancs}$. We still get $\mathbf{O} = Z_1 Z_2$ and this is now optimal since $Z_2$ is always open and any valid set has to contain $Z_1$.

The main result of this work is a set of necessary and sufficient conditions for the existence of graphical optimality and the proof of optimality of the **O**-set which is based on the intuition gained in the preceding examples.

**Theorem 3** (Necessary and sufficient graphical conditions for optimality and optimality of O-set)**.** *Given Assumptions 1 and with* $\mathbf{O} = \mathbf{PCP_C}$ *defined in Def. 4. Denote the set of N-nodes by* $\mathbf{N} = sp(Y\mathbf{MC}) \setminus (\mathbf{forbOS})$. *Finally, given an* $N \in \mathbf{N}$ *and a collider path* $N \leftrightarrow \cdots \leftrightarrow C \leftrightarrow \cdots \leftrightarrow W$ *(including* $N \leftrightarrow W$*) for* $C \in \mathbf{C}$ *and* $W \in Y\mathbf{M}$ *(indexed by i) with the collider path nodes denoted by* $\pi_i^N$ *(excluding* $N$ *and* $W$*), denote by* $\mathbf{O}_{\pi_i^N} = \mathbf{O}(X, Y, \mathbf{S}' = \mathbf{S}N\pi_i^N)$ *the O-set for the causal effect of* $X$ *on* $Y$ *given* $\mathbf{S}' = \mathbf{S} \cup \{N\} \cup \pi_i^N$. *If and only if exactly one valid adjustment set exists, or both of the following conditions are fulfilled, then graphical optimality holds and* **O** *is optimal:*

*(I) For all* $N \in \mathbf{N}$ *and all its collider paths i to* $W \in Y\mathbf{M}$ *that are inside* $\mathbf{C}$ *it holds that* $\mathbf{O}_{\pi_i^N}$ *does not block all non-causal paths from* $X$ *to* $Y$, *i.e.,* $\mathbf{O}_{\pi_i^N}$ *is non-valid,*

*and*

*(II) for all* $E \in \mathbf{O} \setminus \mathbf{P}$ *with an open path to* $X$ *given* $\mathbf{SO} \setminus \{E\}$ *there is a link* $E \leftrightarrow W$ *or an extended collider path* $E *\rightarrow C \leftrightarrow \cdots \leftrightarrow W$ *inside* $\mathbf{C}$ *for* $W \in Y\mathbf{M}$ *where all colliders* $C \in \mathbf{vancs}$.

Condition (I) and (II) essentially rule out the two canonical cases in Examples F and E, respectively, on which non-optimality in any graph is based. Applied to the examples, we obtain that in Example A Cond. (I) holds since no N-node exists and Cond. (II) holds since $X \perp\!\!\!\perp Z_1 | S$. In Example B also no N-node exists and Cond. (II) holds as $X \perp\!\!\!\perp E | \mathbf{SO} \setminus \{E\}$ for every $E \in \mathbf{O} \setminus \mathbf{P}$. In example C $Z_2$ is an N-node, but there is a collider path to $X$ through $Z_1$ which is in $\mathbf{vancs}$ such that Cond. I is fulfilled. Further, while $X \not\perp\!\!\!\perp Z_5 | \mathbf{SO} \setminus \{Z_5\}$, there is a link $Z_5 \leftrightarrow Y$ such that Cond. II holds. In example D $Z_1$ is an N-node, but it has a bidirected link with $X$ and Cond. (II) holds since $X \perp\!\!\!\perp Z_2 | \mathbf{SO} \setminus \{Z_2\}$. In Example E optimality does not hold, but Cond. (I) actually holds since there is no N-node. Cond. (II) is not fulfilled for $E = Z_1$, which has a path to $X$ given **O** and on the extended collider path $Z_1 \rightarrow Z_2 \leftrightarrow Y$ $Z_2 \notin \mathbf{vancs}$. For $\mathbf{Z}' = \emptyset$ and a distribution $\mathcal{P}'$ where the link $Z_2 \leftrightarrow Y$ almost vanishes we then have $J_\mathbf{O} < J_{\mathbf{Z}'}$. Example F has an N-node $Z_2$ and $\mathbf{O}_{\pi_i^N} = \mathbf{O}(X, Y, \mathbf{S}' = Z_2) = Z_1 Z_2$ is valid implying that Cond. (I) does not hold, while Cond. (II) is actually fulfilled with $\mathbf{O} = \emptyset$. For $\mathbf{Z}' = \mathbf{O}_{\pi_i^N} = Z_1 Z_2$ and a distribution $\mathcal{P}'$ where the link $X \rightarrow Z_1$ almost vanishes we then have $J_\mathbf{O} < J_{\mathbf{Z}'}$. Example G is optimal since there are no N-nodes and $Z_2 \in \mathbf{vancs}$.

Similar to SSR20, HPM19, and Witte et al. [2020], I also provide results regarding minimality and minimum cardinality for the hidden variables case in the Supplement.

## 3 Numerical experiments

We now investigate graphical optimality empirically to answer three questions: Firstly, whether for a linear estimator under Assumptions 2 the asymptotically optimal variance also translates into better

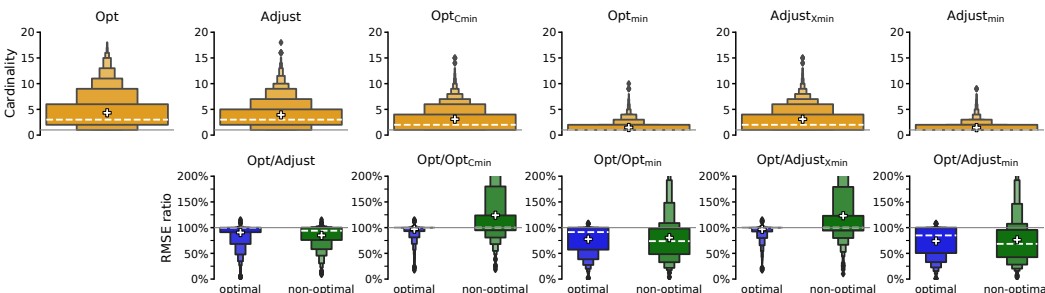

Figure 3: Results of linear experiments with LinReg and sample size $n = 100$. Shown are letter-value plots [Hofmann et al., 2017] of adjustment set cardinalities (top row), as well as RMSE ratios (bottom row) for the $\mathbf{O}$-set vs. other approaches for optimal configurations (left in blue) and non-optimal configurations (right in green). RMSE was estimated from 100 realizations. The dashed horizontal line denotes the median of the RMSE ratios, and the white 'plus' their average. The letter-value plots are interpreted as follows: The widest box shows the 25%–75% range. The next smaller box above (below) shows the 75%–87.5% (12.5%–25%) range and so forth.

*finite-sample* variance. Secondly, how the $\mathbf{O}$-set performs in non-optimal settings (according to Thm. 3). Thirdly, how the $\mathbf{O}$-set and variants thereof perform for estimators not captured by the class for which the theoretical results were derived (Assumptions 2). To this end, we compare the performance of $\mathbf{O}$, Adjust, $\mathbf{O}_{\mathrm{Cmin}}$, $\mathbf{O}_{\min}$, $\mathrm{Adjust}_{\mathrm{Xmin}}$, and $\mathrm{Adjust}_{\min}$ (see definitions in Section 2.3) together with linear least squares estimation (LinReg) on linear models. In the Supplement we also investigate nonlinear models using nearest neighbor regression (kNN), a multilayer perceptron (MLP), random forest regression, and double machine learning for partially linear regression models (DML) [Chernozhukov et al., 2018]. The experiments are based on a generalized additive model and described in detail in Section D. Among these 12,000 randomly created configurations 93% fulfill the optimality conditions in Thm. 3.

The results in Fig. 3 confirm our first hypothesis that for linear experiments with an estimator fulfilling Assumptions 2 and in settings where graphical optimality is fulfilled (Thm. 3) the $\mathbf{O}$-set either has similar RMSE or significantly outperforms all other tested variants. In particular, $\mathbf{O}_{\min}$ and $\mathrm{Adjust}_{\min}$ are bad choices for this setting. Adjust is intermediate and $\mathbf{O}_{\mathrm{Cmin}}$ and $\mathrm{Adjust}_{\mathrm{Xmin}}$ come closest to $\mathbf{O}$, but may still yield significantly higher variance.

Secondly, in non-optimal settings (only 7% of configurations) the $\mathbf{O}$-set still outperforms Adjust (as expected by Thm. 2). Compared to $\mathbf{O}_{\mathrm{Cmin}}$ and $\mathrm{Adjust}_{\mathrm{Xmin}}$ the $\mathbf{O}$-set leads to worse results for about half of the studied configurations, while $\mathbf{O}_{\min}$ and $\mathrm{Adjust}_{\min}$ are still bad choices. Cardinality is slightly higher for $\mathbf{O}$ compared to all other sets. In Fig. S7 we further differentiate the results by the cardinality of the $\mathbf{O}$-set and find that for small cardinalities (up to 4) the $\mathbf{O}$-set has the lowest variance in a majority of cases, but for higher cardinalities either $\mathbf{O}_{\mathrm{Cmin}}$ or again $\mathbf{O}$ have the lowest variance (slightly beating $\mathrm{Adjust}_{\mathrm{Xmin}}$). Hence, either $\mathbf{O}$ or $\mathbf{O}_{\mathrm{Cmin}}$ performs best in non-optimal configurations. For very small sample sizes $n = 30$ (see Fig. S2) that become comparable to the adjustment set cardinality, there tends to be a trade-off and smaller cardinality helps. Then $\mathbf{O}_{\mathrm{Cmin}}$ tends to be better than $\mathbf{O}$ for high cardinalities also in optimal settings, but here this effect is only present for $n = 30$ and for $n = 50$ already negligible compared to the gain in $J_{\mathbf{O}}$. In Appendix D.2 are RMSE ratios for all combinations of adjustment approaches considered here and it is shown that, in general, results are very similar for other sample sizes.

Thirdly, we investigate non-parametric estimators on linear as well as nonlinear models (implementations described in Section D, results in the figures of Section D.3) The different classes of estimators exhibit quite different behavior. For **kNN** (Figs. S8,S9) the $\mathbf{O}$-set has the lowest variance in around 50% of the configurations followed by $\mathbf{O}_{\mathrm{Cmin}}$ and $\mathbf{O}_{\min}$. More specifically (Figs. S15,S16), for small $\mathbf{O}$-set cardinalities up to 2 the $\mathbf{O}$-set and for higher either $\mathbf{O}_{\min}$ or $\mathbf{O}_{\mathrm{Cmin}}$ (the latter only in non-optimal configurations) perform best. For nonlinear experiments the results are less clear for $\mathbf{O}$-set cardinalities greater than 2, but $\mathbf{O}_{\min}$ is still a good choice. Regarding RMSE ratios, we see that, for the cases where $\mathbf{O}$ is not the best, the $\mathbf{O}$-set can have considerably higher variance, while $\mathbf{O}_{\min}$ seems to be most robust and may be a better choice if $\mathbf{O}$ is too large. **MLP** (Figs. S10,S11) behaves much differently. Here in optimal cases neither method outperforms any other for small

**O**-set cardinalities, but for higher cardinalities (Figs. S15,S16) the **O**-set is best in more than 50% of configurations (slightly less for nonlinear experiments) and the others share the rest (except Adjust$_{\text{min}}$). For non-optimal cases **O**, **O**$_{\text{Cmin}}$ and Adjust$_{\text{Xmin}}$ share the ranks. Regarding RMSE, for linear experiments the **O**-results are almost as optimal as for the LinReg estimator in the optimal setting. However, for non-optimal cases **O**$_{\text{Cmin}}$ can have considerably smaller variance and seems to be a robust option then, similarly to Adjust$_{\text{Xmin}}$. Also for nonlinear experiments **O**$_{\text{Cmin}}$ is more robust. The **RF** estimator (Figs. S12,S13) is again different. Here no method clearly is top-ranked, **O**$_{\text{min}}$ and Adjust$_{\text{min}}$ are slightly better for linear experiments and **O** for nonlinear experiments. **O**$_{\text{Cmin}}$ and **O**$_{\text{min}}$ are more robust regarding RMSE ratios (similar to Adjust$_{\text{Xmin}}$). Finally, the **DML** estimator (Fig. S14) was here applied only to linear experiments since its model assumption does not allow for fully nonlinear settings. For optimal settings here **O** is top-ranked in a majority of cases, but closely followed by **O**$_{\text{Cmin}}$ and Adjust$_{\text{Xmin}}$. In non-optimal cases for higher **O**-set cardinalities these two seem like a better choice. Quantitatively, **O**$_{\text{Cmin}}$ and Adjust$_{\text{Xmin}}$ are the most robust choices.

Overall, the **O**-set and its variants seem to outperform or match the Adjust-variants and whether higher cardinality of the **O**-set reduces performance depends strongly on the estimator and data.

## 4   Discussion and Conclusions

The proposed adjustment information formalizes the common intuition to choose adjustment sets that maximally constrain the effect variable and minimally constrain the cause variable. The main **theoretical contributions** are a necessary and sufficient graphical criterion for the existence of an optimal adjustment set in the hidden variables case and a definition and algorithm to construct it. To emphasize, graphical optimality implies that the **O**-set is optimal for *any distribution* consistent with the graph. Note that in cases where graphical optimality does not hold, there will still be distributions for which the **O**-set has maximal adjustment information.

Further, the optimal set is valid if and only if a valid adjustment set exists and has smaller (or equal) asymptotic variance compared to the Adjust-set proposed in Perković et al. [2018] for any graph, whether graphical optimality holds or not. This makes the **O**-set a natural choice in automated causal inference analyses. **Practical contributions** comprise Python code to construct adjustment sets and check optimality, as well as extensive numerical experiments that demonstrate that the theoretical results also hold for relatively small sample sizes.

The theoretical **optimality results are limited** to estimators for which the asymptotic variance becomes minimal for adjustment sets with maximal adjustment information (relation (8)). This is fulfilled for least-squares estimators, where even the direct relation (9) holds, but it is unclear whether this also holds for more general classes. The numerical results show that the **O**-set or minimized variants thereof often yield smaller variance also in non-optimal settings and beyond that estimator class. I speculate that further theoretical properties of maximizing adjustment information can be shown because relation (9) for $f(\cdot) = \frac{1}{\sqrt{n}} e^{H_{Y|X\mathbf{z s}} - H_{X|\mathbf{z s}}}$ seems related to the lower bound of the estimation variance counterpart to Fano's inequality (Theorem 8.6.6 in Cover and Thomas [2006]). For estimators sensitive to high-dimensionality one may consider data-driven criteria or penalties to step-wisely minimize the **O**-set. However, estimating, for example, the adjustment information from a potentially small sample size carries considerable errors itself. Another current limitation is that relation (9) only holds for univariate singleton cause variables $X$. The information-theoretical results, however, also hold for multivariate **X** and preliminary results indicate that, while relation (9) does not hold for multivariate **X**, the less restrictive relation (8) still seems to hold.

The proposed information-theoretic approach can guide **further research**, for example, to theoretically study relations (8),(9) for other estimators and to address other types of graphs as emerge from the output of causal discovery algorithms and the setting where the graph is unknown [Witte et al., 2020, Maathuis et al., 2009, 2010]. At present, the approach only applies to ADMGs and *Maximal Ancestral Graphs* (MAG) [Richardson and Spirtes, 2002] without selection variables. Last, it remains an open problem to identify optimal adjustment estimands for the hidden variables case based on other criteria such as the front-door formula and Pearl's general do-calculus [Pearl, 2009].

The results may carry considerable **practical impact** since, surprisingly, among the randomly created configurations more than 90% fulfill the optimality conditions indicating that also in many real-world scenarios graphical optimality may hold. Code is available in the python package `https://github.com/jakobrunge/tigramite`.

## Acknowledgments and Disclosure of Funding

I thank Andreas Gerhardus for very helpful comments. This work was funded by the ERC Starting Grant CausalEarth (grant no. 948112).

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
