**O**, Adjust, $\mathbf{O}_{\text{Cmin}}$, $\mathbf{O}_{\text{min}}$, $\text{Adjust}_{\text{Xmin}}$, and $\text{Adjust}_{\text{min}}$ (see definitions in Section 2.3) together with linear least squares estimation (LinReg) on linear models. In the Supplement we also investigate nonlinear models using nearest neighbor regression (kNN), a multilayer perceptron (MLP), random forest regression, and double machine learning for partially linear regression models (DML) [Chernozhukov et al., 2018]. The experiments are based on a generalized additive model and described in detail in Section D. Among these 12,000 randomly created configurations 93% fulfill the optimality conditions in Thm. 3.

The results in Fig. 3 confirm our first hypothesis that for linear experiments with an estimator fulfilling Assumptions 2 and in settings where graphical optimality is fulfilled (Thm. 3) the **O**-set either has similar RMSE or significantly outperforms all other tested variants. In particular, $\mathbf{O}_{\text{min}}$ and $\text{Adjust}_{\text{min}}$ are bad choices for this setting. Adjust is intermediate and $\mathbf{O}_{\text{Cmin}}$ and $\text{Adjust}_{\text{Xmin}}$ come closest to **O**, but may still yield significantly higher variance.

Secondly, in non-optimal settings (only 7% of configurations) the **O**-set still outperforms Adjust (as expected by Thm. 2). Compared to $\mathbf{O}_{\text{Cmin}}$ and $\text{Adjust}_{\text{Xmin}}$ the **O**-set leads to worse results for about half of the studied configurations, while $\mathbf{O}_{\text{min}}$ and $\text{Adjust}_{\text{min}}$ are still bad choices. Cardinality is slightly higher for **O** compared to all other sets. In Fig. S7 we further differentiate the results by the cardinality of the **O**-set and find that for small cardinalities (up to 4) the **O**-set has the lowest variance in a majority of cases, but for higher cardinalities either $\mathbf{O}_{\text{Cmin}}$ or again **O** have the lowest variance (slightly beating $\text{Adjust}_{\text{Xmin}}$). Hence, either **O** or $\mathbf{O}_{\text{

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

$). We use "$\ast$" to denote either edge mark. There can be no loops or directed cycles. See Fig. 1A for an example. The results also hold for *Maximal Ancestral Graphs* (MAG) [Richardson and Spirtes, 2002] without selection variables. A path between two nodes $X$ and $Y$ is a sequence of edges such that every edge occurs only once. A path between $X$ and $Y$ is called *directed or causal* from $X$ to $Y$ if all edges are directed towards $Y$, else it is called *non-causal*. A node $C$ on a path is called a *collider* if "$\ast\!\rightarrow\!C\!\leftarrow\!\ast$". Kinships are defined as usual: parents $pa(X, \mathcal{G})$ for "$\bullet\!\rightarrow\!X$", spouses $sp(X, \mathcal{G})$ for "$X\!\leftrightarrow\!\bullet$", children $ch(X, \mathcal{G})$ for "$X\!\rightarrow\!\bullet$", and correspondingly descendants $des$ and ancestors $an$. We omit the $\mathcal{G}$ in the following since all relations are relative to the graph $\mathcal{G}$ in this paper. Our approach does not involve modified graph constructions as in van der Zander et al. [2019] and other works. A node is an ancestor and descendant of itself, but not a parent/child/spouse of itself. The mediator nodes on causal paths from $X$ to $Y$ are denoted $\mathbf{M} = \mathbf{M}(X, Y)$ and exclude $X$ and $Y$ (different from definitions in other works). For sets of variables the kinship relations correspond to the union of the individual variables. For parent/child/spouse-relationships these exclude the set of variables itself. A path $\pi$ between $X$ and $Y$ in $\mathcal{G}$ is blocked (or closed) by a node set $\mathbf{Z}$ if (i) $\pi$ contains a non-collider in $\mathbf{Z}$ or (ii) $\pi$ contains a collider that is not in $an(\mathbf{Z})$. Otherwise the path $\pi$ is open (or active/connected) given $\mathbf{Z}$. Nodes $X$ and $Y$ are said to be m-separated given $\mathbf{Z}$ if every path between them is blocked by $\mathbf{Z}$, denoted as $X \perp\!\!\!\perp Y | \mathbf{Z}$. In the following we will simplify set notation and denote unions of variables as $\{W\} \cup \mathbf{M} \cup \mathbf{A} = W\mathbf{M}\mathbf{A}$.

## B  Further theoretical results and proofs

### B.1  Properties of adjustment information

$J_{\mathbf{Z}}$ is not necessarily positive if the dependence between $X$ and $\mathbf{Z}$ (given $\mathbf{S}$) is larger than that between $\mathbf{Z}$ and $Y$ given $X\mathbf{S}$. By the properties of CMI, it is bounded by

$$-\min(H_{X|\mathbf{S}}, H_{\mathbf{Z}|\mathbf{S}}) \leq J_{XY|\mathbf{S}.\mathbf{Z}} \leq \min(H_{Y|X\mathbf{S}}, H_{\mathbf{Z}|X\mathbf{S}}). \tag{S1}$$

## B.2 Causally sufficient case

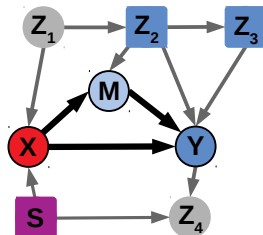

Figure S1: DAG version of graph in Fig. 1A with **O**-set shown as blue boxes.

The optimal adjustment set for the causally sufficient case was derived in HPM19 and Rotnitzky and Smucler [2019]. Here the derivation is discussed from an information-theoretic perspective.

**Definition B.1** (O-set in the causally sufficient case). *Given Assumptions 1 restricted to DAGs with no hidden variables, define the set*

$$\mathbf{O} = \mathbf{P} = pa(Y\mathbf{M}) \setminus \mathbf{forb}.$$

In the causally sufficient case a valid adjustment set always exists and the **O**-set is always valid since **O** contains no descendants of $Y\mathbf{M}$ and all non-causal paths from $X$ to $Y$ are blocked since **P** blocks all paths from $X$ through parents of $Y\mathbf{M}$.

Figure S1 shows an example DAG with a mediator $M$ and conditioned variable $S$. The **O**-set $\mathbf{O} = Z_2 Z_3$ is depicted by blue boxes. Compare **O** with $\mathbf{vancs} = Z_1 Z_2 Z_3 \mathbf{S}$ (Adjust-set in Perković et al. [2018]) in the inequalities (7). Since $Z_1 \perp\!\!\!\perp Y \mid \mathbf{O} X S$, term (iii) is zero and since $\mathbf{O} \setminus \mathbf{vancs} = \emptyset$, also term (iv) is zero. Further, terms (i) and (ii) are both strictly greater than zero (under Faithfulness). Then $J_{\mathbf{O}} > J_{\mathbf{vancs}}$ and under Assumptions 2 by Lemma 2 the **O**-set has a smaller asymptotic variance than **vancs**. Since the parents of $Y\mathbf{M}$ block all paths from any other valid adjustment sets to $Y$ and because any valid adjustment set $\mathbf{Z}$ has to block paths from $X$ to $pa(Y\mathbf{M}) \setminus \mathbf{Z}$, $J_{\mathbf{O}} \geq J_{\mathbf{Z}}$ holds in general for any valid set $\mathbf{Z}$ as proven from an information-theoretic perspective in Proposition B.1.

**Proposition B.1** (Optimality of O-set in causally sufficient case). *Given Assumptions 1 restricted to DAGs with no hidden variables and with $\mathbf{O} = \mathbf{P}$ defined in Def. B.1, graphical optimality holds for any graph and $\mathbf{O}$ is optimal.*

Similar to HPM19 and Witte et al. [2020], there also exist results regarding minimality and minimum cardinality which are covered for the hidden variables case in Corollary B.1.

## B.3 Hidden variables case

Here we provide some further theoretical results for the general hidden variables case in addition to the lemmas and theorems in the main text.

**Corollary B.1** (Minimality and minimum cardinality). *Given Assumptions 1, assume that graphical optimality holds, and, hence, $\mathbf{O}$ is optimal. Further it holds that:*

1. *If $\mathbf{O}$ is not minimal, then $J_{\mathbf{O}} > J_{\mathbf{Z}}$ for all minimal valid $\mathbf{Z} \neq \mathbf{O}$,*

2. *If $\mathbf{O}$ is minimal valid, then $\mathbf{O}$ is the unique set that maximizes the adjustment information $J_{\mathbf{Z}}$ among all minimal valid $\mathbf{Z} \neq \mathbf{O}$,*

3. *$\mathbf{O}$ is of minimum cardinality, that is, there is no subset of $\mathbf{O}$ that is still valid and optimal.*

Another relevant Proposition states that $\mathbf{O}_{\mathrm{Cmin}}$ is a subset of **vancs**, similar to corresponding Lemmas in van der Zander et al. [2019].

**Proposition B.2** (Collider-minimized O-set is a subset of Adjust.). *Given Assumptions 1 with $\mathbf{O} = \mathbf{PCP_C}$ defined in Def. 4 and the $\mathbf{O}_{\mathrm{Cmin}}$-set constructed with Alg. C.2 it holds that $\mathbf{O}_{\mathrm{Cmin}} \subseteq \mathbf{vancs}$.*

## B.4 Proof of Lemma 1

**Lemma** (Necessary and sufficient comparison criterion for existence of an optimal set). Given Assumptions 1, if and only if there is a $\mathbf{Z} \in \mathcal{Z}$ such that either there is no other $\mathbf{Z}' \neq \mathbf{Z} \in \mathcal{Z}$ or for all other $\mathbf{Z}' \neq \mathbf{Z} \in \mathcal{Z}$ and all distributions $\mathcal{P}$ consistent with $\mathcal{G}$ it holds that

$$\underbrace{I_{\mathbf{Z}\setminus\mathbf{Z}';Y|\mathbf{Z}'X\mathbf{S}}}_{(i)} \geq \underbrace{I_{\mathbf{Z}'\setminus\mathbf{Z};Y|\mathbf{Z}X\mathbf{S}}}_{(iii)}, \quad \text{and}$$

$$\underbrace{I_{X;\mathbf{Z}'\setminus\mathbf{Z}|\mathbf{Z}\mathbf{S}}}_{(ii)} \geq \underbrace{I_{X;\mathbf{Z}\setminus\mathbf{Z}'|\mathbf{Z}'\mathbf{S}}}_{(iv)}, \tag{S2}$$

then graphical optimality holds and $\mathbf{Z}$ is optimal implying $J_{\mathbf{Z}} \geq J_{\mathbf{Z}'}$.

*Proof.* If there is no other $\mathbf{Z}'$, the statement trivially holds. Assuming there is another $\mathbf{Z}'$, we prove the two implications as follows by an information-theoretic decomposition.

Define disjunct (possibly empty) sets $\mathbf{R}, \mathbf{B}, \mathbf{A}$ with $\mathbf{Z} = \mathbf{AB}$ and $\mathbf{Z}' = \mathbf{BR}$ with $\mathbf{B} = \mathbf{Z} \cap \mathbf{Z}'$. Note that if both $\mathbf{R} = \emptyset$ and $\mathbf{A} = \emptyset$, then $\mathbf{Z} = \mathbf{Z}'$. Consider two different ways of applying the chain rule of CMI,

$$I_{\mathbf{ABR};Y|X\mathbf{S}} - I_{X;\mathbf{ABR}|\mathbf{S}}$$
$$= I_{\mathbf{AB};Y|X\mathbf{S}} + I_{\mathbf{R};Y|\mathbf{AB}X\mathbf{S}} - I_{X;\mathbf{AB}|\mathbf{S}} - I_{X;\mathbf{R}|\mathbf{ABS}} \tag{S3}$$
$$= I_{\mathbf{BR};Y|X\mathbf{S}} + I_{\mathbf{A};Y|\mathbf{BR}X\mathbf{S}} - I_{X;\mathbf{BR}|\mathbf{S}} - I_{X;\mathbf{A}|\mathbf{BRS}}, \tag{S4}$$

from which with $J_{\mathbf{Z}} = I_{\mathbf{AB};Y|X\mathbf{S}} - I_{X;\mathbf{AB}|\mathbf{S}}$ and $J_{\mathbf{Z}'} = I_{\mathbf{RB};Y|X\mathbf{S}} - I_{X;\mathbf{RB}|\mathbf{S}}$ it follows that

$$J_{\mathbf{Z}} = J_{\mathbf{Z}'}$$
$$+ \underbrace{I_{\mathbf{A};Y|\mathbf{BR}X\mathbf{S}}}_{(i)} + \underbrace{I_{X;\mathbf{R}|\mathbf{ABS}}}_{(ii)} - \underbrace{I_{\mathbf{R};Y|\mathbf{AB}X\mathbf{S}}}_{(iii)} - \underbrace{I_{X;\mathbf{A}|\mathbf{BRS}}}_{(iv)}. \tag{S5}$$

The inequalities (S2) then read

$$\underbrace{I_{\mathbf{A};Y|\mathbf{BR}X\mathbf{S}}}_{(i)} \geq \underbrace{I_{\mathbf{R};Y|\mathbf{AB}X\mathbf{S}}}_{(iii)}, \quad \text{and}$$

$$\underbrace{I_{X;\mathbf{R}|\mathbf{ABS}}}_{(ii)} \geq \underbrace{I_{X;\mathbf{A}|\mathbf{BRS}}}_{(iv)}. \tag{S6}$$

"if": If term (i) is greater or equal to term (iii) and term (ii) greater or equal to term (iv), then trivially $J_{\mathbf{Z}} \geq J_{\mathbf{Z}'}$ for all distributions $\mathcal{P}$.

"only if": We prove the contraposition that if for all valid $\mathbf{Z}$ there exists a valid $\mathbf{Z}' \neq \mathbf{Z}$ and a distributions $\mathcal{P}$ consistent with $\mathcal{G}$ such that

$$\underbrace{I_{\mathbf{A};Y|\mathbf{BR}X\mathbf{S}}}_{(i)} < \underbrace{I_{\mathbf{R};Y|\mathbf{AB}X\mathbf{S}}}_{(iii)}, \quad \text{or} \quad \underbrace{I_{X;\mathbf{R}|\mathbf{ABS}}}_{(ii)} < \underbrace{I_{X;\mathbf{A}|\mathbf{BRS}}}_{(iv)}, \tag{S7}$$

then there always exists a modification $\mathcal{P}'$ of the distribution $\mathcal{P}$ such that $J_{\mathbf{Z}} < J_{\mathbf{Z}'}$. This is because, in both cases, we can always construct a distribution for which terms (ii) and (i), respectively, become arbitrary close to zero. Consider the two cases as follows:

1) there exists a distribution $\mathcal{P}$ with $I_{\mathbf{A};Y|\mathbf{BR}X\mathbf{S}} < I_{\mathbf{R};Y|\mathbf{AB}X\mathbf{S}}$: Since CMIs are always non-negative, it holds that $\mathbf{R} \neq \emptyset$ and there must exist at least one open path between $\mathbf{R}$ and $Y$ where every collider is in $\mathbf{AB}X\mathbf{S}$ and no non-collider is in $\mathbf{AB}X\mathbf{S}$. No such open path can pass through $X$ because if $X$ is a non-collider (as for paths continuing on causal paths from $X$ to $Y$), then the path is blocked, and if $X$ is a collider, then there would be a non-causal path from $X$ to $Y$ given $\mathbf{ZS}$ which would make $\mathbf{Z}$ invalid while $\mathbf{Z}$ is assumed valid. Correspondingly, no open path from $\mathbf{A}$ (if $\mathbf{A} \neq \emptyset$) to $Y$ given $\mathbf{BR}X\mathbf{S}$, if a path exists at all, can pass through $X$ if $\mathbf{Z}'$ is assumed valid. Now we can construct a distribution $\mathcal{P}'$ with associated structural causal model (SCM) consistent with $\mathcal{G}$ where $I_{\mathbf{A};Y|\mathbf{BR}X\mathbf{S}} < I_{\mathbf{R};Y|\mathbf{AB}X\mathbf{S}}$ holds as in $\mathcal{P}$ and still all links "$U\ast\!\!-\!\!\ast X$" for $X \in X$ and $U \notin X\mathbf{M}Y$

*almost vanish*. Consider the three possible links and associated assignment functions in the SCM: (1) "$X \rightarrow U$" with $U := f_U(\ldots, X, \ldots)$, (2) "$X \leftarrow U$" with $X := f_X(\ldots, U, \ldots)$, and (3) "$X \leftrightarrow U$" with $X := f_X(\ldots, L^U, \ldots)$ where $L^U$ denotes one or more latent variables. In each case, to go from $\mathcal{P}$ to $\mathcal{P}'$, we can modify $f. \rightarrow f'.$ where in $f'.$ the dependence on the respective argument is replaced by $X \rightarrow cX$, $U \rightarrow cU$, or $L^U \rightarrow cL^U$ for $c \in \mathbb{R}$, and where we consider the limit $c \rightarrow 0$. This modification does not affect $I_{\mathbf{A};Y|\mathbf{BR}X\mathbf{S}} < I_{\mathbf{R};Y|\mathbf{AB}X\mathbf{S}}$ because the paths contributing to the two CMIs cannot pass through $X$. On the other hand, then term (ii) $I_{X;\mathbf{R}|\mathbf{ABS}} \rightarrow 0$ because all paths passing through $X$ contain almost zero links and there cannot be a path from $\mathbf{R}$ to $X$ through $\mathbf{M}Y$ for a valid $\mathbf{Z}$. Hence, since in Eq. (S5) term (i) is smaller than term (iii) by assumption, and term (ii) is almost zero, it holds that $J_{\mathbf{Z}} < J_{\mathbf{Z}'}$.

2) there exists a distribution $\mathcal{P}$ with $I_{X;\mathbf{R}|\mathbf{ABS}} < I_{X;\mathbf{A}|\mathbf{BRS}}$: As before, since CMIs are always non-negative, it holds that $\mathbf{A} \neq \emptyset$ and there must exist at least one open path between $\mathbf{A}$ and $X$ where every collider is in $\mathbf{BRS}$ and no non-collider is in $\mathbf{BRS}$. No such open path can pass through $Y\mathbf{M}$ because if any node in $Y\mathbf{M}$ is a collider, then the path is blocked, and no path can contain any node in $Y\mathbf{M}$ as a non-collider since then either the graph is cyclic or $\mathbf{Z}'$ contains descendants of $Y\mathbf{M}$ leading to $\mathbf{Z}' \cap \mathbf{forb} \neq \emptyset$ while $\mathbf{Z}'$ is assumed valid. Correspondingly, no open path from $\mathbf{R}$ (if $\mathbf{R} \neq \emptyset$) to $X$ given $\mathbf{ABS}$, if a path exists at all, can pass through $Y\mathbf{M}$ if $\mathbf{Z}$ is assumed valid. Then, analogous to before, we can construct a $\mathcal{P}'$ with associated SCM consistent with $\mathcal{G}$ where $I_{X;\mathbf{R}|\mathbf{ABS}} < I_{X;\mathbf{A}|\mathbf{BRS}}$ holds and where all links "$U \ast\!\!-\!\!\ast W$" for $W \in Y\mathbf{M}$ and $U \notin X\mathbf{M}Y$ *almost vanish*. Then term (i) $I_{\mathbf{A};Y|\mathbf{BR}X\mathbf{S}} \rightarrow 0$ because all paths contain almost zero links and there cannot be a path from $\mathbf{A}$ to $Y$ where $X$ contains a collider for a valid $\mathbf{Z}'$ since this would constitute a non-causal path. Hence, since in Eq. (S5) term (ii) is smaller than term (iv) by assumption, and term (i) is almost zero, it holds that $J_{\mathbf{Z}} < J_{\mathbf{Z}'}$. $\qquad\square$

## B.5  Proof of Proposition B.1

**Proposition** (Optimality of O-set in causally sufficient case). Given Assumptions 1 restricted to DAGs with no hidden variables and with $\mathbf{O} = \mathbf{P}$ defined in Def. B.1, graphical optimality holds for any graph and $\mathbf{O}$ is optimal.

*Proof.* The proof is based on Lemma 1 and relation (S5). We will prove that for any DAG $\mathcal{G}$ term (i)$\geq$(iii) and term (ii)$\geq$(iv) from which optimality follows by Lemma 1.

We have to show that $I_{\mathbf{A};Y|\mathbf{BR}X\mathbf{S}} \geq I_{\mathbf{R};Y|\mathbf{AB}X\mathbf{S}}$ and $I_{X;\mathbf{R}|\mathbf{ABS}} \geq I_{X;\mathbf{A}|\mathbf{BRS}}$ where $\mathbf{O} = \mathbf{AB}$ and $\mathbf{Z}' = \mathbf{RB}$ with $\mathbf{B} = \mathbf{O} \cap \mathbf{Z}'$.

Any path from $X$ or $\mathbf{V} \setminus Y\mathbf{MOS}X$ to $Y\mathbf{M}$ given $\mathbf{OS}$ (denoted by $\boxed{\cdot}$), excluding the causal path from $X$ to $Y$, features at least one of the following motifs: "$X, V \ast\!\!-\!\!\ast \boxed{P} \rightarrow W$" (excluding "$X \rightarrow \boxed{P} \rightarrow W$"), or "$V \leftarrow W$" where, hence, $V \in \mathbf{forb}$.

Now all paths from a valid adjustment set $\mathbf{Z}'$ with $\mathbf{Z}' \in \mathcal{Z}$ to $Y$ are blocked given $\mathbf{OS}$: Motif "$X, V \ast\!\!-\!\!\ast \boxed{P} \rightarrow W$" contains a non-collider in $\mathbf{OS}$ and is, hence, blocked. In motif "$V \leftarrow W$" $V \in \mathbf{forb}$. Since $X \notin des(Y)$ (acyclicity) and $\mathbf{Z}' \cap des(Y) = \emptyset$ (validity of $\mathbf{Z}'$), the paths from $\mathbf{Z}'$ to $V$ either end with a head at $V$ or there must be a collider $K$ that is a descendant of $V$ and hence, $K \in \mathbf{forb}$. Then $K \notin an(\mathbf{OS})$ and $K \notin \mathbf{Z}'$ and the path is therefore blocked. Hence, with $\mathbf{R} \subseteq \mathbf{Z}'$, term (iii) is zero by Markovity.

Term (iv) $I_{X;\mathbf{A}|\mathbf{Z}'\mathbf{S}} = 0$ for any valid $\mathbf{Z}'$ because $\mathbf{A} \subseteq pa(Y\mathbf{M})$ and then otherwise there would be a non-causal path from $X$ through $\mathbf{A}$ to $Y\mathbf{M}$. $\qquad\square$

## B.6  Further Lemmas

**Lemma B.1** (Relevant path motifs wrt. the O-set). *Given Assumptions 1 but* without *a priori assuming that a valid adjustment set exists (apart from the requirement $\mathbf{S} \cap \mathbf{forb} = \emptyset$). With $\mathbf{O} = \mathbf{PCP_C}$ defined in Def. 4 any path from $X$ or $\mathbf{V} \setminus Y\mathbf{MOS}X$ to $Y\mathbf{M}$ given $\mathbf{OS}$ (denoted by $\boxed{\cdot}$), excluding the causal path from $X$ to $Y$, features at least one of the following motifs with certain constraints as indicated. We denote $V \in \mathbf{V} \setminus Y\mathbf{MOS}X$ and further differentiate nodes in $Y\mathbf{M}$ as $W \in Y\mathbf{M}$ and in $\mathbf{O} = \mathbf{PCP_C}$ as $C \in \mathbf{C}$ or $P \in \mathbf{P}$ or $P_C \in \mathbf{P_C}$. Last, we denote those collider path nodes not included in the $\mathbf{O}$-set in Alg. C.1 due to not sufficing Def. 3(1) as $F$ with $F \in \mathbf{forb}$ and those not sufficing Def. 3(2a,b) as $N$ with $N \notin \mathbf{forb}$, $N \notin \mathbf{vancs}$, and $N \not\perp\!\!\!\perp X \mid \mathbf{vancs}$:*

*(1a)* "$*\!\!-\!\!*X\rightarrow\boxed{C}\leftrightarrow$"

*(1b)* "$*\!\!-\!\!*X\rightarrow\boxed{P_C}\rightarrow\boxed{C}\leftrightarrow$"

*(2a)* "$X, V *\!\!-\!\!* \boxed{P}\rightarrow W$" *excluding* "$X\rightarrow\boxed{P}\rightarrow W$"

*(2b)* "$X, V *\!\!-\!\!* \boxed{P_C}\rightarrow\boxed{C}\leftrightarrow$" *excluding (1b)*

*(3a)* "$V\leftarrow W$" *where, hence, $V \in$* **forb**

*(3b)* "$X, V\leftarrow\boxed{C}\leftrightarrow$"

*(4a)* "$*\!\!-\!\!*F\leftrightarrow W$" *with the constraint $F \notin$* **vancs**

*(4b)* "$*\!\!-\!\!*F\leftrightarrow\boxed{C}\leftrightarrow$" *with the constraints $F \notin pa(C)$ and $F \notin$* **vancs**

*(5a)* "$*\!\!-\!\!*N\leftrightarrow W$" *with the constraints $N \notin pa(W)$ and $W \notin pa(N)$*

*(5b)* "$*\!\!-\!\!*N\leftrightarrow\boxed{C}\leftrightarrow$" *with the constraint $N \notin pa(C)$*

*Further it holds that $F, N, X \notin$* **S**.

*Proof.* Any path from $X$ or $\mathbf{V} \setminus Y\mathbf{MOS}X$ to $Y\mathbf{M}$ has to contain a link "$A*\!\!-\!\!*B$" where $A = X$ or $A \in \mathbf{V} \setminus Y\mathbf{MOS}X$ and $B \in Y\mathbf{MO}$ where $*\!\!-\!\!* \in \{\rightarrow, \leftarrow, \leftrightarrow\}$. If we differentiate the left node by $X$ or $V \in \mathbf{V} \setminus Y\mathbf{MOS}X$ and the right node by $W \in Y\mathbf{M}$ or $C \in \mathbf{C}$ or $P \in \mathbf{P}$ or $P_C \in \mathbf{P_C}$, we can in principle have $2 \cdot 4 \cdot 3 = 24$ link types which are motifs if we consider the adjacent links to $A$ and $B$. These are listed in the Lemma except for "$*\!\!-\!\!*X\rightarrow W$" which is part of the causal path from $X$ to $Y$, "$X\rightarrow\boxed{P}\rightarrow W$" which cannot occur since then $P \in \mathbf{M}$, "$V\rightarrow W$" which cannot occur since $\mathbf{P}$ would contain $V$ or $V \in des(Y\mathbf{M})$ leading to a cyclic graph, "$V\rightarrow C$" which cannot occur since $\mathbf{P_C}$ would contain $V$, and "$X\leftarrow W$" which cannot occur since this implies a cyclic graph.

Regarding the constraints listed in motifs (4a,b) for $F \in$ **forb** it holds that $F \notin$ **vancs** because **vancs** $= an(XY\mathbf{S}) \setminus$ **forb** by definition. Further, in (4b) $F \notin pa(C)$ holds because otherwise $C \in$ **forb**. In motif (5a) $N \notin pa(W)$ holds because $N \notin$ **vancs** and $W \notin pa(N)$ holds because $N \notin$ **forb**. In motif (5b) $N \notin pa(C)$ holds because $C \in$ **vancs** contradicts $N \notin$ **vancs** and $N \not\perp\!\!\!\perp X \mid$ **vancs** with $N\rightarrow C$ contradicts $C \perp\!\!\!\perp X \mid$ **vancs**. Last, it holds that $F, N, X \notin$ **S** because $\mathbf{S} \cap$ **forb** $= \emptyset$, $\mathbf{S} \cap X = \emptyset$ by Assumptions 1 and $N \notin$ **vancs** while $\mathbf{S} \subseteq$ **vancs**. $\qquad\square$

**Lemma B.2** (Sufficient condition for non-identifiability). *Given Assumptions 1 but* without *a priori assuming that a valid adjustment set exists (apart from the requirement $\mathbf{S} \cap$* **forb** *$= \emptyset$). With $\mathbf{O} = \mathbf{PCP_C}$ defined in Def. 4, if on any non-causal path from $X$ to $Y$ given $\mathbf{OS}$ any of the motifs (1a) or (4a) or (4b) for $F = X$ occurs as listed in Lemma B.1, then the causal effect of $X$ on $Y$ (potentially through $\mathbf{M}$) is* not *identifiable by backdoor adjustment.*

*Proof.* If motif (4a) "$X\leftrightarrow W$" for $W \in Y\mathbf{M}$ occurs, the case is trivial [Pearl, 2009, Thm. 4.3.1]. In motifs (1a) "$X\rightarrow\boxed{C}\leftrightarrow$" and (4b) "$X\leftrightarrow\boxed{C}\leftrightarrow$" we have that since Def. 3(2b) $C \perp\!\!\!\perp X \mid$ **vancs** is not fulfilled, Def. 3(2a) $C \in$ **vancs** must be the case. Then every $C_k$ on collider paths to $W$ also fulfills $C_k \in$ **vancs** because for all of them $C_k \perp\!\!\!\perp X \mid$ **vancs** does not hold since each collider is opened. Hence, there exists a collider path $X*\!\!-\!\!\rightarrow C\leftrightarrow\cdots\leftrightarrow W$ where every collider $C \in$ **vancs** $= an(XY\mathbf{S}) \setminus$ **forb**. This path cannot be blocked by any adjustment set (given $\mathbf{S}$): colliders with $C \in an(\mathbf{S})$ are always open. For colliders with $C \in an(X)$ or $C \in an(Y)$ there is a directed path to $X$ or $Y$ and either this path is open leading to a non-causal path, or an adjustment set contains a non-collider on that directed path which opens the collider $C$. $\qquad\square$

In Theorem 1 we will prove that the condition in Lemma B.2 is also necessary for non-identifiability by backdoor adjustment. To this end, consider the following Lemmas.

**Lemma B.3** (Collider parents fulfill Def. 3). *Given Assumptions 1. With $\mathbf{O} = \mathbf{PCP_C}$ defined in Def. 4, for every $P \in \mathbf{P_C}$ conditions (1), and (2a) or (2b) in Def. 3 hold.*

*Proof.* Denote a pair $P_C\rightarrow C$ for $C \in \mathbf{C}$ fulfilling conditions (1), and (2a) or (2b) in Def. 3. Firstly, (1) $P_C \notin$ **forb** since if $P_C \in des(Y\mathbf{M})$ also $C \in des(Y\mathbf{M})$ and if $P_C = X$, then by Lemma B.2 no valid adjustment set exists, contrary to Assumptions 1. Secondly, it cannot be that (2a) $P_C \notin$ **vancs** and (2b) $P_C\not\perp\!\!\!\perp X \mid$ **vancs** because then the path from $X$ to $P_C$ would extend to $C$ and would not be

blocked because $P_C \notin$ **vancs**. But then also $C \notin$ **vancs** and $C$ would not fulfill the conditions in Def. 3. $\square$

**Lemma B.4** (Blockedness of parent-child-motifs). *Given Assumptions 1 with $\mathbf{O} = \mathbf{PCP_C}$ defined in Def. 4. Any path from $X$ or a valid adjustment set $\mathbf{Z}$ with $\mathbf{Z} \in \mathcal{Z}$ to $Y$ containing the motifs (1b), (2a), (2b), (3a), (3b) is blocked given $\mathbf{OS}$.*

*Proof.* Motifs (1b), (2a), (2b), and (3b) contain a non-collider in $\mathbf{OS}$ and are, hence, all blocked. In motif (3a) $V \in$ **forb**. Since $X \notin des(Y)$ (acyclicity) and $\mathbf{Z} \cap des(Y) = \emptyset$ (validity of $\mathbf{Z}$), the paths from $\mathbf{Z}$ to $V$ either end with a head at $V$ or there must be a collider $K$ that is a descendant of $V$ and hence, $K \in$ **forb**. Then $K \notin an(\mathbf{OS})$ and $K \notin \mathbf{Z}$ and the path is therefore blocked. $\square$

**Lemma B.5** (Blockedness of F-motifs). *Given Assumptions 1 with $\mathbf{O} = \mathbf{PCP_C}$ defined in Def. 4. Firstly, any path from $X$ to $Y$ containing the motifs (4a) or (4b) for $F \in des(Y\mathbf{M})$ is blocked given $\mathbf{OS}$. Secondly, any path from a valid adjustment set $\mathbf{Z}$ with $\mathbf{Z} \in \mathcal{Z}$ to $Y$ containing the motifs (4a) or (4b) for $F \in des(Y\mathbf{M})$ is blocked given $X\mathbf{OS}$.*

*Proof.* First statement: $F \notin$ **vancs** by Lemma B.1 and, hence, in particular $F \notin an(X)$. Then, if a path exists, either the paths from $X$ to $F$ end with a head at $F$ or there must be at least one collider $K$ with $F \in an(K)$ on a path to $X$. Now $F, K \notin an(\mathbf{OS})$ because $\mathbf{OS} \cap$ **forb** $= \emptyset$ and the path is blocked. Secondly, $F \notin an(\mathbf{Z})$ since $\mathbf{Z}$ is valid. Then similarly, if a path exists, either the paths from $\mathbf{Z}$ to $F$ end with a head at $F$ or there must be at least one collider $K$ on a path to $\mathbf{Z}$ with $F \in an(K)$. Now $F, K \notin an(X\mathbf{OS})$ because $\mathbf{OS} \cap$ **forb** $= \emptyset$ and $F \notin$ **vancs** by Lemma B.1 and the path is blocked. $\square$

**Lemma B.6** (Blockedness of N-motifs). *Given Assumptions 1 with $\mathbf{O} = \mathbf{PCP_C}$ defined in Def. 4. Firstly, any path from $X$ to $Y$ containing the motifs (5a) or (5b) is blocked given $\mathbf{OS}$. Secondly, any path from a valid adjustment set $\mathbf{Z}$ to $Y$ containing the motifs (5a) or (5b) is blocked given $X\mathbf{OS}$ if $\mathbf{Z}$ does not contain any descendants of $N$ ($\mathbf{Z} \cap des(N) = \emptyset$).*

*Proof.* First statement: $N \notin$ **vancs** by definition of $N$ and, hence, in particular $N \notin an(X)$. Then, if a path exists, either the paths from $X$ to $N$ end with a head at $N$ or there must be at least one collider $K$ with $N \in an(K)$ and $K \notin$ **vancs** on a path to $X$. Now $N, K \notin an(\mathbf{OS})$ can be seen by considering the different parts of $\mathbf{O}$: $N, K \notin an(\mathbf{PS})$ since $N, K \notin$ **vancs** and $N, K \notin an(C)$ for $C \in$ **vancs** $\cap \mathbf{CP_C}$. Finally, $N, K \notin an(C)$ for for $C \in \mathbf{CP_C}$ with $C \perp\!\!\!\perp X \mid$ **vancs** because $N, K \not\perp\!\!\!\perp X \mid$ **vancs**. Hence, the path is blocked. Second statement: If $\mathbf{Z}$ does not contain any descendants of $N$, then $N \notin an(\mathbf{Z})$. Then any path from a $\mathbf{Z}$ is blocked by the same reasoning as in the first part with the addition that $N \notin an(X)$ and hence the motif is blocked given $X\mathbf{OS}$. $\square$

The following Lemma is not needed in this paper, but may be of interest for further research.

**Lemma B.7** (Existence of X-N-path). *Given Assumptions 1 with $\mathbf{O} = \mathbf{PCP_C}$ defined in Def. 4. There must be at least one path from $X$ to $N$ (defined in the motifs (5a) or (5b)) that ends with a head at $N$ and where every collider is in **vancs** and every non-collider is not in **vancs**.*

*Proof.* By definition of the N-node, $N \not\perp\!\!\!\perp X \mid$ **vancs**. Now all paths that end with a tail at $N$ are blocked given **vancs** because $N \notin an(X)$ and the first collider $K$ coming from $N$ must be blocked because $K \notin$ **vancs**. Hence, there must be an open path that ends with a head at $N$ and where every collider is in **vancs** and every non-collider is not in **vancs** as stated. $\square$

## B.7 Proof of Theorem 1

**Theorem** (Validity of O-set). Given Assumptions 1 but *without* a priori assuming that a valid adjustment set exists (apart from the requirement $\mathbf{S} \cap$ **forb** $= \emptyset$). If and only if a valid backdoor adjustment set exists, then $\mathbf{O}$ is a valid adjustment set.

*Proof.* **"if"**: Given that a valid backdoor adjustment set exists, we need to prove that (i) $\mathbf{O} \cap$ **forb** $= \emptyset$ with **forb** $= X \cup des(Y\mathbf{M})$ and (ii) all non-causal paths from $X$ to $Y$ are blocked by $\mathbf{O}$ (given $\mathbf{S}$). (i) is true by the construction of $\mathbf{O}$ in Def. 4 and Alg. C.1 where nodes $\in des(Y\mathbf{M})$ are not added and nodes that are $X$ indicate non-identifiability (see Lemma B.2). By Lemma B.3 also $\mathbf{P_C} \cap des(Y\mathbf{M}) = \emptyset$ and $X \notin \mathbf{P_C}$ because otherwise no valid adjustment set exists by Lemma B.2.

Lemma B.1 lists all possible motifs on non-causal paths. By Lemma B.2 the occurrence of the motifs (1a) or (4a) or (4b) for $F = X$ renders the effect non-identifiable, contrary to the assumption. Hence

only the remaining motifs can occur. By Lemma B.4 the motifs (1b), (2a), (2b), (3a), (3b) are blocked given $\mathbf{OS}$. By Lemma B.5 (part one) the motifs (4a,b) for $F \in des(Y\mathbf{M})$ are blocked given $\mathbf{OS}$. By Lemma B.6 (part one) motifs (5a) and (5b) are blocked given $\mathbf{OS}$.

**"only if"** is trivially true since $\mathbf{O}$ is then assumed valid. $\qquad\square$

## B.8 Proof of Theorem 2

**Theorem** (O-set vs Adjust-set ). Given Assumptions 1 with $\mathbf{O}$ defined in Def. 4 and the Adjust-set **vancs** defined in Eq. (2), it holds that $J_\mathbf{O} \geq J_\mathbf{vancs}$ for any graph $\mathcal{G}$. We have $J_\mathbf{O} = J_\mathbf{vancs}$ only if $\mathbf{O} = \mathbf{vancs}$ or $\mathbf{O} \subseteq \mathbf{vancs}$ and $X \perp\!\!\!\perp \mathbf{vancs} \setminus \mathbf{O} \mid \mathbf{OS}$.

*Proof.* We directly use the decomposition in Eq. (S5) with $\mathbf{Z} = \mathbf{O} = \mathbf{AB}$ and $\mathbf{Z}' = \mathbf{vancs} = \mathbf{BR}$ with $\mathbf{vancs} = an(XY\mathbf{S}) \setminus \mathbf{forb}$ and the definitions of $\mathbf{R}, \mathbf{B}, \mathbf{A}$ as in Eq. (S5). For term (iii), $I_{\mathbf{R};Y\mid\mathbf{O}X\mathbf{S}}$, to be non-zero, there must be an active path from $\mathbf{R} \subseteq \mathbf{vancs}$ to $Y$ given $X\mathbf{OS}$. By Lemma B.1, Lemma B.4, Lemma B.5 (second part), and Lemma B.6 (second part), the only possibly open motifs on paths from $\mathbf{R}$ to $Y$ given $\mathbf{O}X\mathbf{S}$ are "$\leftarrow N \leftrightarrow W$" or "$\leftarrow N \leftrightarrow \boxed{C} \leftrightarrow$" where $\mathbf{R} \cap des(N) \neq \emptyset$. But since $\mathbf{R} \subseteq \mathbf{vancs}$ and $N \notin \mathbf{vancs}$, $\mathbf{R}$ cannot contain descendants of $N$. Hence, term (iii) is zero. For term (iv), $I_{X;\mathbf{A}\mid\mathbf{BRS}} = I_{X;\mathbf{A}\mid\mathbf{vancs}}$, note that $\mathbf{A} = \mathbf{O} \setminus \mathbf{vancs}$ and, hence, for all $A \in \mathbf{A}$ it holds that $A \perp\!\!\!\perp X \mid \mathbf{vancs}$ since all $A \in \mathbf{A}$ then fulfill Def. 3(2b) (for $A \in \mathbf{P_C}$ see Lemma B.3). Hence, $I_{X;\mathbf{A}\mid\mathbf{vancs}} = 0$ by Markovity. This proves that $J_\mathbf{O} \geq J_\mathbf{vancs}$.

We are now left with terms (i) and (ii) in Eq. (S5). By construction of the collider path nodes, $\mathbf{A} \subseteq \mathbf{CP_C}$ is connected to $Y$ (potentially through $\mathbf{M}$) conditional on $\mathbf{vancs}X$ since $\mathbf{vancs}$ contains all remaining collider nodes in $\mathbf{C}$. Then by Faithfulness term (i) $I_{\mathbf{A};Y\mid\mathbf{BR}X\mathbf{S}} = I_{\mathbf{A};Y\mid\mathbf{vancs}X}$ can only be zero if $\mathbf{A} = \emptyset$. Then $\mathbf{O} \subseteq \mathbf{vancs}$. Term (ii), $I_{X;\mathbf{R}\mid\mathbf{OS}} = 0$ if $\mathbf{R} = \mathbf{vancs} \setminus \mathbf{O} = \emptyset$ or $X \perp\!\!\!\perp \mathbf{vancs} \setminus \mathbf{O} \mid \mathbf{OS}$ together with Faithfulness. $\qquad\square$

## B.9 Proof of Proposition B.2

**Proposition** (Collider-minimized O-set is a subset of Adjust.). Given Assumptions 1 with $\mathbf{O} = \mathbf{PCP_C}$ defined in Def. 4 and the $\mathbf{O}_{\mathrm{Cmin}}$-set constructed with Alg. C.2 it holds that $\mathbf{O}_{\mathrm{Cmin}} \subseteq \mathbf{vancs}$.

*Proof.* Define $\mathbf{C}_{\min} = \mathbf{O}_{\mathrm{Cmin}} \setminus \mathbf{P}$. We need to show that $C \in \mathbf{C}_{\min} \Rightarrow C \in \mathbf{vancs}$ for all $C \in \mathbf{O} \setminus \mathbf{P}$. Assume $C \notin \mathbf{vancs}$. Since then all $C \in \mathbf{O} \setminus \mathbf{P}$ fulfill Def. 3(2b) (for $C \in \mathbf{P_C}$ see Lemma B.3), it holds that $C \perp\!\!\!\perp X \mid \mathbf{vancs}$ implying that no link $X \ast\!\!-\!\!\ast C$ exists. If a path exists at all, either (i) there must be at least one collider $K$ with $C \in an(K)$ and $K \notin \mathbf{vancs}$ on a path to $X$ or (ii) $C \in des(X)$. We now show that for case (i) $C$ has no open path to $X$ given $\mathbf{SO} \setminus \{C\}$. $K \notin an(\mathbf{OS})$ can be seen by considering the different parts of $\mathbf{OS}$: $K \notin an(\mathbf{PS})$ since $K \notin \mathbf{vancs}$ and $an(\mathbf{PS}) \subseteq \mathbf{vancs}$. Further, $K \notin an(\mathbf{vancs} \cap \mathbf{C})$. Finally, $K \notin an(\mathbf{CP_C} \setminus \mathbf{vancs})$ since $C' \in \mathbf{CP_C} \setminus \mathbf{vancs}$ fulfill (by Def. 3(2b)) $C' \perp\!\!\!\perp X \mid \mathbf{vancs}$ and $K \not\perp\!\!\!\perp X \mid \mathbf{vancs}$. Hence, $X \perp\!\!\!\perp C \mid \mathbf{SO} \setminus \{C\}$ implying that $C$ would be removed in the first loop of Alg. C.2 and $C \notin \mathbf{C}_{\min}$, contrary to assumption.

In case (ii) the directed path from $X$ to $C$ for $C \in \mathbf{C} \setminus \mathbf{P_C}$ is blocked because $\mathbf{P_C} \subseteq \mathbf{O}$ contains all parents of $C$ and $X \notin \mathbf{P_C}$ since we assume identifiability. This implies that $C$ would be removed in the first loop of Alg. C.2 and $C \notin \mathbf{C}_{\min}$, contrary to assumption. Finally, if there exists a directed path from $X$ to $C = P_C \in \mathbf{P_C} \setminus \mathbf{C}$ for $P_C \notin \mathbf{vancs}$ we know that all children $C \in ch(P_C) \cap \mathbf{CP}$ were removed in the first loop of Alg. C.2. Denote the remaining nodes after the first loop of Alg. C.2 by $\mathbf{O}'_{\mathrm{Cmin}}$. $P_C \notin \mathbf{vancs}$ has no directed path to $Y$ and is separated from $Y$ given $\mathbf{SO}'_{\mathrm{Cmin}}$ because the motif $P_C \rightarrow C \leftrightarrow$ is blocked since $C \notin an(\mathbf{O}'_{\mathrm{Cmin}})$. This implies that $P_C$ would be removed in the second loop of Alg. C.2 and $P_C \notin \mathbf{C}_{\min}$, contrary to assumption. $\qquad\square$

## B.10 Proof of Theorem 3

**Theorem** (Necessary and sufficient graphical conditions for optimality and optimality of O-set). Given Assumptions 1 and with $\mathbf{O} = \mathbf{PCP_C}$ defined in Def. 4. Denote the set of N-nodes by $\mathbf{N} = sp(Y\mathbf{MC}) \setminus (\mathbf{forbOS})$. Finally, given an $N \in \mathbf{N}$ and a collider path $N \leftrightarrow \cdots \leftrightarrow C \leftrightarrow \cdots \leftrightarrow W$ (including $N \leftrightarrow W$) for $C \in \mathbf{C}$ and $W \in Y\mathbf{M}$ (indexed by $i$) with the collider path nodes denoted by $\pi_i^N$ (excluding $N$ and $W$), denote by $\mathbf{O}_{\pi_i^N} = \mathbf{O}(X, Y, \mathbf{S}' = \mathbf{S}N\pi_i^N)$ the O-set for the causal effect of $X$ on $Y$ given $\mathbf{S}' = \mathbf{S} \cup \{N\} \cup \pi_i^N$.

If and only if exactly one valid adjustment set exists, or both of the following conditions are fulfilled, then graphical optimality holds and $\mathbf{O}$ is optimal:

(I) For *all* $N \in \mathbf{N}$ and all its collider paths $i$ to $W \in Y\mathbf{M}$ that are inside $\mathbf{C}$ it holds that $\mathbf{O}_{\pi_i^N}$ does not block all non-causal paths from $X$ to $Y$, i.e., $\mathbf{O}_{\pi_i^N}$ is non-valid,

and

(II) for all $E \in \mathbf{O} \setminus \mathbf{P}$ with an open path to $X$ given $\mathbf{SO} \setminus \{E\}$ there is a link $E \leftrightarrow W$ or an extended collider path $E \ast\!\!\to C \leftrightarrow \cdots \leftrightarrow W$ inside $\mathbf{C}$ for $W \in Y\mathbf{M}$ where all colliders $C \in \mathbf{vancs}$.

*Proof.* If exactly one valid adjustment set exists, then optimality holds by Def. 2 and then this set is $\mathbf{O}$ because $\mathbf{O}$ is always valid if a valid set exists (Lemma 1).

The proof is based on Lemma 1 and relation (S5). We will first prove the "if"-statement by showing that Cond. (I) leads to term (i)$\geq$(iii) and Cond. (II) leads to term (ii)$\geq$(iv) from which optimality follows by Lemma 1. Then we prove the "only if"-statement by showing that if either of the two conditions is not fulfilled, then (i)$<$(iii) or (ii)$<$(iv) for some distribution $\mathcal{P}$ consistent with $\mathcal{G}$ and graphical optimality does not hold.

**"if"**: We have to show that if both conditions hold, then $I_{\mathbf{A};Y|\mathbf{BR}XS} \geq I_{\mathbf{R};Y|\mathbf{AB}XS}$ and $I_{X;\mathbf{R}|\mathbf{ABS}} \geq I_{X;\mathbf{A}|\mathbf{BRS}}$ where $\mathbf{O} = \mathbf{AB}$ and $\mathbf{Z}' = \mathbf{RB}$ with $\mathbf{B} = \mathbf{O} \cap \mathbf{Z}'$. Further, we use $\mathbf{A_P} = \mathbf{A} \cap \mathbf{P}$ and $\mathbf{A_C} = (\mathbf{A} \cap \mathbf{CP_C}) \setminus \mathbf{A_P}$ where $\mathbf{A} = \mathbf{A_P} \cup \mathbf{A_C}$.

Condition (I) directly leads to $I_{\mathbf{A};Y|\mathbf{BR}XS} \geq I_{\mathbf{R};Y|\mathbf{AB}XS}$ as follows.

We subdivide condition (I) into two cases where the former implies the latter: (I.1) There are no N-nodes, i.e., $\mathbf{N} = \emptyset$, or (I.2) for *all* $N \in \mathbf{N}$ and all its collider paths $i$ it holds that $\mathbf{O}_{\pi_i^N}$ does not block all non-causal paths from $X$ to $Y$.

If condition (I.1) holds, then there are no N-nodes. If there are no N-motifs on any path from $\mathbf{R}$ to $Y$, then by Lemma B.1, Lemma B.4, and Lemma B.5 (second part) all paths given $X\mathbf{OS}$ are blocked and term (iii) is zero by Markovity.

If condition (I.2) holds, then there are N-nodes. By Lemma B.6 (second part) the only possibly open motifs on paths from $\mathbf{R}$ to $Y$ given $\mathbf{O}XS$ are "$\leftarrow N \leftrightarrow W$" or "$\leftarrow N \leftrightarrow \boxed{C} \leftrightarrow$" where $\mathbf{R} \cap des(N) \neq \emptyset$. Term (iii), $I_{\mathbf{R};Y|\mathbf{B}XS\mathbf{A}} = I_{\mathbf{R};Y|X\mathbf{SO}}$, is then always non-zero since, by definition of the N-nodes, there exists at least one collider path $N \leftrightarrow \cdots \leftrightarrow \boxed{C} \leftrightarrow \cdots \leftrightarrow W$ (including $N \leftrightarrow W$) for $C \in \mathbf{C}$ and $W \in Y\mathbf{M}$. To see under which conditions still term (i)$\geq$(iii) consider two ways of decomposing the following CMI:

$$
\begin{aligned}
I_{\mathbf{AR};Y|\mathbf{B}XS} &= \underbrace{I_{\mathbf{A};Y|\mathbf{B}XS}}_{\text{term (i')}} + \underbrace{I_{\mathbf{R};Y|\mathbf{B}XS\mathbf{A}}}_{\text{term (iii)}} \\
&= \underbrace{I_{\mathbf{R};Y|\mathbf{B}XS}}_{\text{term (iii')}} + \underbrace{I_{\mathbf{A};Y|\mathbf{B}XS\mathbf{R}}}_{\text{term (i)}} \cdot
\end{aligned}
\tag{S8}
$$

From this decomposition we see that term (i)$\geq$(iii) if and only if term (i')$\geq$(iii'). Paths from $\mathbf{R}$ to $Y$ via $X$ given $\mathbf{S}X\mathbf{Z}' \setminus \mathbf{R} = \mathbf{BS}X$ are blocked because if $X$ is a collider, then there would be a non-causal path rendering $\mathbf{Z}'$ invalid. Therefore, for term (iii') to be non-zero $\mathbf{Z}'\mathbf{S}$ must contain at least descendants of an N-node $N$ and all its collider path nodes towards $W$, denoted $\pi_i^N$, for at least one path $i$. Then $\mathbf{R} \cap des(N) \neq \emptyset$ and $\pi_i^N \subseteq \mathbf{BS}$ such that there exists an open path "$N \leftrightarrow \boxed{C} \leftrightarrow \cdots \leftrightarrow \boxed{C} \leftrightarrow W$" (or $N \leftrightarrow W$).

Condition (I.2) now guarantees that for all $N \in \mathbf{N}$ and all collider paths indexed by $i$ the O-set $\mathbf{O}_{\pi_i^N}$, which includes $N\pi_i^N$ as a subset, does *not* block all non-causal paths. By Theorem 1, if $\mathbf{O}_{\pi_i^N}$ is not valid, then no valid adjustment set $\mathbf{Z}'$ containing $N\pi_i^N$ as a subset exists. And this in turn implies that no valid set with $\mathbf{R} \cap des(N) \neq \emptyset$ exists. To show this, assume the contraposition: If there was such a valid set $\mathbf{Z}'$ with $\mathbf{R} \cap des(N) \neq \emptyset$ and $\pi_i^N \subset \mathbf{Z}'$, then it would open the collider motif $\ast\!\!\to N \leftrightarrow$ since $\mathbf{R}$ contains descendants of $N$ and lead to an open path "$N \leftrightarrow \boxed{C} \leftrightarrow \cdots \leftrightarrow \boxed{C} \leftrightarrow W$" (or $N \leftrightarrow W$). If $\mathbf{Z}'$ is still valid, it must block all paths from $X$ that end with an arrowhead at $N$. But then also $\mathbf{Z}' \cup \{N\}$ is valid. Note that since $N \notin \mathbf{forb}$, $\mathbf{S} \cap \mathbf{forb} = \emptyset$, and $\pi_i^N \cap \mathbf{forb} = \emptyset$ since

$\pi_i^N \subseteq \mathbf{C}$, the validity of $\mathbf{O}_{\pi_i^N}$ depends only on its ability to block non-causal paths. Hence, term (iii') is zero and by Eq. (S8) term (i)$\geq$(iii).

Condition (II) directly leads to $I_{X;\mathbf{R}|\mathbf{ABS}} \geq I_{X;\mathbf{A}|\mathbf{BRS}}$ as follows.

Define $\mathbf{E} = \{E \in \mathbf{O} \setminus \mathbf{P} : X \not\perp\!\!\!\perp E \mid \mathbf{SO} \setminus \{E\}\}$. By condition (II) there exists a link $E \leftrightarrow W$ or an extended collider path $E \ast\!\!\to C \leftrightarrow \cdots \leftrightarrow W$ inside $\mathbf{C}$ for $W \in Y\mathbf{M}$ where all colliders $C \in \mathbf{vancs}$. There are two types: (1) $E \to C \leftrightarrow \cdots \leftrightarrow W$ (then $E \in \mathbf{P_C}$) and (2) $E \leftrightarrow W$ or $E \leftrightarrow C \leftrightarrow \cdots \leftrightarrow W$. We consider two cases:

Case (1): $E \in \mathbf{E}$ for which there exists *at least one* path of type (1). Any valid $\mathbf{Z}'$ with $E \notin \mathbf{Z}'$ has to block paths from $X$ to $E$ since otherwise there is a non-causal open path from $X$ to $Y$ through the motif chain $\ast\!\!-\!\!\ast E \to C \leftrightarrow \cdots \leftrightarrow W$ for $W \in Y\mathbf{M}$: $E$ is open since $E \notin \mathbf{Z}'$ and the part from $E$ to $W$ is open since all colliders $C \in \mathbf{vancs}$: if $C \in an(\mathbf{S})$, the collider is always opened and if $C \in an(XY)$ then either the directed path to $X$ or $Y$ is open, or $C$ is opened if $\mathbf{Z}'$ contains a node on that path.

Case (2): $E \in \mathbf{E}$ for which *all paths* are of type (2). Firstly, all paths from $X$ to $E$ that end with a tail at $E$ must be blocked by $\mathbf{Z}'$ since otherwise there is a non-causal path as for case (1). The same holds for paths that end with a head at $E$ if $E \in \mathbf{vancs}$. Consider paths that end with a head at $E$ and $E \notin \mathbf{vancs}$ which implies $E \perp\!\!\!\perp X \mid \mathbf{vancs}$ by Def. 3. Then it follows that $E \perp\!\!\!\perp X \mid \mathbf{SO} \setminus \{E\}$ and, hence, $E \notin \mathbf{E}$ which we can show by considering where $E$ can occur with respect to the different motifs listed in Lemma B.1 as follows (see the definitions of $W, V, F, N, C, P_C$ there): Motif (1a) "$\ast\!\!-\!\!\ast X \to \boxed{C} \leftrightarrow$" is not relevant since then non-identifiability holds and motif (2a) "$X, V \ast\!\!-\!\!\ast \boxed{P} \to W$" is not relevant since $\mathbf{E} \notin \mathbf{P}$. Motifs (3a) "$V \leftarrow W$", (4a) "$\ast\!\!-\!\!\ast F \leftrightarrow W$", and (5a) "$\ast\!\!-\!\!\ast N \leftrightarrow W$" are not relevant since no $E \in \mathbf{O}$ is involved. For the motifs (1b) "$\ast\!\!-\!\!\ast X \to \boxed{P_C} \to E \leftrightarrow$", (2b) "$X, V \ast\!\!-\!\!\ast \boxed{P_C} \to \boxed{C} \leftrightarrow \cdots \leftrightarrow E$", and (3b) "$X, V \leftarrow \boxed{C} \leftrightarrow \cdots \leftrightarrow E$" the path to $X$ is blocked by $\mathbf{SO} \setminus \{E\}$. For motif (4b) "$\ast\!\!-\!\!\ast F \leftrightarrow \boxed{C} \leftrightarrow \cdots \leftrightarrow E$" or "$\ast\!\!-\!\!\ast F \leftrightarrow E$", since $\mathbf{SO} \cap \mathbf{forb} = \emptyset$ and $X \cap des(\mathbf{forb}) = \emptyset$, there must exist a collider $\in \mathbf{forb}$ or $\ast\!\!\to F \leftrightarrow$ on a path to $X$ which is then blocked. Hence, $E \notin \mathbf{E}$. Finally, for (5b) "$\ast\!\!-\!\!\ast N \leftrightarrow \boxed{C} \leftrightarrow \cdots \leftrightarrow E$" or "$\ast\!\!-\!\!\ast N \leftrightarrow E$" with $N \notin \mathbf{vancs}$ and $X \not\perp\!\!\!\perp N \mid \mathbf{vancs}$ we either have $\ast\!\!\to N \leftrightarrow$ or there exists a collider on any path to $X$ with $K \in des(N)$ and, hence, $K \notin \mathbf{vancs}$. $E \not\perp\!\!\!\perp X \mid \mathbf{SO} \setminus \{E\}$ would only be possible if $N$ or $K \in an(\mathbf{O} \setminus \mathbf{vancs})$. The subset $\mathbf{O} \setminus \mathbf{vancs}$ fulfills $\mathbf{O} \setminus \mathbf{vancs} \perp\!\!\!\perp X \mid \mathbf{vancs}$ by Def. 3. However, $N$ or $K \in an(\mathbf{O} \setminus \mathbf{vancs})$ implies that there is a path from $\mathbf{O} \setminus \mathbf{vancs}$ to $N$. Then $X \not\perp\!\!\!\perp N \mid \mathbf{vancs}$ contradicts $\mathbf{O} \setminus \mathbf{vancs} \perp\!\!\!\perp X \mid \mathbf{vancs}$ implying that $N, K \notin an(\mathbf{O} \setminus \mathbf{vancs})$ and, hence, $E \perp\!\!\!\perp X \mid \mathbf{SO} \setminus \{E\}$ and $E \notin \mathbf{E}$.

Both cases taken together, it holds that $X \perp\!\!\!\perp E \mid \mathbf{SZ}' \setminus \{E\}$ for any valid $\mathbf{Z}'$. Furthermore, $X \perp\!\!\!\perp P \mid \mathbf{SZ}' \setminus \{P\}$ with $P \in \mathbf{P}$ for any valid $\mathbf{Z}'$ since $\mathbf{P}$ is directly connected to $Y$ and, therefore, a valid $\mathbf{Z}'$ has to block a non-causal path between $X$ and $Y$ through $\mathbf{P}$.

Now decompose term (iv) as

$$I_{X;\mathbf{A}|\mathbf{Z}'\mathbf{S}} = \underbrace{I_{X;\mathbf{A_P A_E}|\mathbf{Z}'\mathbf{S}}}_{=0} + I_{X;\mathbf{A} \setminus (\mathbf{A_P A_E})|\mathbf{Z}'\mathbf{S}\mathbf{A_P A_E}} \tag{S9}$$

with $\mathbf{A_P} = \mathbf{A} \cap \mathbf{P}$ and $\mathbf{A_E} = (\mathbf{A} \cap \mathbf{E}) \setminus \mathbf{A_P}$. The preceding derivations imply $X \perp\!\!\!\perp \mathbf{A_P A_E}|\mathbf{Z}'\mathbf{S}$ for any valid $\mathbf{Z}'$ and, hence, the first term vanishes.

Consider the set $\mathbf{E}' = \{E' \in \mathbf{O} \setminus \mathbf{P} : X \perp\!\!\!\perp E' \mid \mathbf{SO} \setminus \{E'\}\}$. This implies that $\mathbf{A_{E'}} = \mathbf{A} \setminus (\mathbf{A_P A_E})$ fulfills $\mathbf{A_{E'}} \perp\!\!\!\perp X \mid \mathbf{SO} \setminus \mathbf{A_{E'}}$ and since $\mathbf{SO} \setminus \mathbf{A_{E'}} = \mathbf{SBA_P A_E}$ we have

$$I_{X;\mathbf{A_{E'}}|\mathbf{SBA_P A_E}} = 0. \tag{S10}$$

This now leads to term (ii) $\geq$ term (iv) by considering two ways of decomposing the following CMI:

$$I_{X;\mathbf{RA_{E'}}|\mathbf{SBA_P A_E}} = \underbrace{I_{X;\mathbf{A_{E'}}|\mathbf{SBA_P A_E}}}_{=0 \text{ by Eq. (S10)}} + \underbrace{I_{X;\mathbf{R}|\mathbf{SBA_P A_E A_{E'}}}}_{\text{term (ii)}} \tag{S11}$$

$$= \underbrace{I_{X;\mathbf{R}|\mathbf{SBA_P A_E}}}_{\geq 0} + \underbrace{I_{X;\mathbf{A_{E'}}|\mathbf{SBA_P A_E R}}}_{\text{term (iv) by Eq. (S9)}}. \tag{S12}$$

**"only if"**: We need to prove that if either Condition (I) or Condition (II) or both are not fulfilled, then graphical optimality does not hold (implying that also $\mathbf{O}$ is not optimal).

The negation of Condition (I) directly leads to $I_{\mathbf{A};Y|\mathbf{BR}XS} < I_{\mathbf{R};Y|\mathbf{AB}XS}$ for some distribution $\mathcal{P}$ consistent with $\mathcal{G}$ as follows: There exists at least one N-node with at least one collider path $N\leftrightarrow\cdots\leftrightarrow C\leftrightarrow\cdots\leftrightarrow W$ (including $N\leftrightarrow W$) for $C \in \mathbf{C}$ and $W \in Y\mathbf{M}$ (indexed by $i$) with collider path nodes denoted $\pi_i^N$ such that $\mathbf{O}_{\pi_i^N}$ blocks all non-causal paths from $X$ to $Y$. $\mathbf{O}_{\pi_i^N}$ is the O-set for the causal effect of $X$ on $Y$ given $\mathbf{S}' = \mathbf{S} \cup \{N\} \cup \pi_i^N$. Consider $\mathbf{Z}' = \mathbf{O}_{\pi_i^N}$. Since also $N \notin \mathbf{forb}$, $\mathbf{S} \cap \mathbf{forb} = \emptyset$, and $\pi_i^N \cap \mathbf{forb} = \emptyset$, $\mathbf{Z}' = \mathbf{O}_{\pi_i^N}$ is valid. Since $N \in \mathbf{O}_{\pi_i^N}$ while $N \notin \mathbf{O}$, we have $\mathbf{R} \neq \emptyset$, and since $\pi_i^N \subseteq \mathbf{C}$ we have $\pi_i^N \subseteq \mathbf{BS}$ and there exists an open path $N\leftrightarrow\boxed{C}\leftrightarrow\cdots\leftrightarrow\boxed{C}\leftrightarrow W$ (or $N\leftrightarrow W$) such that $I_{\mathbf{R};Y|\mathbf{B}XS} > 0$. Similar to Lemma 1 we can now construct a distribution $\mathcal{P}$ with associated SCM consistent with $\mathcal{G}$ where all links "$U{*\!\!-\!\!*}A$" for $A \in \mathbf{A}$ *almost vanish* and, hence, term (i'), $I_{\mathbf{A};Y|\mathbf{B}XS} \to 0$: Consider the three possible links and associated arbitrary assignment functions in the SCM: (1) "$A\to U$" with $U := f_U(\ldots, cA, \ldots)$, (2) "$A\leftarrow U$" with $A := f_A(\ldots, cU, \ldots)$, and (3) "$A\leftrightarrow U$" with $A := f_A(\ldots, cL^U, \ldots)$ where $L^U$ denotes one or more latent variables and $c \in \mathbb{R}$. We then consider the limit $c \to 0$ leading to term (i'), $I_{\mathbf{A};Y|\mathbf{B}XS} \to 0$. Since $\mathbf{A} \cap N\pi_i^N = \emptyset$ this does not affect the collider path $N\leftrightarrow\boxed{C}\leftrightarrow\cdots\leftrightarrow\boxed{C}\leftrightarrow W$ (or $N\leftrightarrow W$) such that $I_{\mathbf{R};Y|\mathbf{B}XS} > 0$. By Eq. (S8) then term (i)<(iii). By Lemma 1, where $\mathcal{P}$ is further modified to $\mathcal{P}'$ without affecting term (i)<(iii), then graphical optimality does not hold.

Alternatively, the negation of Condition (II) directly leads to $I_{X;\mathbf{R}|\mathbf{ABS}} < I_{X;\mathbf{A}|\mathbf{BRS}}$ as follows: By the negation of Condition (II) there exists an $E \in \mathbf{O} \setminus \mathbf{P}$ with $X \not\perp\!\!\!\perp E \mid \mathbf{SO} \setminus \{E\}$ such that there is no link $E\leftrightarrow W$ and all extended collider paths $E{*\!\!\to}C\leftrightarrow\cdots\leftrightarrow W$ inside $\mathbf{C}$ for $W \in Y\mathbf{M}$ contain at least one collider $C \notin \mathbf{vancs}$. Define the set of these non-ancestral colliders as

$$\mathbf{C}_E = \{C \in \mathbf{C} : E{*\!\!\to}\cdots\leftrightarrow C\leftrightarrow\cdots\leftrightarrow W\} \setminus \mathbf{vancs}. \tag{S13}$$

We define $E_{\mathbf{C}} = \{E\} \cup (des(\mathbf{C}_E) \cap \mathbf{O})$ and choose $\mathbf{Z}' = \mathbf{O} \setminus E_{\mathbf{C}}$ implying $\mathbf{A} = E_{\mathbf{C}}, \mathbf{B} = \mathbf{O} \setminus E_{\mathbf{C}}$, and $\mathbf{R} = \emptyset$. We need to show that (1) $\mathbf{Z}'$ is valid and (2) $I_{X;\mathbf{A}|\mathbf{BRS}} = I_{X;E_{\mathbf{C}}|\mathbf{SO}\setminus E_{\mathbf{C}}} > I_{X;\mathbf{R}|\mathbf{ABS}} = 0$ (since $\mathbf{R} = \emptyset$).

Ad (1): As a subset of $\mathbf{O}$ we have that $\mathbf{Z}' \cap \mathbf{forb} = \emptyset$. We investigate whether $\mathbf{Z}'$ blocks all non-causal paths between $X$ and $Y$ by considering the motifs in Lemma B.1. In addition to all those motifs listed there there are modified motifs where unconditioned $C$-nodes and $P_C$-nodes occur (denoted without a $\boxed{\cdot}$) due to removing $E_{\mathbf{C}}$ from $\mathbf{O}$.

Firstly, the unmodified motifs are blocked as before (see Theorem 1): Motif (1a) "$*\!\!-\!\!*X\to\boxed{C}\leftrightarrow$" is not relevant since then non-identifiability holds. By Lemma B.4 the motifs (1b), (2a), (2b), (3a), (3b) all contain a non-collider in $\mathbf{SO} \setminus E_{\mathbf{C}}$ and are blocked. By Lemma B.5 (part one) the motifs (4a,b) for $F \in des(Y\mathbf{M})$ are blocked because $\mathbf{Z}' \cap \mathbf{forb} = \emptyset$. By Lemma B.6 (part one) motifs (5a) and (5b) are blocked given $\mathbf{SO} \setminus E_{\mathbf{C}}$ because the proof in Lemma B.6 requires that on paths to $X$ either $N$ is a collider or there exists a descendant collider $K$ and that $N, K \notin an(\mathbf{OS})$. The latter is fulfilled because $\mathbf{SO} \setminus E_{\mathbf{C}}$ is a subset of $\mathbf{SO}$.

Secondly, all paths from $X$ through the removed node $E$ to $W \in Y\mathbf{M}$ are blocked by $\mathbf{SO} \setminus E_{\mathbf{C}}$: Paths through $\mathbf{P}$ are blocked since $E \notin \mathbf{P}$ and $des(\mathbf{C}_E) \cap \mathbf{vancs} = \emptyset$ and, hence, $\mathbf{P} \subseteq \mathbf{SO} \setminus E_{\mathbf{C}}$. Paths through colliders are blocked by the negation of condition (II): there is no link $E\leftrightarrow W$ and all extended collider paths $E{*\!\!\to}C\leftrightarrow\cdots\leftrightarrow W$ inside $\mathbf{C}$ for $W \in Y\mathbf{M}$ contain at least one collider $C \notin \mathbf{vancs}$. By construction, $E_{\mathbf{C}} = \{E\} \cup (des(\mathbf{C}_E) \cap \mathbf{O})$, implying that all these non-ancestral colliders are blocked.

Thirdly, we consider the modified motifs with unconditioned $C, P_C \in (des(\mathbf{C}_E) \cap \mathbf{O})$. By definition of $\mathbf{C}_E$ in (S13), $C, P_C \notin \mathbf{vancs}$. (As a remark, $E$ can potentially be in $\mathbf{vancs}$.) Motif (1a) "$*\!\!-\!\!*X\to\boxed{C}\leftrightarrow$" cannot occur since then non-identifiability holds. Modified motifs (1,2b') "$X, V{*\!\!-\!\!*}\boxed{P_C}\to C\leftrightarrow$" and (1,2b'') "$X, V{*\!\!-\!\!*}\boxed{P_C}\to\boxed{C}\leftrightarrow\cdots\leftrightarrow C\leftrightarrow$" are blocked since they contain a non-collider. Motifs (1,2b''') "$X, V{*\!\!-\!\!*}P_C\to C\leftrightarrow$" are blocked since $C \notin des(\mathbf{SO} \setminus E_{\mathbf{C}})$. Motif (2a') "$X, V{*\!\!-\!\!*}P\to W$" is not possible since $P \in \mathbf{P} \subseteq \mathbf{vancs}$. Motifs (3a) "$V\leftarrow W$", (4a) "$*\!\!-\!\!*F\leftrightarrow W$", and (5a) "$*\!\!-\!\!*N\leftrightarrow W$" are not modified since no conditioned node occurs. Motif (3b') "$X, V\leftarrow C\leftrightarrow$" is blocked because due to $C \notin \mathbf{vancs}$ there must exist a descendant of $C$ that is a collider $K \notin \mathbf{vancs}$ on the path to $X$. Since $E_{\mathbf{C}}$ contains all descendants of $C$, also $K$ and all its descendants are not in $\mathbf{SO} \setminus E_{\mathbf{C}}$ and $K$ is blocked. Finally, motifs (4b') "$*\!\!-\!\!*F\leftrightarrow C\leftrightarrow$" and (5b') "$*\!\!-\!\!*N\leftrightarrow C\leftrightarrow$" are blocked since $\mathbf{SO} \setminus E_{\mathbf{C}}$ does not contain any descendant of $C$. This proves the validity of $\mathbf{Z}'$.

Ad (2): To show that $I_{X;\mathbf{A}|\mathbf{BRS}} = I_{X;E_{\mathbf{C}}|\mathbf{SO}\setminus E_{\mathbf{C}}} > I_{X;\mathbf{R}|\mathbf{ABS}} = 0$ we start from the assumption that $E \not\perp\!\!\!\perp X \mid \mathbf{SO} \setminus \{E\}$. This implies that there exists a path from $X$ to $E$ where no non-collider is in $\mathbf{SO} \setminus \{E\}$ and for every collider $K$ it holds that $des(K) \cap \mathbf{SO} \setminus \{E\} \neq \emptyset$. With $\mathbf{Z}' = \mathbf{O} \setminus E_{\mathbf{C}}$ all non-colliders are still open. Consider those colliders $K$ with $des(K) \cap (\mathbf{O} \setminus \{E\}) \subseteq E_{\mathbf{C}} \setminus \{E\}$. Then these colliders are closed on the path from $E$ to $X$. However, for each such $K$ there is a $C \in E_{\mathbf{C}} \setminus \{E\}$ with $C \in des(K)$. Then the path from $X$ through $*\!\!\to K \to \cdots \to C$ is open given $\mathbf{SO} \setminus E_{\mathbf{C}}$. Hence, at least for the last such collider on the path from $E$ to $X$ there is an open path from $C \in E_{\mathbf{C}} \setminus \{E\}$ to $X$ given $\mathbf{SO} \setminus E_{\mathbf{C}}$. Then Faithfulness implies that $I_{X;\mathbf{A}|\mathbf{BRS}} = I_{X;E_{\mathbf{C}}|\mathbf{SO}\setminus E_{\mathbf{C}}} > I_{X;\mathbf{R}|\mathbf{ABS}} = 0$ and, hence, term (ii) < term (iv) holds for all distributions $\mathcal{P}$ consistent with $\mathcal{G}$. By Lemma 1, where the distribution $\mathcal{P}$ is modified to $\mathcal{P}'$ without affecting term (ii)<(iv), then graphical optimality does not hold.

This concludes the proof of Theorem 3. $\qquad\square$

### B.11 Proof of Corollary B.1

**Corollary** (Minimality and minimum cardinality). Given Assumptions 1, assume that graphical optimality holds, and, hence, $\mathbf{O}$ is optimal. Further it holds that:

1. If $\mathbf{O}$ is not minimal, then $J_{\mathbf{O}} > J_{\mathbf{Z}}$ for all *minimal* valid $\mathbf{Z} \neq \mathbf{O}$,

2. If $\mathbf{O}$ is minimal valid, then $\mathbf{O}$ is the unique set that maximizes the adjustment information $J_{\mathbf{Z}}$ among all *minimal* valid $\mathbf{Z} \neq \mathbf{O}$,

3. $\mathbf{O}$ is of minimum cardinality, that is, there is no subset of $\mathbf{O}$ that is still valid and optimal.

*Proof.* We again define disjunct sets $\mathbf{R}, \mathbf{B}, \mathbf{A}$ with $\mathbf{A} = \mathbf{O} \setminus \mathbf{Z}$, $\mathbf{R} = \mathbf{Z} \setminus \mathbf{O}$, and $\mathbf{B} = \mathbf{O} \cap \mathbf{Z}$, where any of them can be empty, but not both $\mathbf{R}$ and $\mathbf{A}$ since then $\mathbf{Z} = \mathbf{O}$. Hence $\mathbf{O} = \mathbf{AB}$ and $\mathbf{Z} = \mathbf{BR}$. Consider relation (S5) in this case,

$$J_{\mathbf{O}} = J_{\mathbf{Z}}$$
$$+ \underbrace{I_{\mathbf{A};Y|\mathbf{BR}X\mathbf{S}}}_{(i)} + \underbrace{I_{X;\mathbf{R}|\mathbf{ABS}}}_{(ii)} - \underbrace{I_{\mathbf{R};Y|\mathbf{AB}X\mathbf{S}}}_{(iii)} - \underbrace{I_{X;\mathbf{A}|\mathbf{BRS}}}_{(iv)} . \qquad (S14)$$

Part 1 and 2: Since graphical optimality holds, we know that $J_{\mathbf{O}} = J_{\mathbf{Z}}$ can only be achieved if term (i) = term (iii) and term (ii) = term (iv). From Eq. (S8) we know that term (i) = (iii) can only hold if $I_{\mathbf{A};Y|\mathbf{B}X\mathbf{S}} = 0$. But this implies $\mathbf{A} = \emptyset$ by Faithfulness since, by construction, $\mathbf{A} \subset \mathbf{O}$ is always connected to $Y$ (potentially through $\mathbf{M}$) given $X\mathbf{SO} \setminus \mathbf{A}$. Then term (iv) = 0 and, by optimality, $I_{X;\mathbf{R}|\emptyset\mathbf{BS}} = 0$. But the latter would imply that $\mathbf{Z} = \mathbf{BR}$ is either not minimal anymore since $\mathbf{R}$ is not connected to $X$ and, hence, does not block any non-causal path not already blocked by $\mathbf{B}$. Then $J_{\mathbf{O}} > J_{\mathbf{Z}}$ among all minimal valid $\mathbf{Z}$ (Part 1). Or $\mathbf{Z}$ is minimal and $\mathbf{R} = \emptyset$, for which $\mathbf{Z} = \mathbf{O}$ is the unique set maximizing $J_{\mathbf{Z}}$ among all minimal valid $\mathbf{Z} \neq \mathbf{O}$ (Part 2).

Part 3, i.e., that removing any subset from $\mathbf{O}$ decreases $J_{\mathbf{O}}$ follows directly from setting $\mathbf{R} = \emptyset$ and considering $\mathbf{A} \neq \emptyset$ (since otherwise nothing would be removed). Then term (ii) and term (iii) are both zero and by optimality term (iv), which must be smaller or equal to term (ii), is zero. Since $\mathbf{A}$ is connected to $Y$ (see Part 1) by Faithfulness we have $J_{\mathbf{O}} > J_{\mathbf{O}\setminus\mathbf{A}}$. $\qquad\square$

## C Algorithms

---

**Algorithm C.1** Construction of **O**-set and test for backdoor-identifiability.

---

**Require:** Causal graph $\mathcal{G}$, cause variable $X$, effect variable $Y$, mediators $\mathbf{M}$, conditioned variables $\mathbf{S}$

1: Initialize $\mathbf{P} = \emptyset$, $\mathbf{C} = \emptyset$ and $\mathbf{P_C} = \emptyset$
2: **for** $W \in Y\mathbf{M}$ **do**
3:      $\mathbf{P} = \mathbf{P} \cup pa(W) \setminus \mathbf{forb}$
4: **for** $W \in Y\mathbf{M}$ **do**
5:      Initialize nodes in this level $\mathcal{L} = \{W\}$
6:      Initialize ignorable nodes $\mathcal{N} = \emptyset$
7:      **while** $|\mathcal{L}| > 0$ **do**
8:          Initialize next level $\mathcal{L}' = \emptyset$
9:          **for** $C \in sp(\mathcal{L}) \setminus \mathcal{N}$ **do**
10:              **if** $C = X$ **then**
11:                  **return** No valid backdoor adjustment set exists.
12:              **if** $C \notin \mathbf{C}$ and Def. 3 (1) $C \notin \mathbf{forb}$ and ((2a) $C \in \mathbf{vancs}$ or (2b) $C \perp\!\!\!\perp X \mid \mathbf{vancs}$) **then**
13:                  $\mathbf{C} = \mathbf{C} \cup \{C\}$
14:                  $\mathcal{L}' = \mathcal{L}' \cup \{C\}$
15:              **else**
16:                  **if** $C \notin \mathbf{C}$ **then**
17:                      $\mathcal{N} = \mathcal{N} \cup \{C\}$
18:          $\mathcal{L} = \mathcal{L}' \setminus \mathcal{N}$
19: **for** $C \in \mathbf{C}$ **do**
20:      **if** $X \in pa(C)$ **then**
21:          **return** No valid backdoor adjustment set exists.
22:      $\mathbf{P_C} = \mathbf{P_C} \cup pa(C)$
23: **return** $\mathbf{O} = \mathbf{PCP_C}$

---

**Algorithm C.2** Construction of $\mathbf{O}_{\min}$ and $\mathbf{O}_{\mathrm{Cmin}}$-sets. The relevant code for $\mathbf{O}_{\mathrm{Cmin}}$ is indicated in parentheses.

---

**Require:** Causal graph $\mathcal{G}$, cause variable $X$, effect variable $Y$, mediators $\mathbf{M}$, conditioned variables $\mathbf{S}$, $\mathbf{O} = \mathbf{PCP_C}$-set

1: Initialize $\mathbf{O}_{\min} = \mathbf{O}$ ($\mathbf{C}_{\min} = \mathbf{CP_C} \setminus \mathbf{P}$)
2: **for** $Z \in \mathbf{O}_{\min}$ ($Z \in \mathbf{C}_{\min}$) **do**
3:      **if** $Z$ has no active path to $X$ given $\mathbf{SO} \setminus \{Z\}$ **then**
4:          Mark $Z$ for removal
5: Remove marked nodes from $\mathbf{O}_{\min}$ ($\mathbf{C}_{\min}$)
6: **for** $Z \in \mathbf{O}_{\min}$ ($Z \in \mathbf{C}_{\min}$) **do**
7:      **if** $Z$ has no active path to $Y$ given $X\mathbf{SO}_{\min} \setminus \{Z\}$ (given $X\mathbf{SPC}_{\min} \setminus \{Z\}$) **then**
8:          Mark $Z$ for removal
9: Remove marked nodes from $\mathbf{O}_{\min}$ ($\mathbf{C}_{\min}$)
10: **return** $\mathbf{O}_{\min}$ ($\mathbf{O}_{\mathrm{Cmin}} = \mathbf{PC}_{\min}$)

---

# D Further details and figures of further numerical experiments

## D.1 Setup

We compare the following adjustment sets (see definitions in Section 2.3):

- **O**
- Adjust
- $\mathbf{O}_{\text{Cmin}}$
- $\mathbf{O}_{\text{min}}$
- Adjust$_{\text{Xmin}}$
- Adjust$_{\text{min}}$

To investigate the applicability of different estimators, we use above adjustment sets together with the following estimators from `sklearn` (version 0.24.2) and the `doubleml` (version 0.4.0) package (see instantiated class for parameters):

- Linear ordinary least squares (LinReg) regressor `LinearRegression()`
- $k$-nearest-neighbor (kNN) regressor `KNeighborsRegressor(n_neighbors=3)`
- Multilayer perceptron (MLP) regressor `MLPRegressor(max_iter=2000)`
- Random forest (RF) [Breiman, 2001] regressor `RandomForestRegressor()`
- Double machine learning for partially linear regression models (DML) [Chernozhukov et al., 2018] `DoubleMLPLR(data, ml_g, ml_m)` from `doubleml` with `ml_g=ml_g=MLPRegressor(max_iter=2000)` from `sklearn`

Sklearn [Pedregosa et al., 2011] and `doubleml` [Bach et al., 2021] are both available under an MIT license.

As data generating processes we consider linear and nonlinear experiments generated with the following generalized additive model:

$$V^j = \sum_i c_i f_i(V^i) + \eta^j \quad \text{for} \quad j \in \{1, \ldots, \tilde{N}\}. \tag{S15}$$

To generate a structural causal model among $\tilde{N}$ variables we randomly choose $L$ links whose functional dependencies are linear for linear experiments and one half is $f_i(x) = (1 + 5xe^{-x^2/20})x$ for nonlinear experiments. Coefficients $c_i$ are drawn uniformly from $\pm[0.1, 2]$. For linear experiments we use normal noise $\eta^j \sim \mathcal{N}(0, \sigma^2)$ and, in addition, for nonlinear models $\frac{1}{3}$ of the noise terms is Weibull-distributed, both with standard deviation $\sigma$ drawn uniformly from $[0.5, 2]$. From the $\tilde{N}$ variables of each dataset we randomly choose a fraction $\lambda$ as unobserved and denote the number of observed variables as $N$. For each combination of $N \in \{5, 10, 15, 20\}$, $L \in \{2\tilde{N}, 3\tilde{N}\}$, and $\lambda \in \{30\%, 40\%, 50\%\}$ we randomly create a structural causal model and then randomly pick an observed pair $(X = V^i, Y = V^j)$ connected by a causal path, set $\mathbf{S} = \emptyset$, and consider the intervention $do(V^i = V^i + 1 = x)$ relative to the unperturbed data $(x')$ as ground truth, which corresponds to the linear regression coefficient in the linear case. We further assert that the following criteria hold: (1) the effect is identifiable, (2) the minimal adjustment cardinality is $|\mathbf{vancs}_{\text{min}}(X, Y)| > 0$, and (3) the (absolute) causal effect is $\geq 10^{-3}$ to make sure that Faithfulness holds (if these criteria cannot be fulfilled, another model is generated). We create 500 models for each combination of $N, L, \lambda$. Surprisingly, among in total 12,000 randomly created configurations 93% fulfill the optimality conditions in Thm. 3. This may indicate that also in many real-world scenarios graphical optimality actually holds. Here we do not consider the effect of a selected conditioning variable $\mathbf{S}$ since it would have a similar effect on all methods considered.

For the considered graphs the computation time to construct adjustment sets is very short and arguably negligible to the actual cost of fitting methods that use these adjustment sets. The results were evaluated on Intel Xeon Platinum 8260.

## D.2 Figures for linear least squares estimator

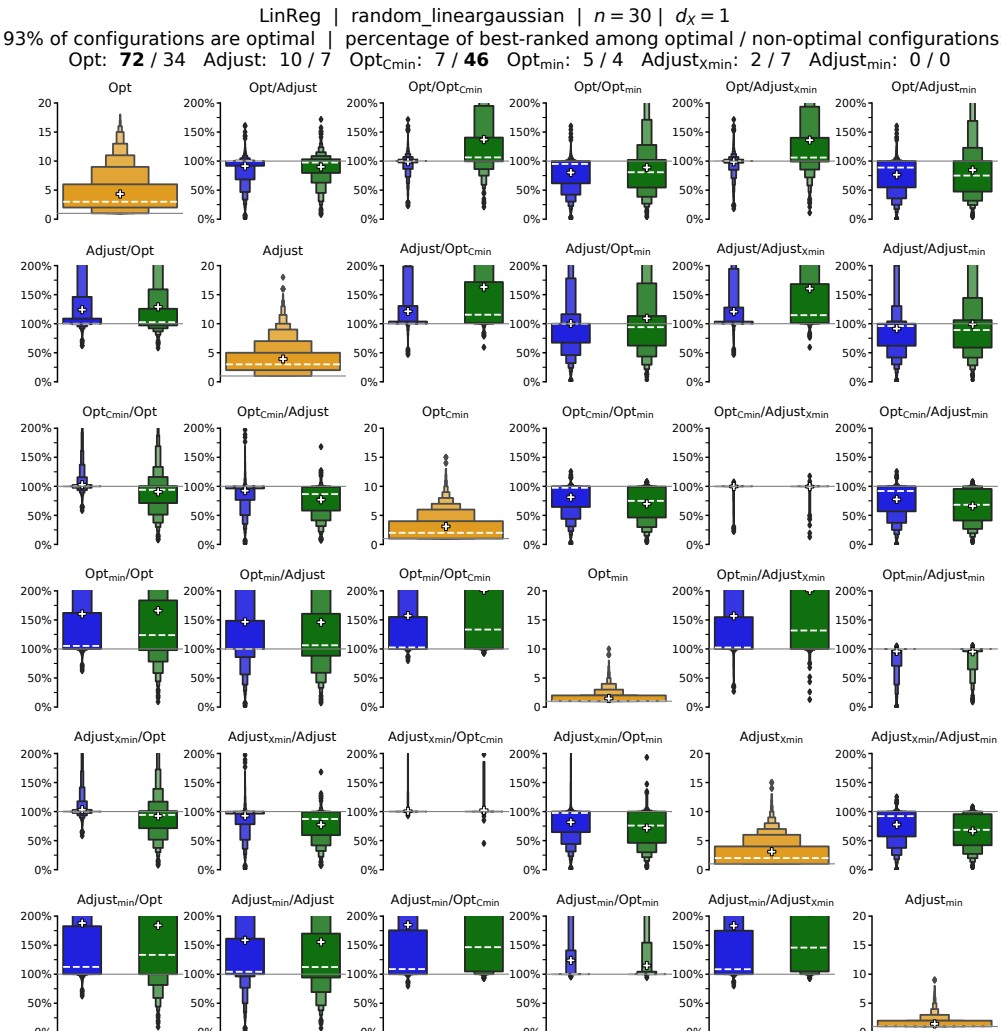

Figure S2: Results of linear experiments with linear estimator and sample size $n = 30$. The diagonal depicts letter-value plots [Hofmann et al., 2017] of adjustment set cardinalities and the off-diagonal shows pairs of RMSE ratios for all combinations of (**O**, Adjust, $\mathbf{O}_{\mathrm{Cmin}}$, $\mathbf{O}_{\mathrm{min}}$, Adjust$_{\mathrm{Xmin}}$, Adjust$_{\mathrm{min}}$) for optimal configurations (left in blue) and non-optimal configurations (right in green). Values above 200% are not shown. The dashed horizontal line denotes the median of the RMSE ratios, and the white plus their average. The letter-value plots are interpreted as follows: The largest box shows the 25%–75% range. The next smaller box above (below) shows the 75%–87.5% (12.5%–25%) range and so forth. The numbers on best-ranked methods at the top indicate the percentage of the 12,000 randomly created configurations where the method had the lowest variance. The highest percentage is marked in bold. Note that the highest ranked method may outperform others only by a small margin. The results in the letter-value plots provide a more quantitative picture. See also Fig. S7 where the ranks are further distinguished by the **O**-set cardinality.

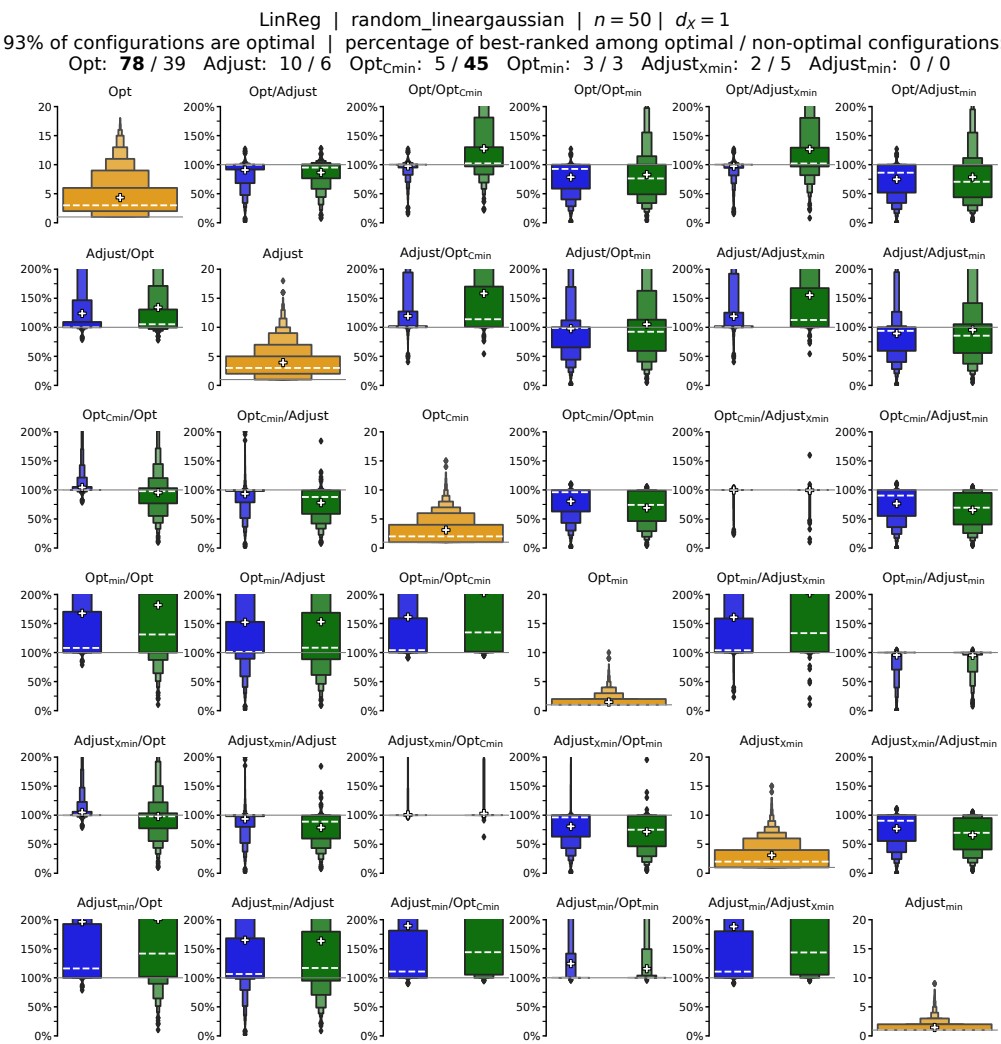

Figure S3: As in Fig. S2 but for $n = 50$.

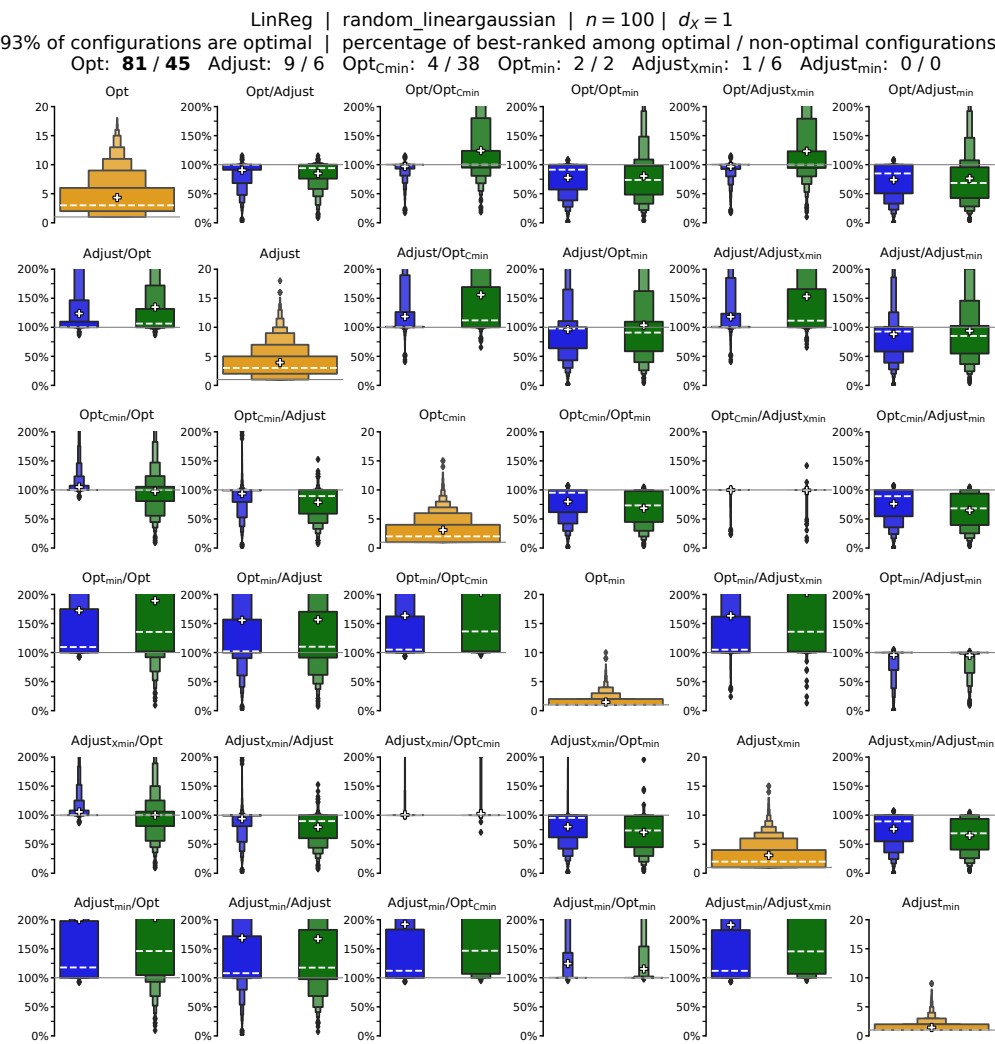

Figure S4: As in Fig. S2 but for $n = 100$.

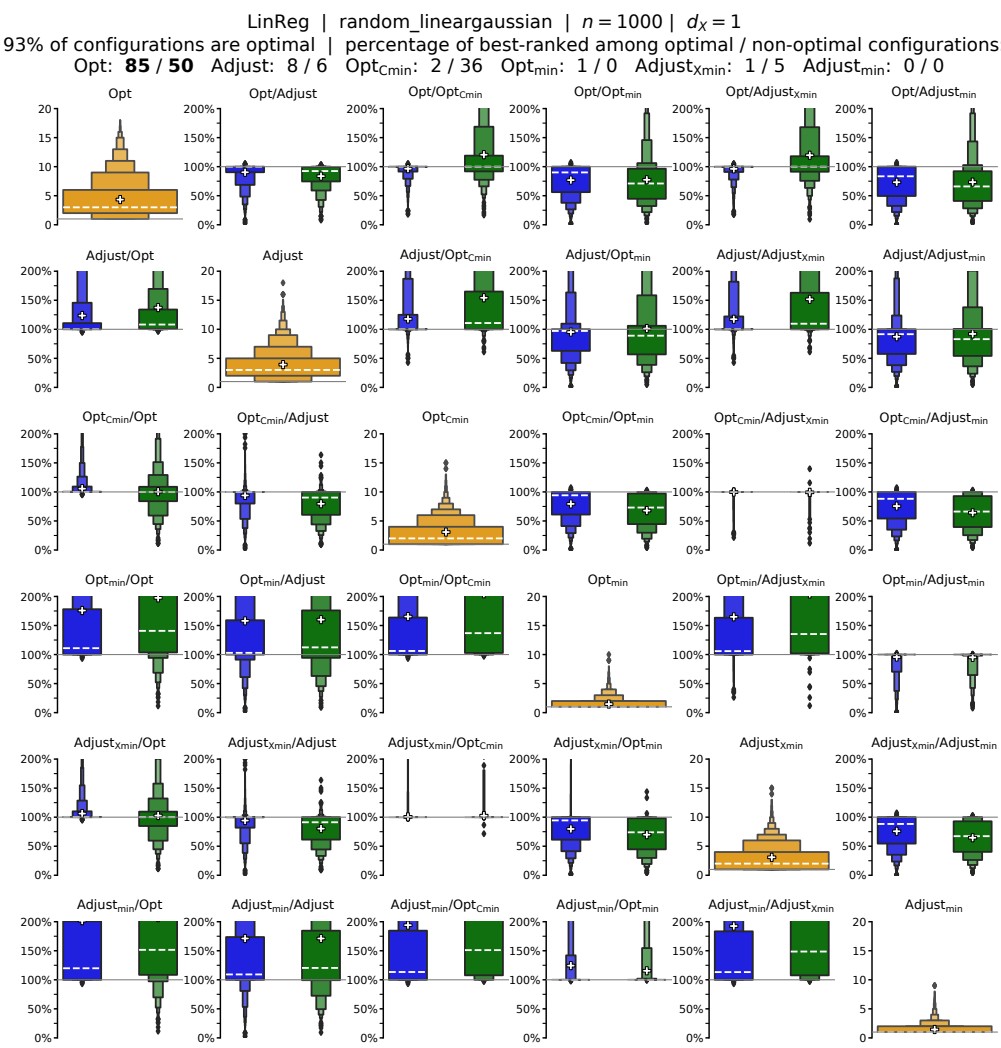

Figure S5: As in Fig. S2 but for $n = 1000$.

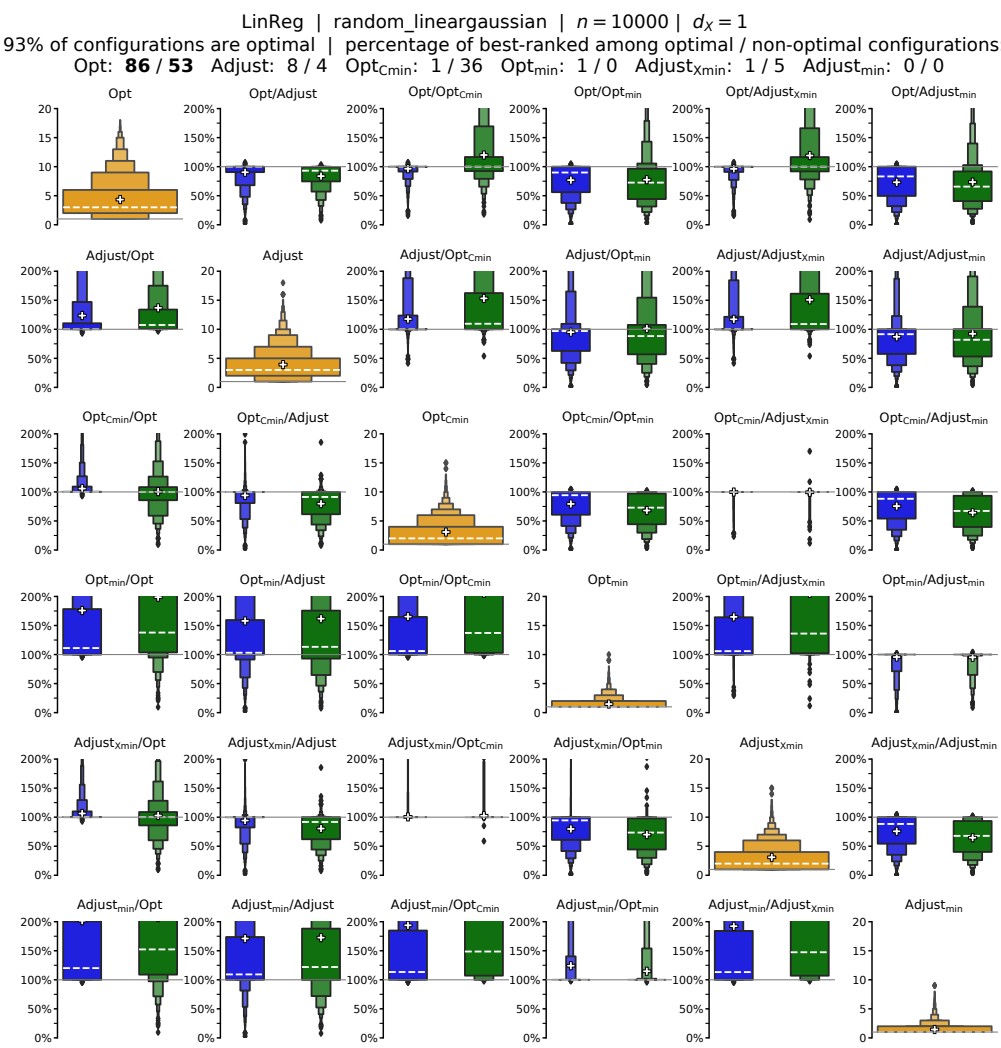

Figure S6: As in Fig. S2 but for $n = 10000$.

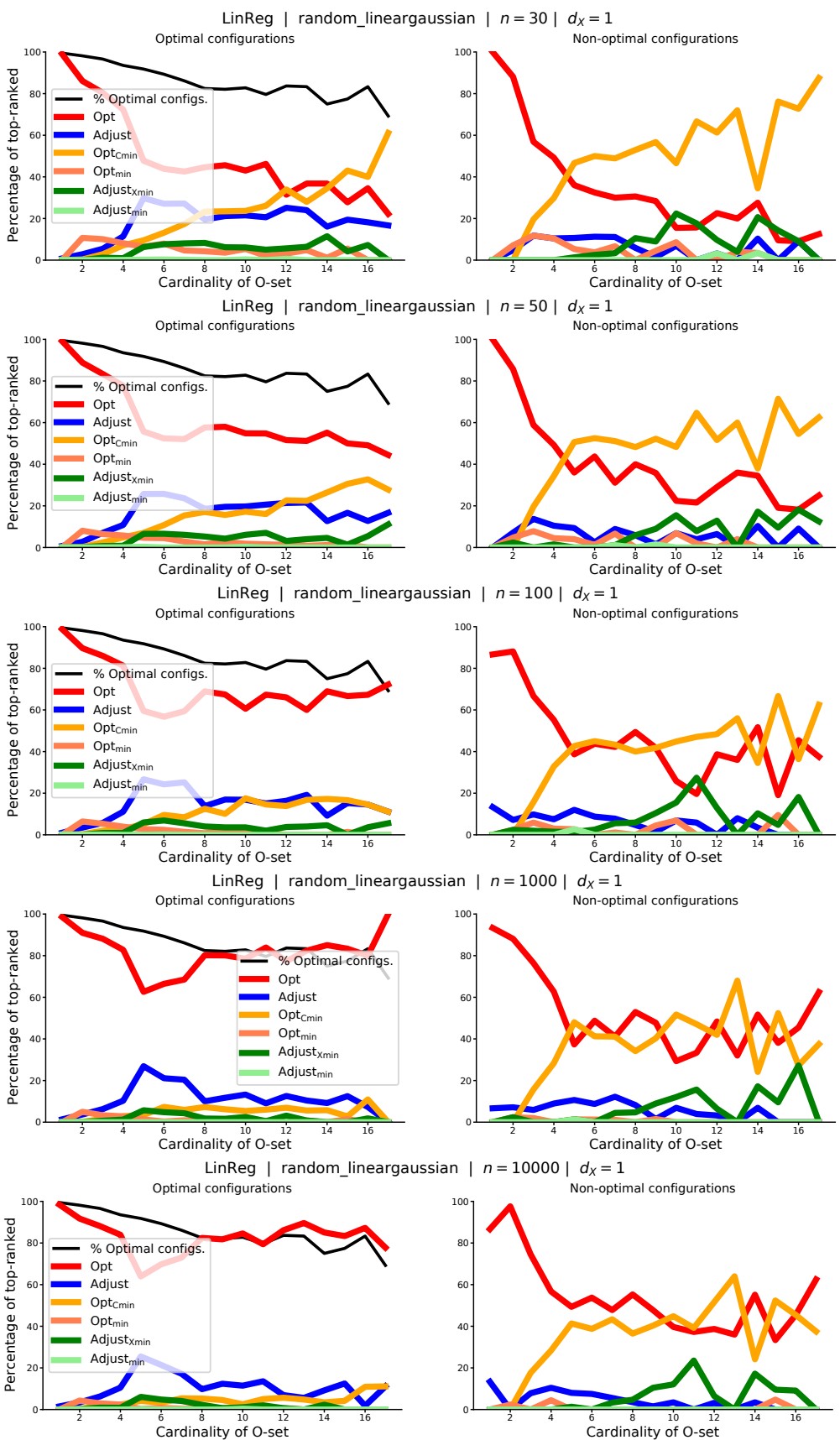

Figure S7: Percentage of configurations where each method has the lowest variance for linear experiments, stratified by the cardinality of the **O**-set ($x$-axis) for $n = 30$ (top) to $n = 10,000$ (bottom).

## D.3    Figures for non-parametric estimators

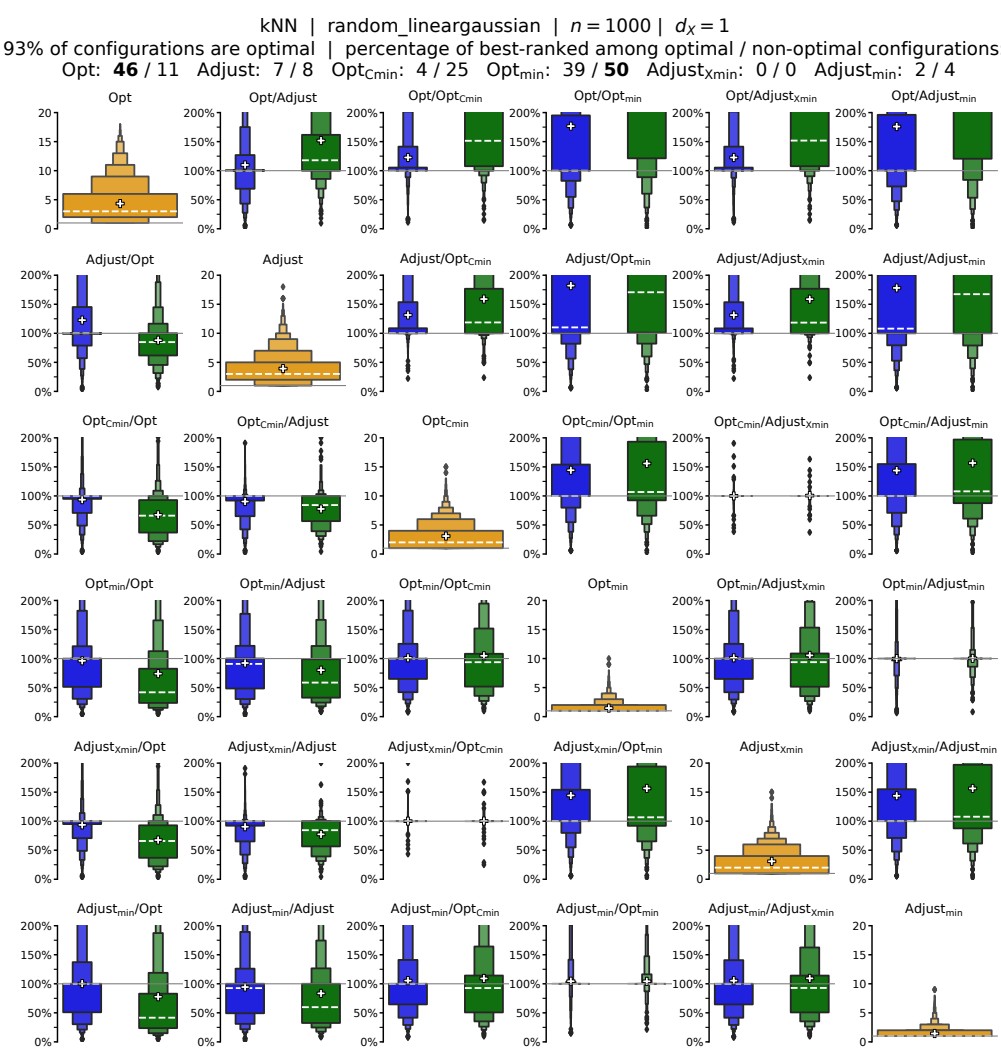

Figure S8: As in Fig. S2 but with kNN estimator ($k = 3$) and $n = 1000$. See also Figs. S15,S16 where the ranks are further distinguished by the **O**-set cardinality.

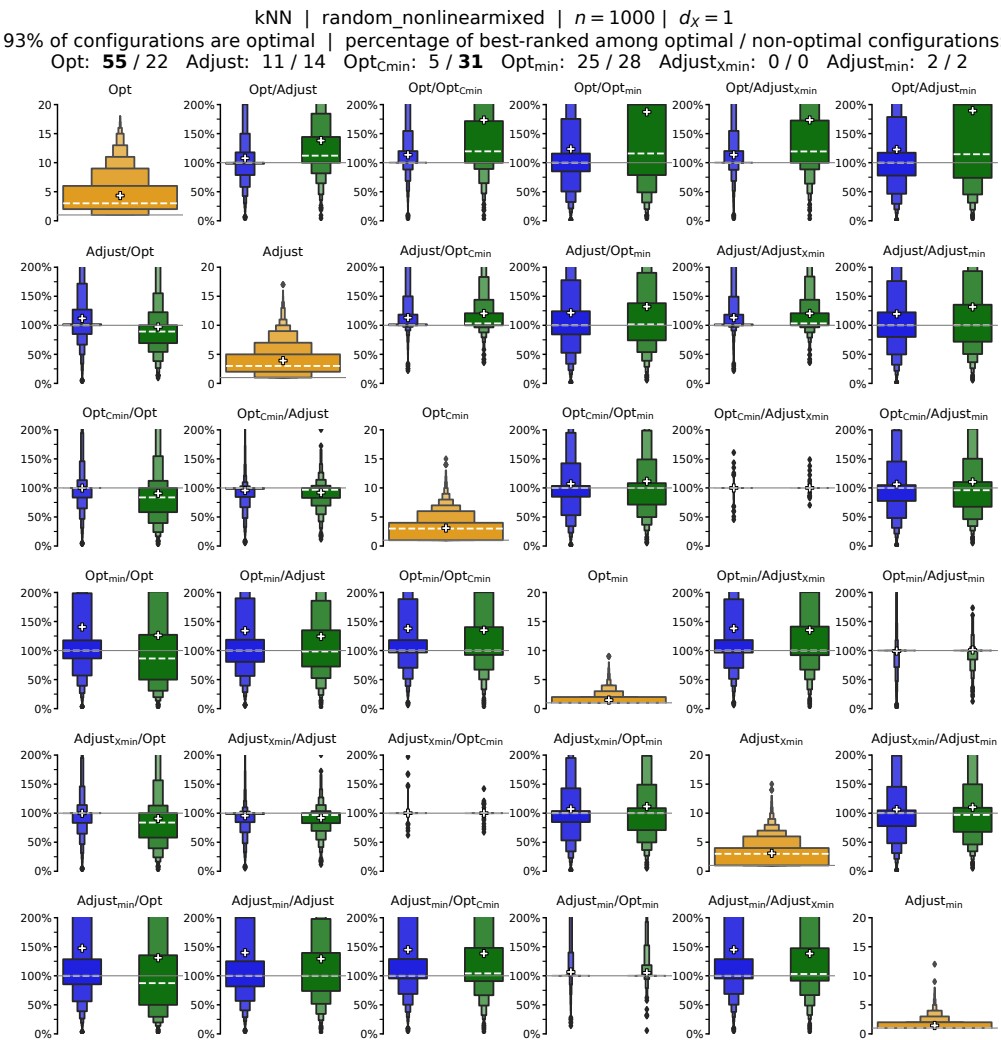

Figure S9: As in Fig. S2 but for kNN estimator ($k = 3$), the nonlinear model, and $n = 1000$.

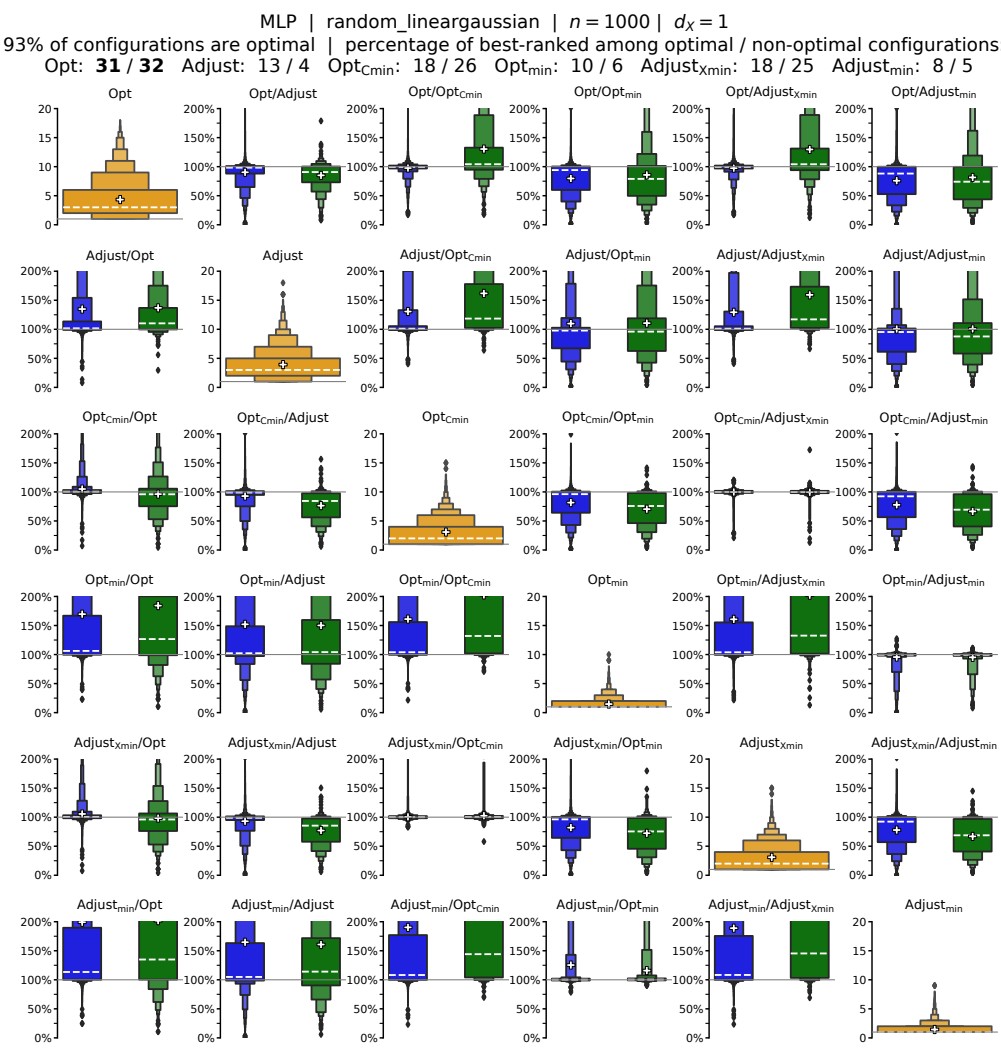

Figure S10: As in Fig. S2 but for MLP estimator and $n = 1000$.

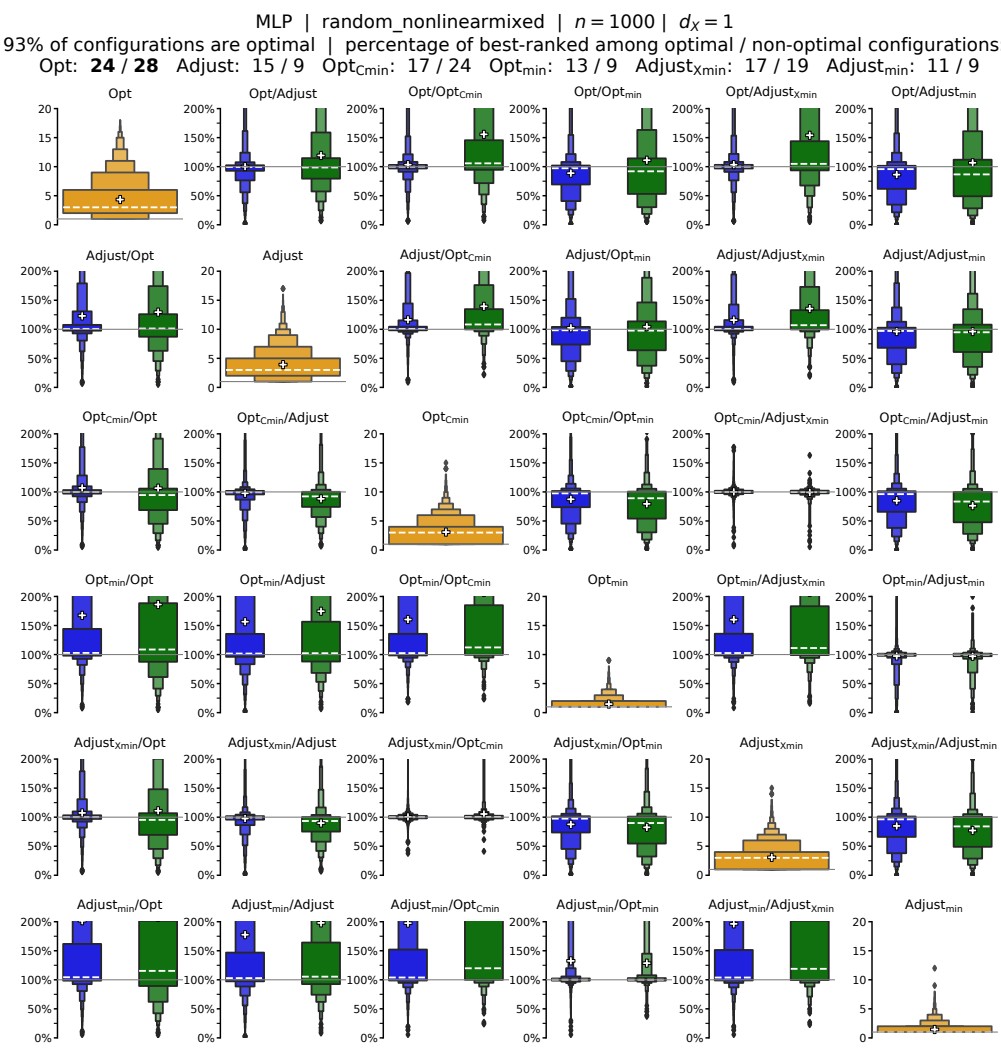

Figure S11: As in Fig. S2 but for MLP estimator, the nonlinear model, and $n = 1000$.

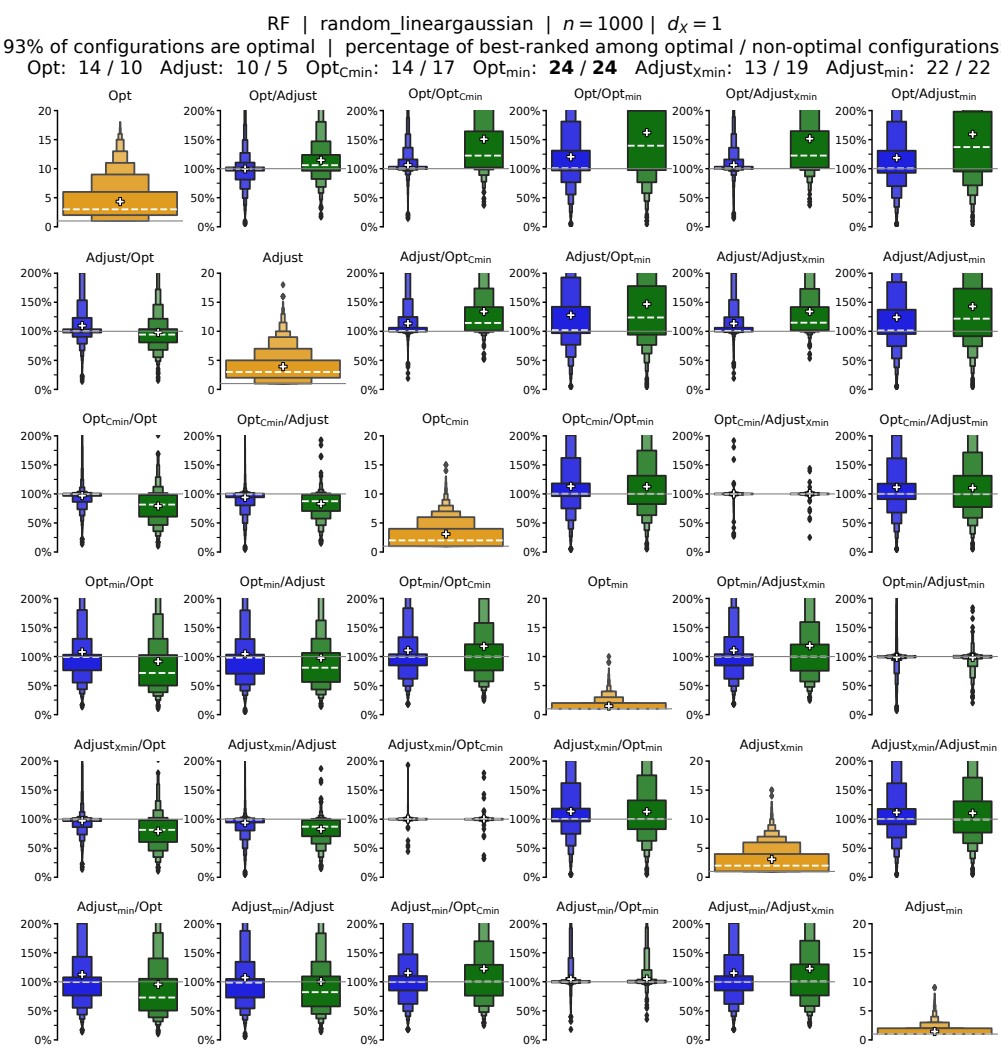

Figure S12: As in Fig. S2 but for RF estimator and $n = 1000$.

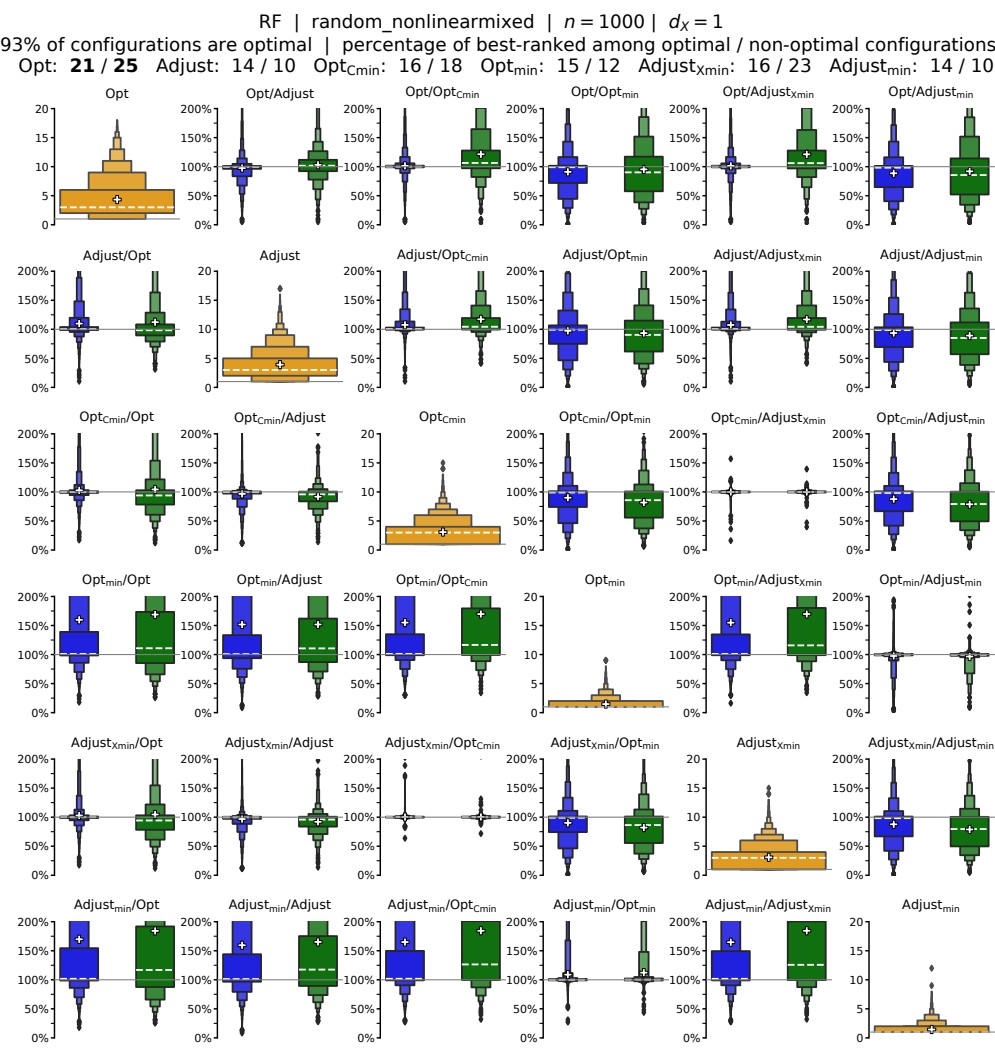

Figure S13: As in Fig. S2 but for RF estimator, the nonlinear model, and $n = 1000$.

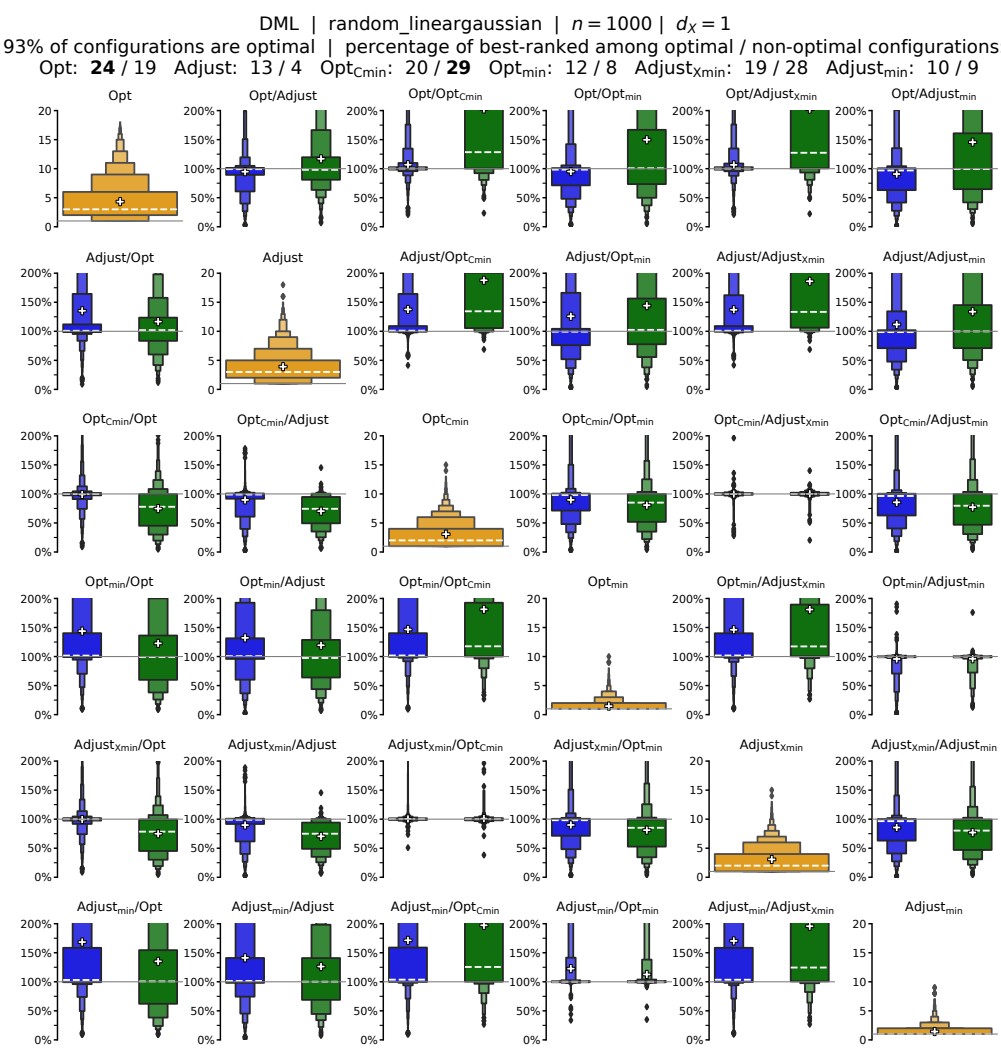

Figure S14: As in Fig. S2 but for DML estimator and $n = 1000$.

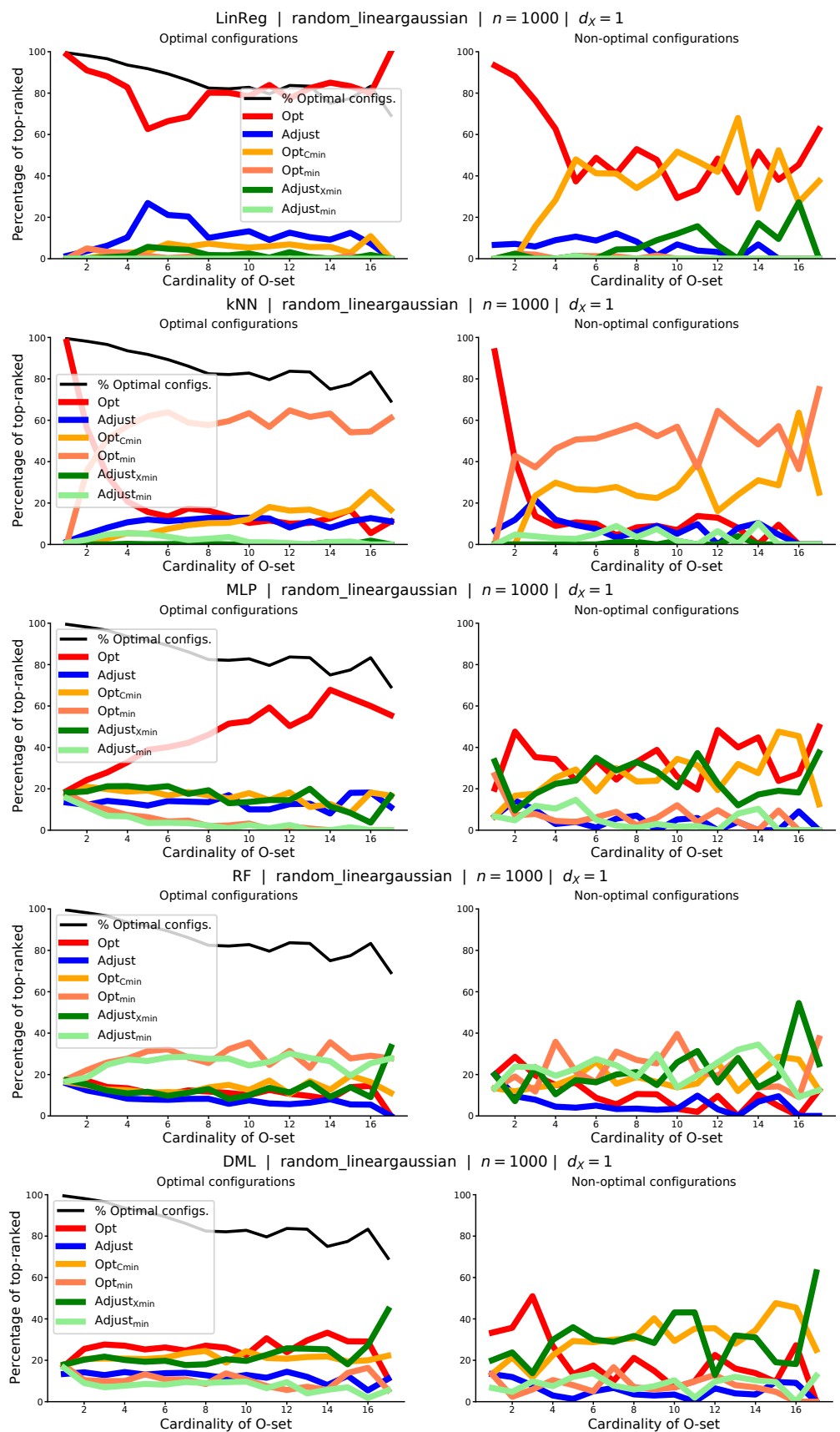

Figure S15: As in Fig. S7 but including non-parametric estimators for $n = 1000$.

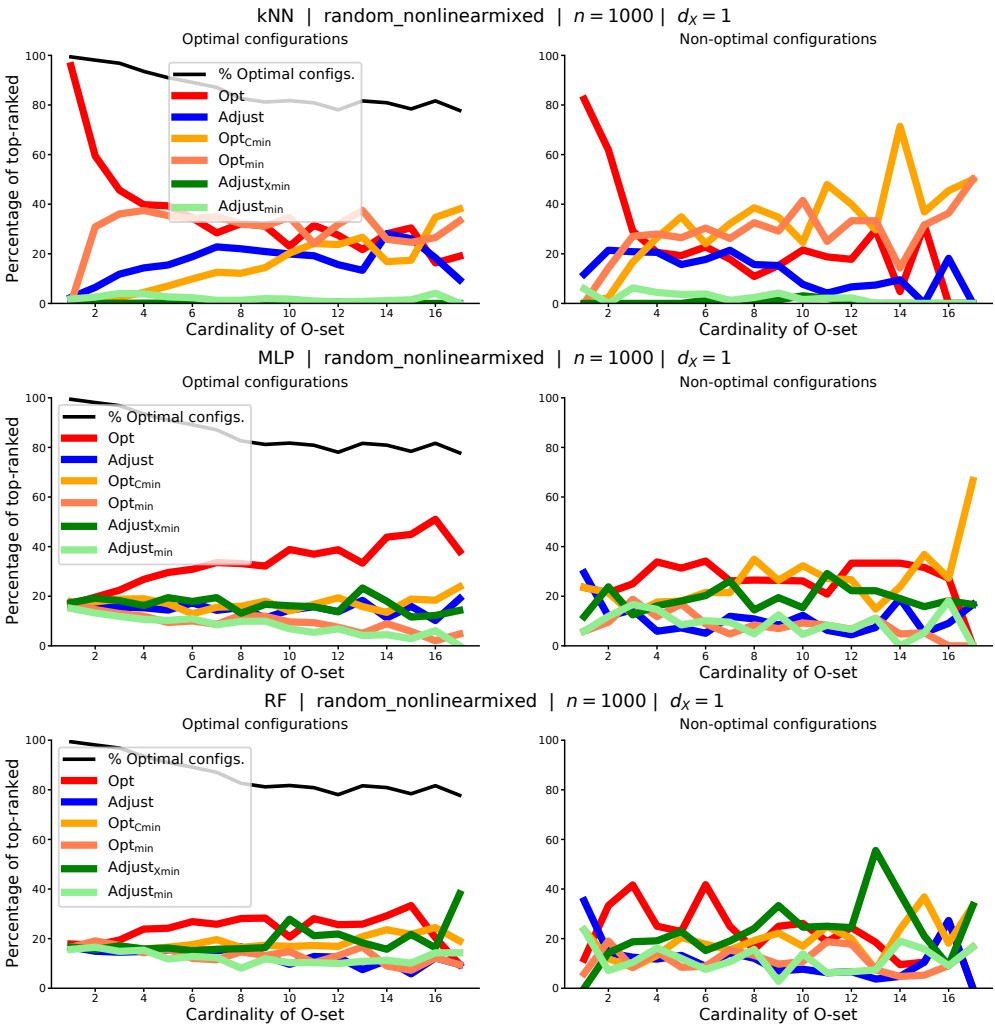

Figure S16: As in Fig. S15 but for nonlinear experiments.