# OpenReview forum: "Necessary and sufficient graphical conditions for optimal adjustment sets in causal graphical models with hidden variables"
_NeurIPS.cc/2021/Conference — NeurIPS 2021 Spotlight_

### Official Review · Reviewer_rosK · 2021-06-26

**Rating:** 7
**Confidence:** 2

**Summary:**

In this paper, the authors propose a criterion to select the optimal adjustment set. The authors propose a criterion (Eq. 4 and Def. 2) based on information theory to select which adjustment set is best. And they build the connections between this criterion to the common used minimized asymptotic variance (Eq. 7) for linear Gaussian causal model. Based on these, they propose a necessary and sufficient criterion for the existence of an optimal set. However, it is inefficient due to the large number of comparisons. Hence, the authors present an algorithm to find the o-set, which could be as the optimal set under some conditions as Thm. 3.

**Limitations And Societal Impact:**

Yes.

**Main Review:**

After rebuttal: The authors have addressed my questions. I think this paper tackles an interesting problem. I am happy to see it accepted.

-------------------------
Although I try my best to read this paper, it is hard for me to understand the details. Hence if some parts I summarized in the ``Summary'' part are inconsistent to the paper, I will be grateful if the authors could point them out.

The proposed method is technically solid. And the experimental results are convincing. My main concern is about the motivation to provide the criterion that does not depend on the distribution. As shown by ``Our goal is to provide graphical criteria for optimal adjustment sets, i.e., criteria that depend only on the structure of the graph G and not on the distribution'' on Line 143. I do not think this target is well motivated. In reality, we often have observational data. In that case, the distribution is known (although it is estimated) and which set is optimal for example in Example F seems not to be agnostic. Hence I look forward to an illustration for the motivation to provide the criterion that does not depend on the distribution.

I am curious about the computational complexity of Lemma 2. It seems that this step will not cost so much because there are usually not so many valid adjustment sets. If it does not cost so much in reality, it is feasible to directly find the set by Lemma 2. What's more, if we know the (estimated) distribution of all the observed variables, we could just find the optimal st by Lemma. It seems to be more sensible than the o-set.

It is a bit puzzling by the sentence ``Z that maximally constrains the effect variable Y and minimally constrains the cause variable X'' on Line 110 implies. Could the authors please provide a detailed illustration? What does the constrain here mean?


A suggestion:
I understand the equation on line 124 as f(H_{Y|X,Z,S}-H_{X|Z,S})=\frac{1}{\sqrt{n}}e^{H_{Y|X,Z,S}-H_{X|Z,S}}, is it right? The format right now is not quite strict.

**Time Spent Reviewing:**

6

---

> ### Author Response · Authors · 2021-08-09
> **Response to review**
>
> Dear Reviewer rosK,
>
> thank you very much for your review which will help to improve the paper! Below I clarify your raised points:
>
> * "Although I try my best to read this paper, it is hard for me to understand the details. Hence if some parts I summarized in the ``Summary'' part are inconsistent to the paper, I will be grateful if the authors could point them out."
>
> Your summary is correct! To elaborate more on Thm. 3: One of the main contributions of our work is Thm 3 which states the exact *necessary* and *sufficient* conditions that characterize which graphs admit optimal sets and which don't. For graphs where an optimal set exists at all, the O-set is optimal. Our experiments indicate that graphical optimality actually holds in more than 80% of the randomly created graphs indicating that this might not be just a "special case".
>
>
> * "My main concern is about the motivation to provide the criterion that does not depend on the distribution. ... In reality, we often have observational data. In that case, the distribution is known (although it is estimated) and which set is optimal for example in Example F seems not to be agnostic."
>
> The scope of our work is indeed on a purely graphical criterion that will hold for *any* distribution. This means that for graphs where graphical optimality holds (Thm 3) it (asymptotically) does not make sense to use the distribution in choosing the best adjustment set. Note that this is an asymptotic result, but our numerical experiments indicate that it holds also for very small sample sizes (50 for adjustment set cardinalities of up to ~20). However, as you suggest, for graphs where Thm 3 says that graphical optimality does *not* hold there may be non-graphical distribution-dependent criteria for an optimal set. We have done some preliminary numerical investigations on approaches that use the oberved distribution, e.g., adding/removing variables to the adjustment set based on AIC/BIC, but these didn't turn out to be useful. We will mention your point to further make use of the distribution in the Discussion section.
>
>
> * "Hence I look forward to an illustration for the motivation to provide the criterion that does not depend on the distribution."
>
> As discussed in the point just above, for the cases covered in Thm 3 the optimality (asymptotically) holds for *any* distribution. We will add in the discussion that a graphical criterion has the advantage that it does not require estimating properties of the observed distribution which will have associated errors and may lead to sub-optimal estimators.
>
>
> * "I am curious about the computational complexity of Lemma 2. It seems that this step will not cost so much because there are usually not so many valid adjustment sets. If it does not cost so much in reality, it is feasible to directly find the set by Lemma 2."
>
> The complexity of all possible valid sets *does* cost a lot! This is because there are typically many valid sets. They are only required to block confounding paths and variables that are descendants of YM are excluded. A confounding path can be blocked in many ways. For example, if there are 5 variables on a confounding path, then it can be blocked by 2^5-1 different subsets. We've tried iterating through all sets computationally using the algorithms by van der Zander et al and it was not feasible for relevant graph sizes.
>
>
> * "What's more, if we know the (estimated) distribution of all the observed variables, we could just find the optimal st by Lemma. It seems to be more sensible than the o-set."
>
> Thm. 3 not only avoids this computationally infeasible search to compare all Z's with each other, but also decides when this search makes sense at all. Namely, if no graphical criterion exists. Then one can indeed exclude some adjustment sets that are stricly worse than others. But there won't be a single set that is optimal. We will add this discussion in the text.
>
>
> * "It is a bit puzzling by the sentence ``Z that maximally constrains the effect variable Y and minimally constrains the cause variable X'' on Line 110 implies. Could the authors please provide a detailed illustration? What does the constrain here mean?"
>
> Adjustment information is defined as (ignoring the S for simplicity)
>
> J_Z ≡ I_Z;Y|X − I_X;Z
>
> The common intuition requires to maximally constrain the effect variable and minimally constrain the cause variable. If Z contains a lot of information about the effect variable Y, i.e., Z well constrains Y, then I_Z;Y|X is large. Further, if Z minimally constrains the cause variable X, then I_X;Z is small. Hence, if I_Z;Y|X is large and I_X;Z is small, then the adjustment information J_Z is large. Eq (5) relates J_Z to the adjustment entropy which by Eq. (7) is directly related to the asymptotic estimator variance.
>
>
> * "A suggestion: I understand the equation on line 124 as f(H_{Y|X,Z,S}-H_{X|Z,S})=\frac{1}{\sqrt{n}}e^{H_{Y|X,Z,S}-H_{X|Z,S}}, is it right? The format right now is not quite strict."
>
> We wanted to keep the assumption in Eq. (7) as flexible as possible: f(.) must only be a strictly monotonously increasing function. For linear Gaussians this becomes the expression in line 128, but further research might show that for other estimators such a strictly monotonously increasing function takes a different form. Then these results can use the general information-theoretic results presented in this work.
>
>
> Thank you for these remarks, I will include them in the revision which will help to improve the paper. I am happy to further respond if anything is still unclear.
>
> Kind Regards
> The authors

---

> > ### Comment · Reviewer_rosK · 2021-08-18
> > **Response to authors**
> >
> > Thank you very much for the thorough response! It addresses my questions. I increase my score and am happy to see it accepted.
> >
> > Best Regards

---

### Official Review · Reviewer_wcA7 · 2021-07-15

**Rating:** 7
**Confidence:** 4

**Summary:**

The problem of selecting optimal backdoor adjustment sets to estimate causal effects in graphical models with hidden and conditioned variables is addressed.

**Limitations And Societal Impact:**

* As written above, I'd like to recommend to write the linear SCM assumption (line 125-126) more explicitly.
* Does this result hold for the singleton X,Y?

**Main Review:**

This paper solves the problem of selecting the optimal backdoor adjustment sets by leveraging the theory of information theories. This paper is motivated by the interesting observation that the optimal admissible set is the set providing the most information (adjustment information, Def. 1).

However, I have to confess that I couldn't capture the main claim of the paper. Does this paper answer that O-set in Def. 4 is the optimal backdoor admissible set whenever there exists a valid admissible set? It seems that the main claim in Thm.3 only guarantees the optimality of O-set for some special cases.

Despite its interesting and novel approaches, I also had to agree that it is hard to understand this paper. I will discuss those in detail in **Readability** paragraphs. Here are some main questions regarding the contribution/novelty of this work:

- I am not sure about the distinction between SSR20 and the result of this paper. Up to my understanding, Thm 2 of SSR20 provides an (statistically-) optimal adjustment set for the nonparametric setting, and the only condition is the existence of the back-door admissible set. That is, I don't understand which particular points have never been addressed by SSR20.
- In Eq. (3), why do you consider the conditioned variables S? Why did you prevent the possibility of S being arbitrary? (what's the role of the assumption S \cap des(X) = \emptyset in Line 87?)
- Line 109-110, why is this a valid intuition? That is, what would be the specific benefits of invoking the information theory?


## Readability
Here are comments discussing which particular parts I felt difficult to understand. I also added my comments and recommendations to help to improve comprehensibility of the paper.

- In general, it is not desirable to put the essential contents in the Appendix, because readers must go back and forth to parse the main and appendix at the same time. For example, it's hard for me to understand Thm. 3 and Line 78.
- Please write the assumption more explicitly. I think one of the crucial assumption is the linear SCM setting (where all the variables are in the linear relation), as you wrote in line 125-126.
- I'd like to add some overview or guideline for every starts of the section, Definitions, Theorems, and Lemma. For example, I couldn't guess what authors will discuss in Section 2.2 even after reading the first paragraph.
- In Line 62-63, I'd recommend to give a detailed definition of ADMG, because readers might not want to look for the definition of ADMG from other sources. Without the definition, it's hard to parse the graph in Fig. 1; For example, what does the bidirected edge mean?
- In Line 62-70, what is the definition of An(X), De(X)? Do they include the variable X? Please write it more explicitly.
- In Line 62-70, I am not sure why you wanted to introduce the notion of MAG, given that this notion has never been used. Since MAG, PAG, Markov equivalence, "visible edge" are difficult concepts, I recommend to drop those in your work unless it's essential.
- In Section 1.1, {X,Y} are singleton? Otherwise, I recommend to use the bold letters.
- In Figure 1, what is the meaning of "-" and "+"?
- I think the question in Line 106 is misleading and unclear what the research question is. Why did you give vancs (in line 105) before giving the research question? It distracts readers who want to focus on the research question. Also, the goal is supposed to minimize **asymptotic** estimation variance, but the word "asymptotic" is absent. Finally, since the line 106 is the main statement giving the research question, I think this deserves more than a line. Please highlight it more explicitly, because, as a reader, this is the very sentence that I want to know from Section 1.
- In Line 122 and Eq. (7), I think this term is a mean-square error, not asymptotic variance.
- I'd recommend to move line 125-126 after Assumption 2, given that this gives a concrete picture what you want to study within the assumption 2. Without concrete examples for Assumption 2, it's hard to parse this paper.
- I don't parse Def. 2 (what is the meaning of "no other Z' or for all other Z'"?). Does it mean that Z is optimal if Jz is maximized among all Z' that is a valid adjustment set?
- It's hard to parse Def. 4, because there is no definition for the notation that you used for defining C.
- It seems that Thm 3 provides special cases when O-set is the optimal. It seems that there must be the optimal set even when the conditions in Thm. 3 fails (for sure, O-set is not the optimal one in this case). Can we have a result that we always can have a O-set whenever there are valid back-door admissible set?

**Time Spent Reviewing:**

4 hours

---

> ### Author Response · Authors · 2021-08-09
> **Response to review**
>
> Dear Reviewer wcA7,
>
> thank you very much for your in-depth review and the many comments that I will try to answer thoroughly. First, your main questions:
>
> * "Does this paper answer that O-set in Def. 4 is the optimal backdoor admissible set whenever there exists a valid admissible set? It seems that the main claim in Thm.3 only guarantees the optimality of O-set for some special cases."
>
> It was found already by Henckel et al. 2019 that there exist graphs G where *no* single adjustment set is optimal for *all* possible distributions that are compatible with G. Examples are Figs. 2E,F. In these cases, one simply *cannot* read-off any adjustment set that is optimal for any distribution. Of course, this also holds for the O-set. One of the main contributions of our work is Thm 3 which states the exact *necessary* and *sufficient* conditions that characterize which graphs admit optimal sets and which don't. For graphs where an optimal set exists at all, the O-set is optimal. Our experiments indicate that graphical optimality actually holds in more than 80% of the randomly created graphs indicating that this might not just be a "special case".
>
>
> * "I am not sure about the distinction between SSR20 and the result of this paper. Up to my understanding, Thm 2 of SSR20 provides an (statistically-) optimal adjustment set for the nonparametric setting, and the only condition is the existence of the back-door admissible set. That is, I don't understand which particular points have never been addressed by SSR20."
>
> SSR20 only provide a *sufficient* condition for graphical optimality and an adjustment set for cases covered by their criterion. Our main contribution is a *necessary* and *sufficient* criterion, i.e., a criterion that covers *all* cases. Furthermore, SSR20's sufficient criterion is very restrictive and, for example, does not apply to any of the examples shown in Fig. 2 (except for Example G due to the added conditioning set S).
>
>
> * "In Eq. (3), why do you consider the conditioned variables S? Why did you prevent the possibility of S being arbitrary? (what's the role of the assumption S \cap des(X) = \emptyset in Line 87?)"
>
> The case with a fixed conditioning set S was treated in SSR20 and we wanted to also cover this case. Fixing some conditions implies the estimation of a *conditional* causal effect. S is not arbitray since S \cap des(MY) = \emptyset is needed because otherwise any adjustment set is invalid. On second thought, the additional assumption S \cap des(X) = \emptyset is not strictly needed for the results in this paper and we will clarify this in the revised version. Thank you very much!
>
>
> * "Line 109-110, why is this a valid intuition? That is, what would be the specific benefits of invoking the information theory?"
>
> The validity of this intuition comes from our theoretical result in Lemma 1: Eq (5) relates the adjustment information J_Z to the adjustment entropy which by Eq. (7) is directly related to the asymptotic estimator variance. The information-theoretic characterization then allows to make use of the many nice theoretical properties of conditional mutual information that facilitate relatively short and concise proofs.
>
>
> Regarding your points on readability:
>
> * Stating assumptions in Thm. 3, Line 78 and the linear SCM setting more explicitly:
>
> Thank you very much for this suggestion. We will use the additional content page for the camera-ready version to make assumptions more explicit.
>
>
> * Adding an overview or guideline for every starts of the section, Definitions, Theorems, and Lemma... a detailed definition of ADMG.
>
> We will also use the additional space for overviews and a detailed definition of ADMGs. A bidirected edge indicates hidden confounding.
>
>
> * "In Line 62-70, what is the definition of An(X), De(X)? Do they include the variable X? Please write it more explicitly."
>
> They denote ancestors and descendants and yes, they include X. This is currently only explained in the Supplementary Section A.1 on graph terminology.
>
>
> * "In Line 62-70, I am not sure why you wanted to introduce the notion of MAG, given that this notion has never been used. Since MAG, PAG, Markov equivalence, "visible edge" are difficult concepts, I recommend to drop those in your work unless it's essential."
>
> I've seen quite some work on MAGs and PAGs in the literature on latent causal discovery and hope that these become more widely applied. But you're right, they are difficult concepts. We deem it important to also cover these cases here, but will separate them in the text.
>
>
> * "In Section 1.1, {X,Y} are singleton? Otherwise, I recommend to use the bold letters."
>
> They are assumed singleton here since Eq(7) only holds for the asymptotic variance in the singleton case. However, all information-theoretic results hold also for multivariate X,Y. Further research might be able to also find a relation between J_Z and multivariate X,Y. This is not straightforward because there is no simple expression for the  variance of the sum of multivariate regression coefficients (given Z).
>
>
> * "In Figure 1, what is the meaning of "-" and "+"?"
>
> They are intended to stand for "minimizing" and "maximizing", respectively. Will be clarified.
>
>
> * "I think the question in Line 106 is misleading and unclear what the research question is."
>
> Indeed this is a bit misleading. We mean that the research question is which of the six valid back-door adjustment sets mentioned just before is optimal here.
>
>
> * "Why did you give vancs (in line 105) before giving the research question? It distracts readers who want to focus on the research question."
>
> Vancs is defined before to illustrate the concept of validity since it is valid if and only a valid set exists. But we will make the research question more explicit before.
>
>
> * "Also, the goal is supposed to minimize asymptotic estimation variance, but the word "asymptotic" is absent."
>
> Indeed, in that sentence we forgot the word "asymptotic", we will fix it and add it also in some other sentences where it was left out!
>
>
> * "Finally, since the line 106 is the main statement giving the research question, I think this deserves more than a line. Please highlight it more explicitly, because, as a reader, this is the very sentence that I want to know from Section 1."
>
> We will make it more explicit. Note that that main research question is also stated in the introduction.
>
>
> * "In Line 122 and Eq. (7), I think this term is a mean-square error, not asymptotic variance."
>
> Since the estimator is assumed unbiased (line 120), mean-square error is equal to asymptotic variance.
>
>
> * "I'd recommend to move line 125-126 after Assumption 2, given that this gives a concrete picture what you want to study within the assumption 2. Without concrete examples for Assumption 2, it's hard to parse this paper."
>
> Good idea!
>
>
> * "I don't parse Def. 2 (what is the meaning of "no other Z' or for all other Z'"?). Does it mean that Z is optimal if Jz is maximized among all Z' that is a valid adjustment set?"
>
> Yes!
>
>
> * "It's hard to parse Def. 4, because there is no definition for the notation that you used for defining C."
>
> The definition of valid collider paths is given above in Def. 3. We will make this more clear.
>
>
> * "It seems that Thm 3 provides special cases when O-set is the optimal. It seems that there must be the optimal set even when the conditions in Thm. 3 fails (for sure, O-set is not the optimal one in this case). Can we have a result that we always can have a O-set whenever there are valid back-door admissible set?"
>
> It was found already by Henckel et al. 2019 that there exist graphs G where *no* single adjustment set is optimal for *all* possible distributions that are compatible with G. Examples are Figs. 2E,F. In these cases, one simply *cannot* read-off any adjustment set that is optimal for any distribution. Of course, this also holds for the O-set. One of the main contributions of our work is Thm 3 which states the exact *necessary* and *sufficient* conditions that characterize which graphs admit optimal sets and which don't. For graphs where an optimal set exists at all, the O-set is optimal. Our experiments indicate that graphical optimality actually holds in more than 80% of the randomly created graphs indicating that this might not be just a "special case".
>
>
> * "As written above, I'd like to recommend to write the linear SCM assumption (line 125-126) more explicitly."
>
> Will be done!
>
>
> * "Does this result hold for the singleton X,Y?"
>
> Yes, see my comment to the same point above.
>
> Thank you for these remarks, I will include them in the revision which will help to improve the paper. I am happy to further respond if anything is still unclear.
>
> Kind Regards
> The authors

---

> > ### Comment · Reviewer_wcA7 · 2021-08-18
> > **Response to Authors**
> >
> > Thank you very much for the detailed response and the hard work! I sincerely appreciate for that. I read the paper again, and checked my review and the responses. The response answers what I wanted to know very clearly. Most of all, I understood the distinction between SSR20 and this paper, now. I agree the significance of the main result of the paper. I will raise my score and vote for acceptance.

---

> > > ### Author Response · Authors · 2021-09-17
> > > **Remark to meta-reviewer / reviewer**
> > >
> > > Dear Meta-Reviewer and Reviewer wcA7,
> > >
> > > I thought again about the comment to remove the parts on MAGs and focus on the more standard ADMGs in the paper. You wrote: " Since MAG, PAG, Markov equivalence, "visible edge" are difficult concepts, I recommend to drop those in your work unless it's essential."
> > >
> > > I would now agree to drop these from the paper. It will make it more focused and leave more space to address the other points. I just wanted to mention that here and hope that the meta-reviewer also agrees.
> > >
> > > Kind Regards
> > > The authors

---

### Official Review · Reviewer_9L1v · 2021-07-16

**Rating:** 7
**Confidence:** 3

**Summary:**

The paper defines the optimality of a (valid) adjustment set in terms of a notion called 'adjustment information.'  Then, it focuses on estimators for which maximizing adjustment information equates to minimizing asymptotic estimation variance. The authors provide a sufficient and necessary graphical criterion for the existence of an optimal adjustment set under this notion. They also give an explicit and efficient construction of an adjustment set called the O set, which is proven to be optimal whenever graph optimality holds (the constraints in the graph are sufficient to determine an optimal adjustment set).

**Limitations And Societal Impact:**

The limitations of the proposed criteria and algorithm have been well explained, discussed and empirically verified.
I can foresee any societal impact that needs to be addressed at this time.

**Main Review:**

The results presented by the authors are novel and advance previous work by considering the hidden variable setting and characterizing the existence of optimal adjustment sets for estimators satisfying certain information-theoretic properties. It appears relevant work has been appropriately cited and discussed.

As far as I can tell, the results and claims presented are supported both theoretically and experimentally. The authors are very clear about the assumptions needed for their results to hold and also consider its limitations. Experiments consider both scenarios where the assumptions hold and where they don't; making sense of observed results.

I find the paper well written. Relevant previous work, definitions, and results are introduced clearly and making good use of several examples. The experimental section and further discussion help understand the benefits of using the proposed result and its limitations. I was wondering why the experimental section only considers the case where $\mathbf{S}=\emptyset$.

Overall, the paper makes important theoretical and practical contributions to the problem of identifying causal effects, using covariate adjustment, in settings where the graphical structure is not fully known. In particular, the results advance the state of the art for choosing and constructing optimal adjustment sets for certain estimators.

**Time Spent Reviewing:**

7

---

> ### Author Response · Authors · 2021-08-06
> **Response to review**
>
> Dear Reviewer 9L1v,
>
> thank you very much for your positive review stating that "the paper makes important theoretical and practical contributions to the problem of identifying causal effects"!
>
> You wondered why we only considered the case for S=\emptyset in the experiments, i.e., with no *fixed* set of conditioning variables. We wanted to treat the case with S in the theoretical part since this makes it more general and better relates to previous work in Smucler et al 2020. However, this case did not appear as especially illuminating to us for the experiments. Some preliminary experiments show that the qualitative results are essentially the same. Since S is the same for all methods, its effect on performance is very similar for all considered adjustment sets. We therefore suggest to omit these experiments due to space constraints. We will, however, discuss this choice in the paper.
>
> One small bit of clarification regarding your sentence "...the [graph optimality] constraints in the graph are sufficient to determine an optimal adjustment set": Our theorem actually states not only sufficient, but also necessary conditions for graph optimality.
>
> I am happy to further respond if anything is still unclear.
>
> Kind Regards
> The authors

---

> > ### Comment · Reviewer_9L1v · 2021-08-23
> > **Re: Response**
> >
> > I thank the authors for their answers, which I find satisfactory.

---

### Official Review · Reviewer_uv7C · 2021-07-16

**Rating:** 7
**Confidence:** 4

**Summary:**

This paper provides necessary and sufficient graphical conditions for ‘optimal’ adjustment sets in causal graphical models with hidden variables, where the optimality is defined by a newly proposed metric, instead of the commonly used smallest asymptotic estimation variance.

**Limitations And Societal Impact:**

The authors have described the limitations in Section 4.

**Main Review:**

I think this paper needs to justify and further explain the proposed adjustment information (Definitions 1) and information-theoretical graphical optimality (Definition 2). Intuitively, an estimator is statistically preferred if it is less biased and less variable than others. In Section 1, the authors mentioned that maximizing the adjustment information formalizes the common intuition to choose adjustment sets that maximally constrain the effect variable and minimally constrain the cause variable. I am wondering whether and how this “common intuition” is related to statistically preferred estimators. (Note that, the variance is  a measure of variability, but not the only one.) Besides, please provide a more detailed explanation of the linkage between this “common intuition” and the definition of adjustment information.

**Time Spent Reviewing:**

7

---

> ### Author Response · Authors · 2021-08-09
> **Response to review**
>
> Dear Reviewer uv7C,
>
> thank you very much for your review! I am happy to clarify the two newly introduced notions (Def. 1 and 2) here (and in the revised version) and respond to your other points.
>
> Firstly, you mention that "an estimator is statistically preferred if it is less biased and less variable than others". In the causal inference framework we foremost seek for an estimator that is unbiased. Biases can come from two sources: (1) a misspecification of the functional form of the statistical fitting model (e.g., using linear regression for nonlinear dependencies) and (2) an invalid adjustment set which introduces a confounding bias (e.g., not including a relevant confounding variable in the regression model). The general adjustment criterion (lines 77ff.) states the requirements of a valid adjustment set. In this paper, and in the theory of adjustment sets in general, we only consider *valid* adjustment sets and assume that the statistical model is correct. Hence, all considered estimators are unbiased and the remaining question is how to find a set that is "less variable than others". Here we use the estimator's asymptotic variance since at least for the linear regression there exists an analytical relation between the estimator's variance and the covariance of the variables X,Y,Z (Eq.(7) with f(.) given in line 128). But you're right that other expressions might be preferred in other cases.
>
> Secondly, you ask how the "common intuition" (to choose adjustment sets that maximally constrain the effect variable and minimally constrain the cause variable) is related to statistically preferred estimators. Our adjustment information formalizes the common intuition and the relation between adjustment information and asymptotic estimator variance is explicated in Lemma 1: An adjustment set with higher adjustment information always has smaller asymptotic variance. This holds under Assumptions 2 which are fulfilled by the linear OLS regression estimator.
>
> Thirdly, you ask to "please provide a more detailed explanation of the linkage between this “common intuition” and the definition of adjustment information." Adjustment information is defined as (ignoring the S for simplicity)
>
> J_Z ≡ I_Z;Y|X − I_X;Z
>
> The common intuition requires to maximally constrain the effect variable and minimally constrain the cause variable. If Z contains a lot of information about the effect variable Y, i.e., Z well constrains Y, then I_Z;Y|X is large. Further, if Z minimally constrains the cause variable X, then I_X;Z is small. Hence, if I_Z;Y|X is large and I_X;Z is small, then the adjustment information J_Z is large. Eq (5) relates J_Z to the adjustment entropy which by Eq. (7) is directly related to the asymptotic estimator variance.
>
> Thank you for these remarks, I will include them in the revision which will help to improve the paper. I am happy to further respond if anything is still unclear.
>
> Kind Regards
> The authors

---

> > ### Comment · Reviewer_uv7C · 2021-08-15
> > **Reply to the authors**
> >
> > Thanks for the reply! It has addressed all my concerns, and I will change my score and vote for acceptance.

---

### Decision · Program_Chairs · 2021-09-27

**Decision:**

Accept (Spotlight)

**Comment:**

The paper aims to get around a known negative result in the theory of covariate adjustment due to Rotnitzky and Smucler: in hidden variable causal models, an adjustment set that minimizes the variance of the target parameter does not depend only on the model, but also on the element within the model (in other words, not only on the graph, but also on particular parameter settings).

To address this, the authors propose a criterion based on information theory that aims to maximally constrain the outcome, and minimally constrain the treatment. They then investigate cases when this criterion yields a variance-minimizing adjustment set.

While initially reviewer opinion on this paper was split, discussions with the authors led to a consensus that the paper is an interesting, novel, and worthwhile addition to NeurIPS proceedings.